# Multi-targeted therapy resistance via drug-induced secretome fucosylation

Mark Borris D Aldonza[1,2,3,4,5†], Junghwa Cha[6†§], Insung Yong[6], Jayoung Ku[1], Pavel Sinitcyn[7], Dabin Lee[4,5], Ryeong-Eun Cho[1], Roben D Delos Reyes[8], Dongwook Kim[3,5#], Soyeon Kim[9,10], Minjeong Kang[1], Yongsuk Ku[1], Geonho Park[1], Hye-Jin Sung[3,5#], Han Suk Ryu[11], Sukki Cho[12], Tae Min Kim[9,10], Pilnam Kim[6,13*‡], Je-Yoel Cho[3,4,5*‡], Yoosik Kim[1,13,14,15*‡]

[1]Department of Chemical and Biomolecular Engineering, Korea Advanced Institute of Science and Technology (KAIST), Daejeon, Republic of Korea; [2]Department of Biological Sciences, KAIST, Daejeon, Republic of Korea; [3]Department of Biochemistry, College of Veterinary Medicine, Seoul National University, Seoul, Republic of Korea; [4]Comparative Medicine Disease Research Center (CDRC), Seoul National University, Seoul, Republic of Korea; [5]BK21 PLUS Program for Creative Veterinary Science Research and Research Institute for Veterinary Science, Seoul National University, Seoul, Republic of Korea; [6]Department of Bio and Brain Engineering, KAIST, Daejeon, Republic of Korea; [7]Computational Systems Biochemistry Research Group, Max Planck Institute of Biochemistry, Martinsried, Germany; [8]Department of Electrical Engineering, KAIST, Daejeon, Republic of Korea; [9]Department of Internal Medicine, Seoul National University Hospital, Seoul, Republic of Korea; [10]Cancer Research Institute, Seoul National University College of Medicine, Seoul, Republic of Korea; [11]Department of Pathology, Seoul National University Hospital, Seoul National University College of Medicine, Seoul, Republic of Korea; [12]Department of Thoracic and Cardiovascular Surgery, Seoul National University Bundang Hospital, Seongnam, Republic of Korea; [13]KAIST Institute for Health Science and Technology (KIHST), KAIST, Daejeon, Republic of Korea; [14]KAIST Institute for BioCentury (KIB), KAIST, Daejeon, Republic of Korea; [15]BioProcess Engineering Research Center and Bioinformatics Research Center, KAIST, Daejeon, Republic of Korea

**\*For correspondence:**
pkim@kaist.ac.kr (PK);
jeycho@snu.ac.kr (J-YC);
ysyoosik@kaist.ac.kr (YK)

†These authors contributed equally to this work
‡These authors also contributed equally to this work

**Present address:** §Department of Bioengineering, University of California, Berkeley, United States; #ProtanBio, Seoul National University, Seoul, South Korea

**Abstract** Cancer secretome is a reservoir for aberrant glycosylation. How therapies alter this post-translational cancer hallmark and the consequences thereof remain elusive. Here, we show that an elevated secretome fucosylation is a pan-cancer signature of both response and resistance to multiple targeted therapies. Large-scale pharmacogenomics revealed that fucosylation genes display widespread association with resistance to these therapies. In cancer cell cultures, xenograft mouse models, and patients, targeted kinase inhibitors distinctly induced core fucosylation of secreted proteins less than 60 kDa. Label-free proteomics of N-glycoproteomes identified fucosylation of the antioxidant PON1 as a critical component of the therapy-induced secretome (TIS). N-glycosylation of TIS and target core fucosylation of PON1 are mediated by the fucose salvage-FUT8-SLC35C1 axis with PON3 directly modulating GDP-Fuc transfer on PON1 scaffolds. Core fucosylation in the Golgi impacts PON1 stability and folding prior to secretion, promoting a more degradation-resistant PON1. Global and PON1-specific secretome de-N-glycosylation both limited the expansion of resistant clones in a tumor regression model. We defined the resistance-associated transcription factors (TFs) and genes modulated by the N-glycosylated TIS via a focused and transcriptome-wide analyses. These genes characterize the oxidative stress, inflammatory niche, and unfolded protein response as important factors for this modulation. Our findings demonstrate that core fucosylation

is a common modification indirectly induced by targeted therapies that paradoxically promotes resistance.

## Editor's evaluation

This study demonstrates that elevated secretome fucosylation is a pan cancer signature of both response and resistance to multiple FDA approved targeted therapies using both disease relevant cell lines and in vivo model systems. The authors go on to identify the antioxidant protein PON1 as a critical regulator of therpy induced changes in the secretome. Lastly, the authors define the resistance associated transcription factors and gene modulated by changes in the secretome. Collectively, these studies have the potential to define the mechanisms of drug resistance and identify novel targetable pathways for cancer treatment.

## Introduction

Complete responses to targeted therapies remain rare for a vast majority of cancer patients (*Krause and Van Etten, 2005*). While long-term disease stabilization can be achieved by therapeutic inhibition of oncogenic drivers, resistance to this targeted strategy is inevitable (*Krause and Van Etten, 2005*; *Baselga, 2006*; *Lin and Shaw, 2016*). In the clinic, partial remission can be achieved by classes of inhibitors that target amplified or mutationally activated kinases such as EGFR mutations or ALK translocations in lung adenocarcinoma, BRAF mutations in melanoma, or HER2 amplifications in breast cancer (*Lin and Shaw, 2016*; *Chapman et al., 2011*; *Arteaga et al., 2011*). Both genetic and non-genetic mechanisms of resistance to these inhibitors exist (*Salgia and Kulkarni, 2018*). However, the innate nature of many of these resistance acquisition models precludes the critical role of the tumor microenvironment (TME) in contributing to an incomplete tumor regression after therapy. For instance, a complex network of secreted signals from drug-stressed tumors termed therapy-induced secretomes (TIS) was shown to facilitate the selective expansion of a small number of pre-existing resistant clones, paradoxically explaining relapse to targeted therapy (*Obenauf et al., 2015*). Systemic understanding of this therapy-induced niche could lead to a paradigm shift in our current management of clinical drug resistance in cancer.

The cancer secretome comprises a set of secreted proteins that is pro-tumorigenic in nature. Many components of this secretome serve as disease biomarkers and are major druggable targets (*Robinson et al., 2019*). Both classical and non-classical pathways regulate the secretion of these components including extracellular matrix proteins, exosomes, growth factors, cytokines, shed receptors, and proteases (*Robinson et al., 2019*; *Hanash et al., 2008*; *Kalluri and LeBleu, 2020*). During stress, these secretome components are remodeled depending on tissue architecture and cell composition of the TME, stress-inducing stimuli, or conditions that affect liver homeostasis—a systemic dictator of the secretome and plasma proteome states (*Gupta and Massagué, 2006*; *Uhlén et al., 2015*). Substantially, secreted soluble proteins undergo post-translational modifications (PTMs) that functionally predominate their trafficking, stability, and folding prior to secretion (*Barlowe and Miller, 2013*). These PTMs in the secretory pathway are constantly employed to form tumorigenic niches upon chemotherapy, radiotherapy, targeted therapy, or immunotherapy (*Barlowe and Miller, 2013*; *Madden et al., 2020*; *Pitt et al., 2016*). Among these PTMs, phosphorylation and glycosylation are the most common. Glycosylation—the covalent addition of sugar moieties to target scaffolds—is the most abundant PTM of the secretome, as nearly all secreted mammalian proteins have at least one glycan, a sugar-based assembly, attached to them at a specific site (*Spiro, 2002*; *Reily et al., 2019*). For example, therapy-induced apoptotic disassembly of the Golgi is associated with the anomalous synthesis of specific glycan types (*Zhang and Wang, 2016*; *Wlodkowic et al., 2009*). In some cases, direct glycosylation of apoptotic signals upon therapy can restrain or trigger their cell-killing capacity (*Lichtenstein and Rabinovich, 2013*). Moreover, therapies that act as endoplasmic reticulum (ER) stressors can inhibit protein glycosylation and reduce disulfide bonds initiating an unfolded protein response (UPR) (*Cubillos-Ruiz et al., 2017*; *Costa et al., 2020*). While there is little evidence suggesting a post-ER quality control that operates at the Golgi following UPR, stress-induced regulation of terminal glycosylation is a complementary mechanism of Golgi-localized machinery that predominates the assembly of newly synthesized secretory proteins (*Pothukuchi et al., 2019*).

An abnormal glycome is a cancer hallmark (*Pinho and Reis, 2015*). Cancer-specific changes in two of the most frequent glycosylation types, O- and N-linked glycosylation, are coordinated with expression of genes encoding for glycosyltransferases–enzymes that catalyze glycosidic linkages–and glycosidases–enzymes that cleave glycosidic bonds–and their localization within the secretory pathway (Golgi apparatus and ER) (*Pinho and Reis, 2015*). We note that expressions of other enzyme-coding genes–many are known to mediate congenital disorders of glycosylation (i.e. those involved in sugar metabolism and transport and glycan sulfation)–are also relevant for understanding aberrant glycosylation in this context (*Leroy, 2006*). Throughout malignant transformation, unique alterations in both glycan level and composition, their conjugation and linkages, are reflected in the cell surface, intracellular, and extracellular scaffolds of mostly lipids and proteins (*Lauc et al., 2016*; *Pearce, 2018*). Lewis antigens, components of exocrine epithelial secretions, are among the most frequently overexpressed fucosylated epitopes during carcinogenesis (*Blanas et al., 2018*). Most obviously, this is attributed to the extensive activity of glycosyltransferases, mainly by fucosyltransferases (FUTs; *Schneider et al., 2017*). However, more nuanced and complicated dysregulations can arise from incomplete synthesis–truncated glycosylation common in early carcinogenesis–or neo-synthesis–de novo production of atypical glycosylation patterns–which are mediated by a complex interplay of glycosyltransferases such as FUTs and other factors that regulate fucose metabolism in the Golgi/ER (*Reily et al., 2019*; *Keeley et al., 2019*). As a result, several types of Lewis antigens, including sialylated Lewis structures, are currently being utilized in the clinic as prognostic cancer biomarkers (*Blanas et al., 2018*; *Keeley et al., 2019*). Given that these glycan alterations influence the cancer secretome, therapy-induced remodeling of the local TME, particularly its secreted components, must involve modified functionalities in the multi-step process of glycosylation.

Here, we identify that core fucosylation, modification at the N-glycan core, is a major post-translational signature of the pan-cancer TIS. Using pharmacogenomics, label-free proteomics, and a panoply of perturbation assays, we reveal that the therapy-induced aberration in secretome fucosylation involves (i) a differential induction of relatively smaller fucosylated proteins (<60 kDa), (ii) α1,6-fucosyltransferase (FUT8)-dependent transfer of GDP-β-L-fucose (GDP-Fuc) onto N-glycan core structures in the Golgi compartment, (iii) expression of fucose salvage genes and the GDP-Fuc transporter SLC35C1, and most significantly, (iv) core fucosylation of the antioxidant paraoxonase 1 (PON1). By utilizing several cellular and xenograft mouse models of drug resistance paired with patient specimens, we show that an elevated secretome fucosylation is likely a complementary mechanism of cancer relapse and targeted therapy resistance. In addition to uncovering the regulation of this TIS modification, we tested the functional consequences of generally blocking secretome core fucosylation or specifically constraining fucosylated PON1. Indeed, secretome de-N-glycosylation by a glycosidase, fucosylation inhibition by FUT8 or SLC35C1 RNA interference (RNAi), or site-specific blockade of PON1 core fucosylation dramatically prevented TIS-directed rebound of minority resistant clone population in a regressing heterogeneous cell pool. Furthermore, a targeted screen and transcriptome-wide gene expression analysis unveil effectors of redox stress sensing, inflammation regulation, and the UPR as secretome fucosylation-specific resistance modulators. Our findings point to a new view of the TIS that extends its role in establishing a resistance-promoting microenvironment niche via core fucosylation.

## Results

### Core fucosylation of therapy-induced cancer secretomes

While fucose is naturally present in a variety of glycolipids and glycoproteins, fucose moieties on N-glycans of secreted proteins are often dysregulated in cancer and are among the most aberrant sugar moieties of cancer glycoproteomes (*Spiro, 2002*). How therapies alter their on-site linkages and regulate their overall levels remain obscure. We investigated whether fucosylation is correlated with drug sensitivity by comprehensive mining of available data on genes involved in fucose metabolism (FUK, FPGT, FX, GMDS), fucosylation branching (FUTs, protein O-fucosyltransferases [POFUTs]), and GDP-Fuc transport (SLC35C1) in the Genomics of Drug Sensitivity in Cancer (GDSC, https://www.cancerrxgene.org/) and the Cancer Cell Line Encyclopedia (CCLE, https://sites.broadinstitute.org/ccle), two of the largest publicly available pharmacogenomics data sets (*Iorio et al., 2016*; *Barretina et al., 2012*). We first evaluated the consistency of the pharmacogenomic data from the two datasets

in the context of our query. Comparative analysis using the correlation between FUT gene expression and overall drug sensitivity (IC50 for GDSC and area under the curve, AUC for CCLE) as a metric showed that the molecular data are in concordance despite the apparent differences in cell lines and drug components (*Figure 1—figure supplement 1*). Although we should emphasize that there are obvious variabilities between the two datasets (i.e. variation in FUT expression values) that should be taken into consideration which might be the result of different cell lines representing a cancer lineage or/and assay protocols. Regardless, the consistent correlation between FUT expression and drug sensitivity reiterates the findings of previous efforts that looked into the reproducibility and biological consilience between profiling data from GDSC and CCLE (*Cancer Cell Line Encyclopedia Consortium and Genomics of Drug Sensitivity in Cancer Consortium, 2015*; *Haverty et al., 2016*).

Upon clustering of cell-line-derived data into 30 cancer types, we determined a univariate correlation between gene expression and a summary drug response measure (based on IC50 or AUC means). Spearman's correlation coefficient indicated that there is a variable but widespread association between fucosylation gene expression and drug resistance in both data sets (*Figure 1A* and *Figure 1—figure supplement 2A*). Of interest in terms of its consistent high pan-cancer expression profile in both data sets is FUT8—notably the only enzyme-encoding gene known to directly mediate core fucosylation via N-linkages (*Yang et al., 2017*). To scrutinize whether the correlation between FUT8 expression and drug resistance is significantly cumulated in drug-resistant cells, we categorized cell lines that are either sensitive or resistant based on the generalized drug response measurement and determined their correlation per class of drugs (*Figure 1B*). Indeed, FUT8 broadly correlated with resistance to a variety of compounds but more strongly to inhibitors of receptor tyrosine kinase (RTK), epidermal growth factor receptor (EGFR), and insulin-like growth factor receptor (IGFR). Across all compound types, resistance to targeted therapies displayed the strongest correlation with FUT8 expression. Moreover, cell lines that contain mutations near or specifically at GDP-Fuc binding sites (resulting in amino acid change that eliminates or decreases fucosylation) in FUTs or other fucosylation genes collectively exhibited higher sensitivity to drugs (*Figure 1A* and *Figure 1—figure supplement 2B and C*).

In a separate analysis of the Cancer Therapeutics Response Portal (CTRP), a large-scale small-molecule sensitivity data set, using the Computational Analysis of Resistance (CARE) scoring algorithm (*Jiang et al., 2018*), we showed that fucosylation gene expression displays a significant correlation with resistance to kinase inhibitors (data on at least 84 drugs; *Figure 1—figure supplement 2D and E*). In addition, using publicly available microarray and RNA-seq data, we found that high expressions of FUK, SLC35C1, and FUT8 are generally correlated with poor first progression or relapse-free survival (RFS) in various cancer patient cohorts (*Figure 1—figure supplement 3*).

Given that many of the target N-glycoprotein scaffolds of FUT8-mediated fucosylation are secreted (*Yang et al., 2017*), we next asked whether the association between fucosylation gene expression and drug resistance is interrelated with expression changes in the components of the core cancer secretome (CCS). Using defined component gene sets for CCS and protein glycosylation (*Robinson et al., 2019*), we observed coordinated pan-cancer increase or decrease of CCS and wide-ranging increase in expression of genes associated with glycosylation in general (*Figure 1C*). It is important to note that the glycosylation gene set contains subsets of annotated gene classes involved in secretome glycosylation (i.e. FUTs, solute carriers, positive/negative regulators of glycosylation in the Golgi). To add resolution to this analysis, we also evaluated two of the largest glycosylation subsets in the dataset, protein O- and N-linked glycosylation. Similarly, there is an extensive pan-cancer gene expression increase in both groups (*Figure 1C*). Overlapping genes between CCS and glycosylation significantly correlated with resistance to both targeted cytotoxic drugs, which may indicate that glycosylation of CCS components predicates drug sensitivity states.

Regulation of the DNA methylome influences the N-glycomes of the cancer secretome and plasma proteome (*Saldova et al., 2011*; *Wahl et al., 2018*). Curious as to how promoter methylation of FUTs can associate with drug sensitivity, we analyzed the methylation status at 1 kb upstream of the transcription start sites (TSS) of each FUT (since this TSS proximal region often are loci for dense hyper- and hypo-methylation in cancer cell lines) (*Jones, 2012*) and queried drug sensitivity data in the GDSC. The overall fraction of FUT methylated loci varied across tumor types (*Figure 1D*). As predicted, we observed a significant negative correlation between FUT mRNA expression and promoter methylation. While the association between FUT methylation and drug sensitivity appears indiscriminately,

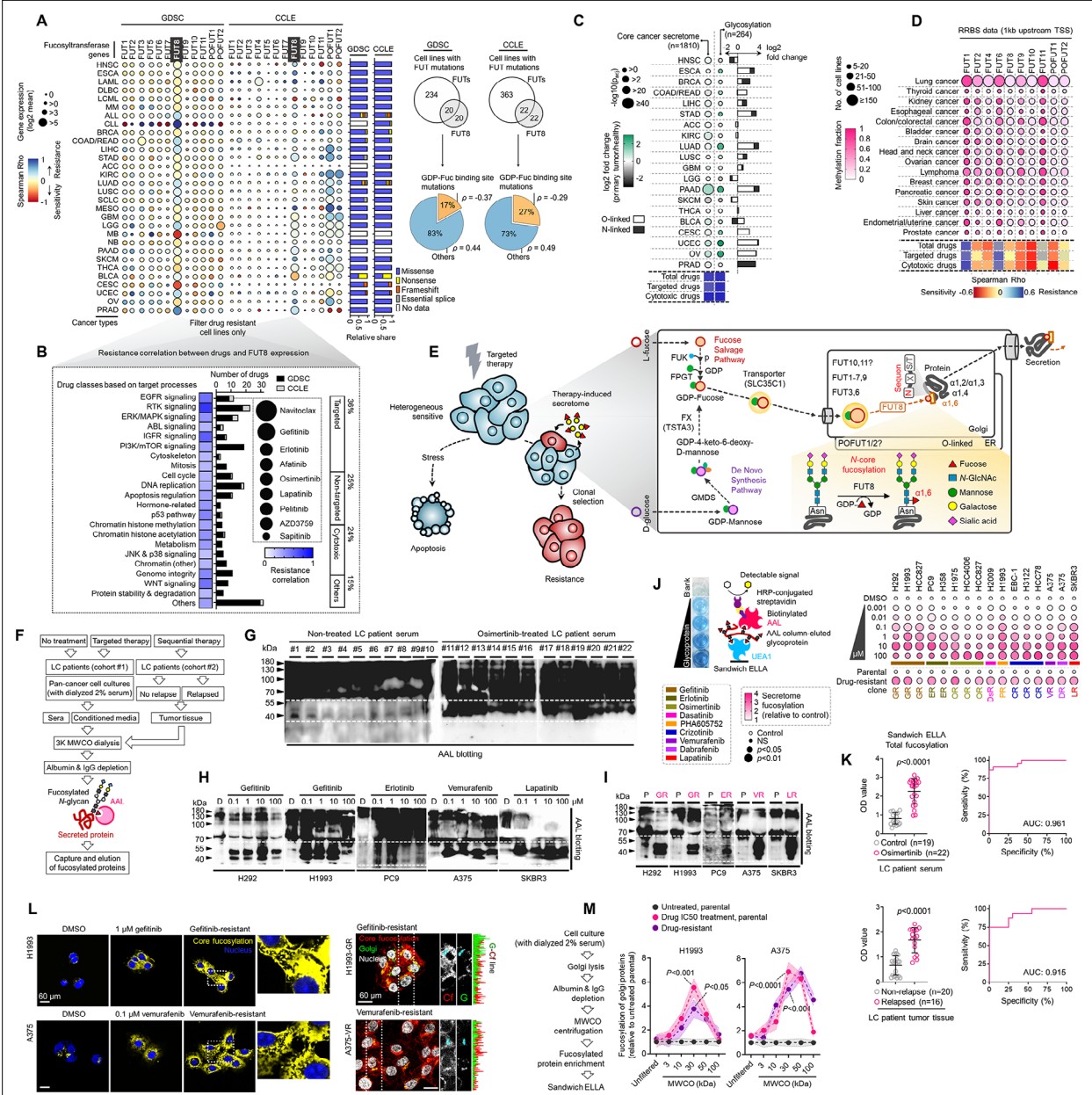

**Figure 1.** Secretome fucosylation is a post-translational mechanism associated with targeted therapy resistance in cancer. (**A**) Dot plot visualization of correlation between indicated FUT gene expression and drug response per cancer type screened in GDSC and CCLE. Size of circle refers to mean log2 gene expression while color corresponds to Spearman's rank coefficients. Per-sample estimates of area under the fitted dose-response curve were used as a metric of drug response per cell line. Only statistically significant correlations are shown (p<0.05). Beside is a relative mean proportion of mutational signatures of all FUT genes per cancer type queried in GDSC and CCLE. FUT mutations were classified as "GDP-Fuc binding site mutations" if any mutations (amino acid change) occurred near (±5 amino acid position) or at the annotated GDP-Fuc binding sites. Domain information was queried in UniProt. Spearman's rank coefficients (correlation between FUT expression and drug response) were calculated in cell lines carrying these mutations as opposed to those that do not ('others'). Note that FUT4 data is not available in the GDSC dataset. (**B**) Positive correlation between FUT8 gene expression and resistance to drugs grouped accordingly per target process in GDSC. Data from both GDSC and CCLE are summarized. Color represents Spearman's rank coefficients per target process. Only statistically significant correlations are shown (p<0.01). Bars indicate the number of drugs per class while the size of the circle corresponds to relative Spearman's rank coefficients per drug. Beside is a proportion of drug categories (GDSC classification) from all drugs with resistance profiles positively correlated with FUT8 expression. (**C**) Dot plot visualization of differential (TCGA primary tumor versus paired normal) CCS and overlapped glycosylation gene set expressions (including O-/N-linked glycosylation) per cancer type. Size of circle refers to adjusted -log10 p-value while color corresponds to log2 fold change in expression. Statistically significant (p<0.05) Spearman's correlation between drug sensitivity and CCS or glycosylation expression derived from GDSC are shown as a heatmap. In total, 169 drug profiles were queried; 33 are targeted, and 10 are cytotoxic drugs. (**D**) Dot plot visualization of mean promoter methylation fraction 1 kb upstream of the TSS

*Figure 1 continued*

of indicated FUT genes per cancer type from CCLE RRBS dataset. Size of circle refers to the number of screened cell lines while color corresponds to FUT promoter methylation. Only statistically significant changes are shown (p<0.05). Correlation between drug sensitivity and methylation is shown as heatmap as in C. (**E**) Schematic of secretome N-glycoprotein core fucosylation in the context of cancer TIS. (**F**) Preparation of patient- and cell culture-derived samples for capture and enrichment of fucosylated proteins and downstream fucosylation assays. (**G**) AAL blot analysis of total fucosylation in indicated crude patient sera prepared as in F. Representative of two independent experiments. Samples were originally performed in a single midi-SDS-PAGE format and blots were incised per sample group prior to incubations (samples #1–10, #11–16, and #17–22). Re-run of select samples in a single SDS-PAGE format, equal loading controls, and AAL specificity are presented in *Figure 1—figure supplement 4*. (**H and I**) AAL blot analysis of total fucosylation in indicated secretomes from sensitive cells (**H**) following treatment with or without indicated drugs for 48 hr or DR clones (**I**). Samples were prepared as in F. Representative of two independent experiments. Blot incisions per cell line or pair are shown. Equal loading controls and AAL specificity are presented in *Figure 1—figure supplement 4*. (**J**) Dot plot visualization of fucosylation characterization by sandwich ELLA in indicated secretomes from sensitive cells or DR clones prepared as in F following treatment with or without indicated drugs for 48 hr. Color indicates fold change values relative to DMSO or parental (means ± SD of three biological replicates) while size indicates p values; Student's *t*-test. NS, not significant. Schematic of in-house developed sandwich ELLA and representative colorimetric output are displayed on the left panel. (**K**) Fucosylation characterization by sandwich ELLA in indicated crude patient sera (top) or tissues (bottom) prepared as in F. Each point indicates mean absorbance at 450 nm from two to three replicates. Representative of two independent experiments. ROC curves are shown. For statistical analysis, the nonparametric Kruskal-Wallis test was used. (**L**) Representative confocal images of indicated sensitive cells or DR clones stained for: core fucosylation (fluorescein-conjugated AAL; yellow) and nuclei (DAPI; blue) or golgi (RCAS1; green), core fucosylation(fluorescein-conjugated AAL; red), and nuclei (DAPI; blue). Cells were treated with and without 1 µM gefitinib or 0.1 µM vemurafenib for 48 h. The co-localization histogram plot of the indicated line is shown. Representative of two independent experiments. (**M**) Fucosylation characterization by sandwich ELLA in indicated golgi-fractionated cell lysates from sensitive cells or DR clones prepared as in schematic following treatment with or without gefitinib (for H1993) or vemurafenib (for A375) IC50 for 48 hr and filtered according to their indicated nominal molecular weight limit (NMWL). Values are relative to untreated parental (means ± SD of two biological replicates). For statistical analysis, a Student's t-test was used.

The online version of this article includes the following source data and figure supplement(s) for figure 1:

**Source data 1.** Uncropped blots (labeled and unlabeled) for *Figure 1G, H and M*.

**Source data 2.** Uncropped blots and gels (labeled and unlabeled) for *Figure 1—figure supplement 4A,B and C*.

**Source data 3.** Uncropped blots (labeled and unlabeled) for *Figure 1—figure supplement 11C*.

**Figure supplement 1.** Comparison of correlations between FUT expression profiles and drug sensitivity and other shared components in GDSC and CCLE datasets.

**Figure supplement 2.** Correlation between fucose salvage and de novo synthesis gene expression and drug sensitivity.

**Figure supplement 3.** Fucose salvage gene expression is associated with cancer patient relapse and poor survival during/after therapy.

**Figure supplement 4.** Equal loading controls and AAL specificity.

**Figure supplement 5.** Therapy resistance-associated N-glycan enrichment in >30 kDa proteins.

**Figure supplement 6.** Characterization of DR clones.

**Figure supplement 7.** Sandwich ELLA.

**Figure supplement 8.** Expression and activity of fucose salvage genes and fucosyltransferases are associated with cancer patient relapse, therapy-induced apoptosis, and multiple acquired targeted therapy resistance.

**Figure supplement 9.** FUT8 and SLC35C1 confer resistance to kinase inhibitors in sensitive cancer cells.

**Figure supplement 10.** Therapy resistance-associated core fucosylation of proteins between 30 and 100 kDa is regulated by FUT8 or SLC35C1.

**Figure supplement 11.** Systemic secretome fucosylation is associated with drug resistance and residual tumor mass persistence after targeted therapy.

**Figure supplement 12.** Distinct fucosylation in mouse tumor xenografts.

FUT methylation profiles contradicted the correlation between FUT gene expression and resistance. In other words, cancer types exhibiting higher FUT methylation are more sensitive to targeted therapies with the exception of FUT1 and FUT6 (*Figure 1D*), suggesting that cancer cells can inhibit fucosylation upon increased methylation of FUT promoter are more susceptible to therapy. Albeit in preliminary stage, our analysis supports the growing evidence on epigenetic regulation of glycosylation-related genes, especially those that encode for glycosyltransferases, by DNA methylation with phenotypic consequences in cancer (*Horvat et al., 2011*), which in the context of our analysis might be important for modulating targeted therapy response.

Based on our analysis, we hypothesized that response and resistance to targeted therapies involve the systemic regulation of core fucosylation of CCS components (*Figure 1E*). We performed a potpourri of biochemical assays to characterize fucosylation in multiple cancer cell lines, cell secretomes, xenograft mouse and human patient sera and tissues (see *Supplementary file 1—supplementary file*

*1a and b* for LC patient cohort information). To enrich protein samples for core fucosylation, we used a lectin-conjugated bead capture strategy, where *Aleuria aurantia* lectin (AAL) served as the carbohydrate probe for core fucose (*Figure 1F*; see Materials and methods). Remarkably, lectin blotting revealed a distinct signature of enriched core fucosylation of serum proteins between 30 and 60 kDa in lung cancer (LC) patients who received multiple cycles of osimertinib, a third-generation EGFR-tyrosine kinase inhibitor (TKI), compared to those of treatment-naive patients (*Figure 1G* and *Figure 1—figure supplement 4*). To quantitatively validate this result, we modified an N-glycan oxidation assay originally developed to assess the activity of PNGases in releasing N-linked oligosaccharide chains from glycosylated scaffolds. These cleaved N-glycans, upon deamination by water, possess hemiacetal moiety at their reducing terminus that is highly reactive to water-soluble WST-1, a tetrazolium salt dye that serves as an oxidation agent for N-glycans. In this reaction, WST-1 is converted to a formazan, producing a colorimetric readout (see Materials and methods) (*Wang et al., 2019*; *Freeze and Kranz, 2010*). Due to its simplicity, we decided to adapt and optimize this assay to quantify the release of N-glycans from our samples using the glycoamidase PNGase F and glycosidases Endo S and F1. Following analysis of in-gel excised 30~60 kDa patient serum proteins, PNGase F-released N-glycans showed significantly higher levels in osimertinib-treated patients compared to treatment-naive patients (*Figure 1—figure supplement 5A*), while this apparent difference was considerably moderated when N-glycans were released by either Endo S or F1. While PNGase F can cleave all N-glycans, we assumed that the glycans released from our samples are mostly those that contain core fucose (cleavage at α1,6 site) because the subjected N-glycoproteins were captured using AAL (*Figure 1F* and *Figure 1—figure supplement 4B*). Thus, the reduction in detected N-glycans released by Endo S or F1 reflects a specificity in cleaving different N-glycans. Note that Endo S has a high specificity for removing N-glycans within the chitobiose core of native IgG, while Endo F1 cleaves high mannose and some hybrid type N-glycans (*Trimble and Tarentino, 1991*; *Collin and Olsén, 2001*; *Goodfellow et al., 2012*). The results potentially suggest that the cancer TIS from patients contains an elevated pool of both core fucosylated glycoproteins <60 kDa and N-glycans.

We next characterized fucosylation in cancer cell-derived secretomes to verify the differential secretome core fucosylation signature. Targeted kinase inhibition by EGFR-TKIs (gefitinib, erlotinib), HER2-TKI (lapatinib), or BRAFi (vemurafenib) differentially induced fucosylation of secreted proteins <60 kDa (*Figure 1H*). To extend these findings to models of therapy resistance, we generated 16 stable drug-resistant (DR) clones from various cancer types (lung adenocarcinoma, melanoma, and breast cancer) following stepwise evolution to appropriate targeted inhibitor pressures (*Figure 1—figure supplement 6*). Similarly, secretomes derived from DR clones displayed an induced <60 kDa protein fucosylation (*Figure 1I*). In addition, both 30~60 kDa TIS and secretome proteins from DR clones contained unanimously higher amounts of PNGase F-released N-glycans than those from DMSO or parental cell secretomes (*Figure 1—figure supplement 5B*), while N-glycans released by either Endo S or F1 did not discriminate the amounts from all samples mirroring our observations from the patient sera. Further, following targeted therapy, pooled >30 kDa N-glycoproteins from sensitive cells displayed increased release of fucosylated N-glycans even at very low drug concentrations (from 0.001 μM), particularly in hypersensitive cell lines (*Figure 1—figure supplement 5C*). These results mimic the osimertinib-induced core fucosylation signatures in LC patient sera.

To couple these results with an overall measure of fucosylation in various samples and perturbation models, we developed a sandwich enzyme-linked lectin assay with varying affinities for AAL-captured fucosylated proteins (sandwich ELLA; *Figure 1—figure supplement 7*; see Methods). Using *Ulex europaeus* agglutinin I (UEA1)-AAL-based sandwich ELLA, we measured core fucosylation of cell-derived secretomes (*Figure 1J*). TIS derived from cancer cells treated with targeted inhibitors of EGFR, BRAF, or HER2 signaling unanimously led to an elevated secretome fucosylation (*Figure 1J*). Likewise, all DR clone-derived secretomes showed increased fucosylation compared to secretomes derived from parental clones (*Figure 1J*). These are further accompanied by an overall core fucosylation increase in relapsed LC patient tissues after sequential therapy and osimertinib-treated LC patient sera (*Figure 1K*). Using receiver operating characteristic (ROC) curves, we investigated whether core fucosylation can discriminate between non-treated and osimertinib-treated LC patient sera or between non-relapsed and relapsed LC patient tissues. Total core fucosylation discriminated against the conditions with high sensitivity and specificity with the associated area under the curve (AUC) values of 0.961 and 0.915, respectively, based on sandwich ELLA measurements (*Figure 1K*).

To further substantiate these results, we analyzed tissues from small cohorts of breast cancer (BC) (see *Supplementary file 1—supplementary file 1c* for BC patient cohort information) and LC patients that received sequential multi-component therapy. Gene expression and enzyme activity analysis revealed that high expression of the fucose salvage pathway, FUT8, and SLC35C1 are strongly correlated with relapse (*Figure 1—figure supplement 8A and B*). There was an overall immediate increase (16 h post-treatment) in Golgi-localized core fucosylation in drug-stressed LC and melanoma cells and sustained activation in their respective DR clones (*Figure 1L*). We next profiled the expression of fucosylation genes in cancer cells with various oncogenic drivers upon apoptosis-inducing targeted therapy. While drug-induced expression changes varied between FUTs responsible for O- and N-linked glycosylation, there was a marked increase in FUT8 and SLC35C1 expression (*Figure 1—figure supplement 8C*), all of which are associated with apoptosis (3-day treatment; *Figure 1—figure supplement 8D and E*). In DR clones, both expressions are also amplified except with a pronounced fucose salvage pathway (*Figure 1—figure supplement 8F*). Because FUT8 is highly expressed in relapsed patient tumors and in both drug-stressed cells and DR clones, we probed its potential role in therapy resistance. We first analyzed independent, genome-wide RNAi screening data from the Cancer Dependency Map (DepMap, https://depmap.org/portal/) project (*Tsherniak et al., 2017*), which houses pan-cancer genetic vulnerability maps. FUT8 is not classified as an essential gene in both sensitive and resistant cancer cell lines (*Figure 1—figure supplement 9A*), despite marginally higher essentiality scores in TKI-resistant cells than sensitive cells (*Figure 1—figure supplement 9B*). Regardless, treatment with EGFR-TKI or BRAFi and selection for resistance both led to higher FUT8-dependent GDP-Fuc catalytic activity (*Figure 1—figure supplement 9C*). Non-lethal concentrations of nine kinase inhibitors induced FUT8 expression while near-lethal concentrations moderately mitigated this effect (*Figure 1—figure supplement 9D*). We then used RNAi to dissect the role of FUT8 upon targeted therapy. FUT8-targeting siRNAs augmented drug-induced cell killing and subsequent rescue was observed upon transfection with FUT8 cDNA (*Figure 1—figure supplement 9E*), all independent of cell proliferation (*Figure 1—figure supplement 9F*). We obtained similar results with SLC35C1 (*Figure 1—figure supplement 9G–I*).

Furthermore, well-known core fucosylated cancer biomarkers α-fetoprotein (AFP) and α–1-antitrypsin (A1AT), both >50 kDa, displayed systemic elevation in LC patient sera following osimertinib treatment and in secretomes of drug-stressed cells and DR clones, at least those expressing basal A1AT (*Figure 1—figure supplement 10A and B*). Using molecular weight cut-off (MWCO) filtration, we confirmed that concentrated secreted proteins of >30 kDa from targeted inhibitor-treated cells, their respective DR clones, and EGFR-TKI-treated LC patients display distinctively enriched core fucosylation and core α–1,6-linkages-containing N-glycans, but less so in >100 kDa pooled proteins (*Figure 1—figure supplement 10C and D*). These can be controlled by FUT8 or SLC35C1, at least shown in vitro (*Figure 1—figure supplement 10E*). Similar size-dependent effects were observed in the golgi fractions of targeted inhibitor-treated cells and their respective DR clones (*Figure 1M*). These results are consistent with the idea that direct or indirect mediators of core fucosylation confer resistance to targeted therapies.

To dissect whether these findings are supported in EGFR-TKI-resistant and relapsed tumors within a physiological environment in vivo, we subjected mouse xenografts derived from our H1993, PC9, and HCC827 models to EGFR-TKI therapy (*Figure 1—figure supplement 11A*). Aside from xenografted DR clones which are also resistant in vivo, residual parental tumors following EGFR-TKI therapy were also considered as 'resistant'. Using the same array of experiments to assay core fucosylation, we observed an overall increase in serum fucosylation and a distinct signature of enriched core fucosylated serum proteins between 30 and 60 kDa in EGFR-TKI-treated parental-xenografted mice compared to vehicle-treated mice (*Figure 1—figure supplement 11B and C*). Following analysis of in-gel excised 30~60 kDa serum proteins, EGFR-TKI therapy led to significant increase in PNGase-F-released N-glycans in parental-xenografted mice (*Figure 1—figure supplement 11D*), while minimal to no significant differences where observed when N-glycans were released by either Endo S or F1. Using MWCO filtration, we confirmed that these >30 kDa serum proteins from EGFR-TKI-treated parental-xenografted mice display distinctively enriched core fucosylation and core fucose-containing N-glycans (*Figure 1—figure supplement 11E*), but less so in >100 kDa pooled proteins. As expected, while DR clone-xenografted mice displayed an overall increase in serum core fucosylation and enrichment of core fucose-containing serum N-glycans compared to untreated parental-xenografted mice,

EGFR-TKI therapy did not induce significant changes in the DR setting, at least in sera. In xenograft tumor tissues, gene expression profiling showed variable FUT expressions but a unanimous marked increase in FUK-FPGT, FUT8, and SLC35C1 expressions associated with EGFR-TKI resistance (*Figure 1—figure supplement 11F*), reiterating the involvement of fucose salvage pathway in the systemic core fucosylation changes associated with TIS and EGFR-TKI resistance. Intriguingly, golgi-associated core and terminal fucosylation are both enriched in tumors of parental-xenografted and DR clone-xenografted mice following EGFR-TKI therapy (*Figure 1—figure supplement 12*).

Of note, while we consistently observed a marked 30~60 kDa core fucosylation of the TIS and drug resistance or relapse-associated secretomes across cell lines, xenografted mouse models, and LC patients, we noticed varying signatures in the core fucosylation of relatively larger proteins (>60 and >100 kDa) across samples. Whether this is a result of technical differences in the assays (i.e. discrepancy between AAL blots, sandwich ELLA, and N-glycan release assay) or an inherent phenotype of specific samples (i.e. depletion of fucosylated >100 kDa proteins in cell-derived TIS and DR clone secretomes but not in TKI-treated LC patients or mouse models), determinants of this variation can be confronted with more relevant experimental set-up and systems-level interrogation of glycosylated proteins and glycans such as glycoproteomics. Regardless, taken together, our results suggest that targeted therapies induce a prevalent pan-cancer secretome core fucosylation that is primarily regulated by the fucose salvage-SLC35C1-FUT8 pathway and is enriched in the Golgi prior to secretion. This therapy-induced modification presumably is an evolvable mechanism towards establishing resistance.

## Therapy resistance via drug-induced secretome fucosylation

Limited tumor regression upon targeted therapy implicates that the microenvironment undergoes remodeling to sustain the remaining tumor population (*Obenauf et al., 2015*; *Massagué and Obenauf, 2016*). TIS, which consists of soluble mediators from this remodeled niche, predominantly promotes the survival and outgrowth of remnant tumor cells fostering subsequent disease relapse (*Obenauf et al., 2015*). Considering that our data point to core fucosylation as a widespread PTM of the pan-cancer TIS, we proposed that de-N-glycosylation of the TIS prevents the outgrowth of residual DR tumor cells. To model a regressing tumor in vitro, we performed a multicolor homotypic 'one-pot' admixture assay by mixing a small percentage (1%) of red-tracker-labeled DR clones with a large pool (99%) of green-tracker-labeled sensitive cells in both 2D and 3D cultures. We then subjected these admixtures to targeted therapy, exogenously added PNGase F to de-N-glycosylate secretome proteins, and tracked the rebound of DR clones and regression of the sensitive cell pool (*Figure 2A*). Following the formation of a 3D tumor spheroid, the population of the admixed gefitinib-resistant (GR) clone gradually expanded (observable after day 1 and steady from day 5), while the sensitive cell population significantly decreased upon targeted therapy (*Figure 2B* and *Videos 1 and 2*; representative PC9 admixture in *Video 3*). Addition of PNGase F to these admixture secretomes led to a striking protein de-N-glycosylation in culture (*Figure 2C and D*). Therefore, therapy-induced regression of mostly sensitive cells and population expansion of admixed minority GR and erlotinib-resistant (ER) clones are tightly linked with increased secretome core fucosylation (*Figure 2E*).

Considering an abundant core fucosylation in all biologically active conditioned media (CM) occur before apoptosis or senescence and is enriched in the soluble secretome rather than apoptotic bodies, it is likely that cell-derived TIS and its N-glycosylation are actively produced as a result of targeted oncogene inhibition (*Figure 2—figure supplement 1*). In both 3D and 2D admixture assays, secretome de-N-glycosylation blocked the growth acceleration of the DR clone promoted by TIS in various cancer backgrounds and targeted therapy settings (*Figure 2F and G* and *Figure 2—figure supplement 2A*), delayed the S-phase cycle of residual cell populations, and promoted apoptosis (*Figure 2G* and *Figure 2—figure supplement 2B and C*). Consistently, in a CM co-culture assay, TIS stimulated the proliferation of low-density seeded DR clones while exposure to de-N-glycosylated TIS limited their outgrowth (*Figure 2—figure supplement 2D*). Of note, de-N-glycosylation in fresh media or DMSO CM did not affect DR clone proliferation (*Figure 2—figure supplement 2D*). At day 5, depletion of FUT8 or SLC35C1 efficiently blocked the expansion of DR clone population in a regressed cell admixture (*Figure 2H* and *Figure 2—figure supplement 2E*), suggesting that the similar effect afforded by PNGase F is via protein de-N-glycosylation. In such a circumstance, we observed depletion of fucosylation (in both apoptotic bodies and soluble secretome) and intracellular kinase

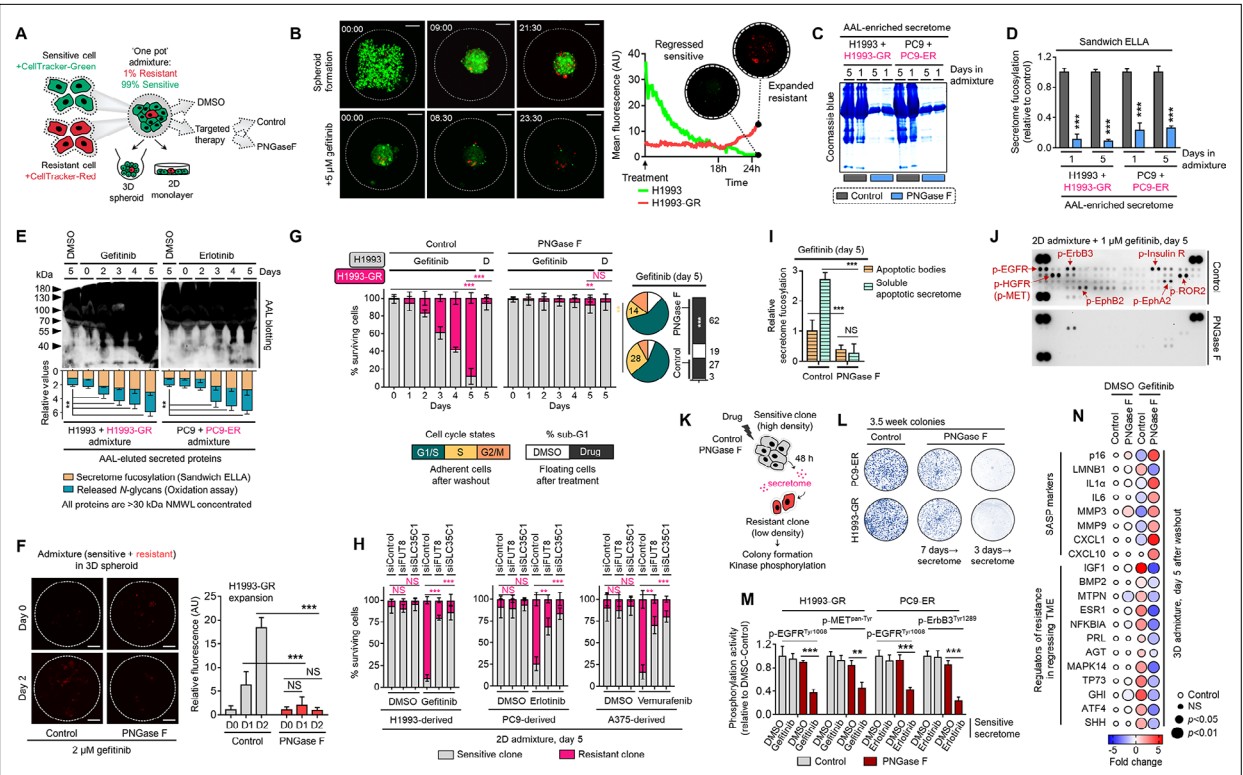

**Figure 2.** Secretome fucosylation promotes resistance rebound in regressing cell admixtures. (**A**) Schematic of multi-color cell tracker assay in 'one pot' admixture culture. (**B**) Representative live-imaging confocal images of indicated 3D tumor spheroid admixture prepared as in A and treated with or without 2 μM gefitinib for 24 hr. Scale bar indicates 100 μm. Mean intensity profiles of both fluorescently tagged cells are shown. Note that the timing of live-imaging is different between the top and bottom panels. In the top panel, imaging started prior to the formation and settlement of the spheroids. In the bottom panel, imaging started after the spheroids have formed and settled. See also *Figure 2—figure supplements 1–6*. (**C**) Representative Coomassie-stained SDS-PAGE gels showing fucosylated secretome proteins from indicated 3D cell admixtures prepared as in A, treated with 2 μM gefitinib or 0.1 μM erlotinib for 1 or 5 day/s, and incubated with or without 10 μg/mL recombinant PNGase F. Secretomes were concentrated using a>3 kDa NMWL filter. Representative of two independent experiments. (**D**) Characterization of fucosylation by sandwich ELLA in indicated cell admixture secretomes with conditions as in C. Values are relative to non-treated secretome (means ± SD of three biological replicates). ***p<0.001, Student's *t*-test. (**E**) Characterization of fucosylation by AAL blotting, sandwich ELLA, and N-glycan release assay in indicated 2D cell admixtures prepared as in A, treated with or without 1 μM gefitinib or 0.1 μM erlotinib, and incubated with or without 10 μg/mL recombinant PNGase F for up to 5 days. Secretomes were concentrated using a>30 kDa NMWL filter. Blots are representative of two independent experiments. Values are relative to day 0 (means ± SD of two biological replicates). **p<0.01, Student's *t*-test. (**F**) Representative confocal images of fluorescently-tagged GR clone in 3D cell admixtures prepared as in A, treated with 2 μM gefitinib, and incubated with or without 10 μg/mL recombinant PNGase F for 24 or 48 hr. Scale bar indicates 100 μm. Intensity profiles of tracker-tagged GR clone are shown. Values are relative to day 0 (means ± SD of three biological replicates). ***p<0.001, Student's *t*-test. NS, not significant. (**G**) Tracking of both fluorescently-tagged cells in 2D cell admixtures prepared as in A, treated with or without 1 μM gefitinib, and incubated with or without 10 μg/mL recombinant PNGase F for indicated times. Values are relative to day 0 (means ± SD of three biological replicates). Beside shows cell-cycle states of adherent cells and apoptosis of floating cells in indicated cell admixtures with same conditions at day 5. Cell cycle assays are representative of two independent experiments. **p<0.01, ***p<0.001, two-tailed Mann–Whitney *U* test. NS, not significant. (**H**) Similar tracking experiments as in G, except upon FUT8 or SLC35C1 RNAi in sensitive cells for 48 hr prior to admixing and culture for 5 days. H1993 admixture was treated with or without 1 μM gefitinib, PC9 admixture was treated with or without 0.1 μM erlotinib, and A375 admixture was treated with or without 0.1 μM vemurafenib. Values are relative to day 0 (means ± SD of two biological replicates). **p<0.01, ***p<0.001, two-tailed Mann–Whitney *U* test. NS, not significant. PNGase F controls are presented in *Figure 2—figure supplement 2E*. (**I**) Characterization of fucosylation by sandwich ELLA in indicated apoptotic debris and secretomes from the same cell admixtures as in G. Values are relative to control apoptotic debris (means ± SD of three biological replicates). ***p<0.001, Student's *t*-test. NS, not significant. (**J**) Phospho-RTK array of indicated cell admixtures in the same conditions as in G. The blots reflect the phosphorylation status of 49 RTKs. Each RTK is spotted in duplicate, and the three pairs of dots in each corner are positive or negative controls. Representative of two independent experiments. (**K**) Schematic of CM co-culture. (**L**) Colony formation of indicated DR clones prepared as in K. Representative of two independent experiments. (**M**) ELISA sandwich-based measurement of indicated RTK phosphorylation in indicated DR clones prepared as in K. Values are relative to DMSO (means ± SD of three biological replicates). **p<0.01, ***p<0.001, Student's *t*-test. (**N**) Dot plot visualization of indicated gene expression by qPCR analysis in 3D cell admixtures prepared as in A, treated with or without 2 μM gefitinib, and incubated with or without 10 μg/mL recombinant PNGase F for 5 days. Color indicates log-transformed fold change values relative to DMSO and normalized to GAPDH levels (means ± SD of three biological replicates) while size indicates p values; Student's *t*-test. NS, not significant.

*Figure 2 continued on next page*

*Figure 2 continued*

The online version of this article includes the following source data and figure supplement(s) for figure 2:

**Source data 1.** Uncropped blots and gels (labeled and unlabeled) for *Figure 2C, E and J*.

**Source data 2.** Uncropped gels (labeled and unlabeled) for *Figure 2—figure supplement 4A and D*.

**Figure supplement 1.** Drug-induced secretome fucosylation occurs before apoptosis and senescence.

**Figure supplement 2.** Resistance rebound and apoptosis in 2D admixtures.

**Figure supplement 3.** Effects of de-N-glycosylation in cell cultures.

**Figure supplement 4.** Cell membrane-bound N-glycoproteins are dispensable in promoting drug-resistant population expansion in drug-treated admixture cultures.

**Figure supplement 5.** Outgrowth of DR clones in regressing 3D spheroids is associated with expression of fucose salvage genes.

**Figure supplement 6.** TIS de-N-glycosylation promotes senescence in long-term grown DR clones.

---

phospho-proteome (*Figure 2I and J*). We corroborated these in a CM co-culture assay (*Figure 2K*), wherein de-N-glycosylated TIS prevented DR clones to form colonies and decreased kinase phosphorylation activity of EGFR, MET, and ErbB3, at least in GR and ER clones, respectively (*Figure 2L and M*). De-N-glycosylation by PNGase F in CM co-cultures (fresh media or CM from same cell/clone source) did not significantly influence the drug sensitivity of both sensitive cells and DR clones (*Figure 2— figure supplement 3*), except in sensitive cells cultured in their own de-N-glycosylated TIS, where there is a widespread drug sensitization (*Figure 2—figure supplement 3A*). These point to the idea that fucosylation of the TIS from drug-treated sensitive cells is critical to its survival-enhancing effects not only on DR clones but also in drug-sensitive cells. Across all cell lines and DR clones, PNGase F in-culture for up to 5 days did not affect cell proliferation (*Figure 2—figure supplement 3B*). We assumed that PNGase F in our cell admixture assays not only de-N-glycosylates secreted scaffolds but should also affect cell surface N-glycans. We inspected the potential changes on N-glycosylation of cell membrane proteins in our admixtures by pooling subcellular fractions (admixture set-up as in *Figure 2A*). At day 5, we only observed a significant increase in fucosylation from TIS and ER/ Golgi fractions, not from cell membrane fractions, of EGFR-TKI-treated H1993 and PC9 admixtures (*Figure 2—figure supplement 4A and B*). In addition, there were no changes in the <60 kDa fucosylation signature in cell membrane fractions of the admixtures, unlike the significant increase in ER/Golgi fraction (*Figure 2—figure supplement 4C*). Regardless, PNGase F effectively de-N-glycosylated cell membrane proteins in-culture of both EGFR-TKI-treated H1993 and PC9 admixtures (*Figure 2— figure supplement 4D and E*). Although we cannot completely rule out alternative possibilities, these results favor the idea that core fucosylation of the TIS, and not of membrane proteins, promotes the DR clone population expansion observed in our cell admixture experiments.

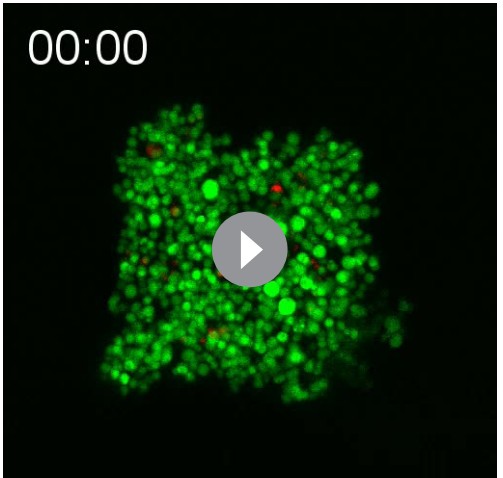

**Video 1.** Movie showing 3D spheroid formation of CellTracker-Green-labeled H1993 and CellTracker-Red-labeled H1993-GR admixture within 24 hr.

https://elifesciences.org/articles/75191/figures#video1

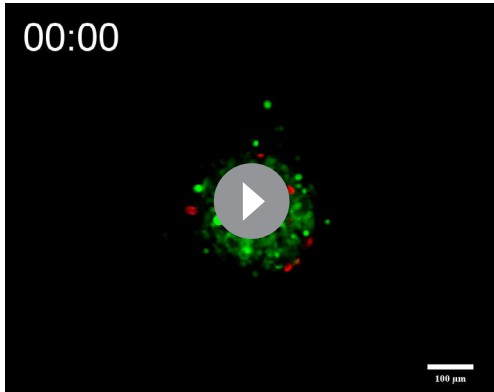

**Video 2.** Movie showing live-imaging of CellTracker-Green-labeled H1993 and CellTracker-Red-labeled H1993-GR admixture upon treatment with 2 μM gefitinib within 24 h.

https://elifesciences.org/articles/75191/figures#video2

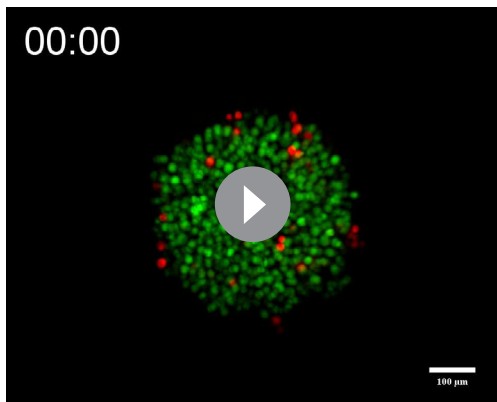

**Video 3.** Movie showing live-imaging of CellTracker-Green-labeled PC9 and CellTracker-Red-labeled PC9-ER admixture upon treatment with 0.1 µM erlotinib within 24 h.
https://elifesciences.org/articles/75191/figures#video3

Next, we established a 'sequentially layered' 3D spheroid in vitro co-culture and monitored the growth of red fluorescent protein (RFP)-expressing DR clones (H1993-GR, PC9-ER) in the absence or presence of sensitive cells treated with kinase inhibitors or vehicle (*Figure 2—figure supplement 5A*). Resembling our initial findings, co-culture with EGFR-TKI (gefitinib or erlotinib)-treated sensitive cells significantly promoted the growth of DR clones while the addition of PNGase F in the culture pronouncedly led to their growth retardation (*Figure 2—figure supplement 5B and C*). In these 3D admixtures at day 5, TIS de-N-glycosylation triggered the senescence-associated secretory phenotype (SASP) and impeded the gene expression of factors previously described to promote resistant cell outgrowth in a regressing TME (*Obenauf et al., 2015*; *Figure 2N*). It appears that the response of DR clones in these admixtures upon

TIS de-N-glycosylation is independent of fucosylation gene activity since there were no marked changes in expression (*Figure 2—figure supplement 5D*). Notably, long-term passaging and culture of DR clones in de-N-glycosylated TIS initiated a senescence response shown by strong senescence-associated β-galactosidase (SA-β-gal) activity, SASP activation, and arrested growth (*Figure 2—figure supplement 6*), elucidating the inhibited proliferative capacity of these clones in cell admixtures upon TIS de-N-glycosylation. These results demonstrate that the rebound of the DR clone population in a model of tumor regression is dependent on fucosylated scaffolds of the TIS.

## PON1 fucosylation is a critical feature of therapy-induced cancer secretomes

To identify relevant components of the TIS- and DR clone-specific N-glycomes, we performed label-free in-gel proteomics using AAL-captured 30~70 kDa secretome proteins derived from H1993 cells treated with or without gefitinib or the GR clone (*Figure 3A*). Our analysis retrieved a fairly reproducible amount of peptide sequences per sample, which we used for downstream target identification (*Figure 3—figure supplement 1A*). Base peak chromatogram revealed differential mass ranges in all samples, with a relatively higher overlapping similarity between gefitinib-treated cells and GR clone (*Figure 3—figure supplement 1B*). Because of the preliminary culture (i.e. 2% serum) and stress condition (drug treatment) requirements to produce TIS and semi-quantitative nature of our screen, many proteins identified by this method are expected to be 'contaminants' derived from non-secreted apoptotic proteins, serum proteins (trypsin, albumin, keratin), uncharacterized proteins, immunoglobulins, and proteins below or above the range of excised in-gel sections (<30 and >70 kDa). As expected, we derived >60% 'contaminant' protein coverage. We filtered these out and retained proteins that are only classified as 'secretory' or 'extracellular' based on the annotation criteria by UniProt (i.e. possession of N-terminal signal sequence), yielding a total of 57 unique, secretory-predicted proteins across the three conditions. Gene ontology (GO) analysis showed significant enrichment of biological processes (BPs) implicated in stress response, secretory pathway, and protein maturation in the ER/Golgi (*Figure 3B*). Interestingly, BPs related to the metabolic regulation of oxidative stress were significantly overrepresented. Following the selection of overlapping fucosylated secretome proteins between the gefitinib-treated H1993 cells and GR clone, we identified 11 top hits using two different quantitative approaches (label-free quantification [LFQ] and intensity-based absolute quantification [iBAQ]). Many of these hits are serum proteins described to have aberrant N-glycosylation during cancer progression, such as AFP (*Pinho and Reis, 2015*) and the protein disulfide isomerase PDIA3 (*LaMantia et al., 1991*).

Among the identified fucosylated proteins, we focused on PON1, an antioxidant enzyme, as its biological function matched the overrepresented BPs (*Figure 3C* and *Figure 3—figure supplement*

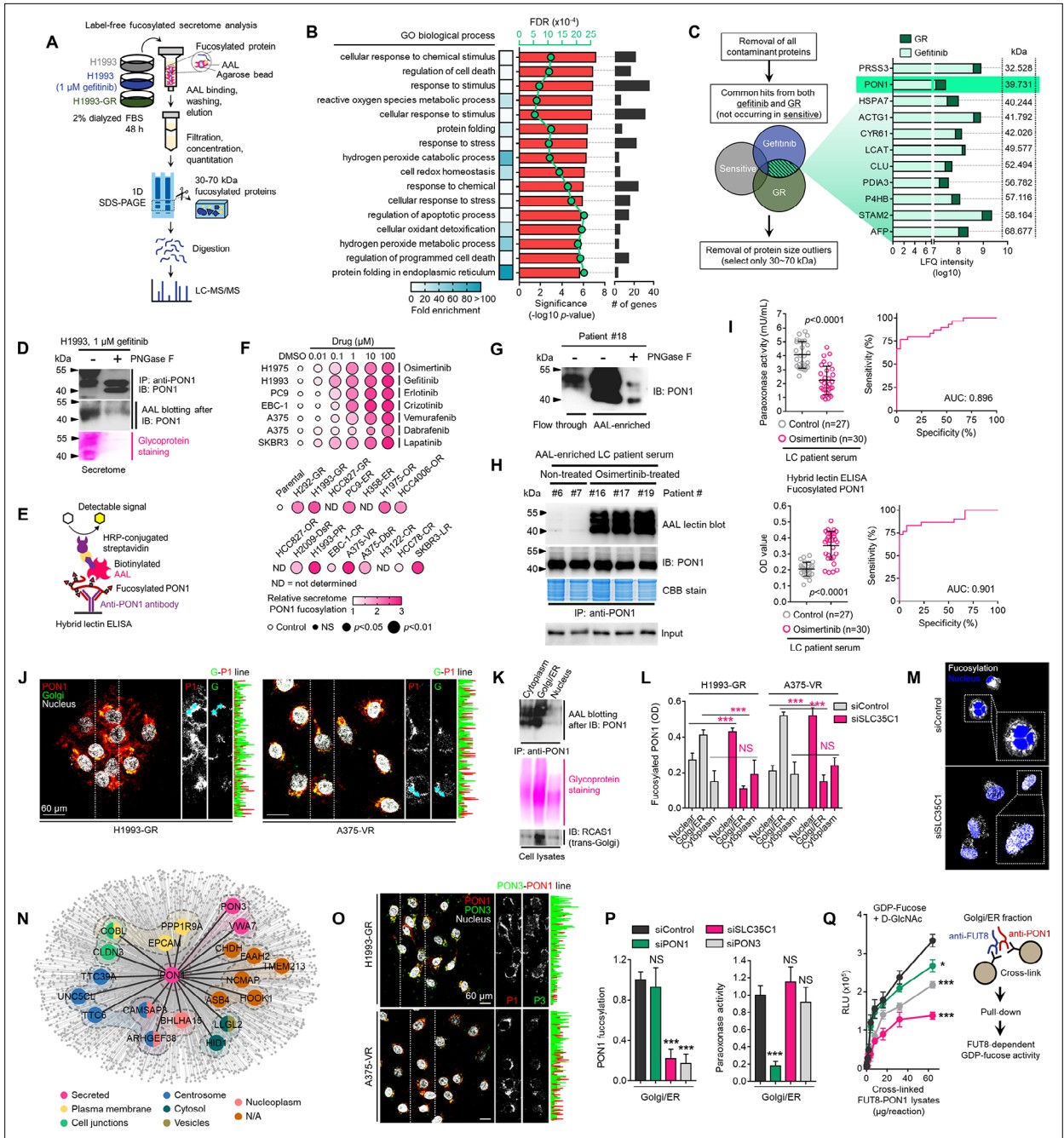

**Figure 3.** Identification of fucosylated PON1 as a critical component of therapy-induced cancer secretomes. (**A**) Schematic of the label-free secretome analysis workflow. (**B**) GO enrichment analysis for overrepresented BPs in cell-specific secretomes. Fold enrichment is shown as a heatmap. -log10 p values (red), false discovery rates (green), and the number of gene components per BP (gray) are displayed. Results were analyzed from two biological replicates. (**C**) Protein candidate screening approach and log10 LFQ intensities (relative protein abundances) of indicated overlapped proteins in secretomes of both gefitinib-treated H1993 cells and GR clone. Top 11 protein hits with MWs between 30 and 70 kDa are shown. Results were analyzed from two biological replicates. (**D**) Immunoblot and AAL blot analyses of PON1 expression and fucosylation status in PON1 immunoprecipitates from 1 µM gefitinib-treated H1993 secretomes. Secretomes were exogenously treated with or without 8 U PNGase F. Bottom panel shows glycoprotein stained SDS-PAGE gel of the same PON1 immunoprecipitates. Representative of two independent experiments. (**E**) Schematic of HLE for detecting PON1 fucosylation. (**F**) Dot plot visualization of PON1 fucosylation characterization by HLE analysis in secretomes from indicated cells and DR clones treated with or without indicated drug concentrations for 48 h. Color indicates fold change values relative to DMSO or parental (means ± SD of three biological replicates) while size indicates p values; Student's *t*-test. NS, not significant. (**G**) Immunoblot analysis of PON1 expression in indicated crude patient sera exogenously treated with or without 8 U PNGase F. Secretomes were either flow-through or enriched in AAL columns. Representative of two independent experiments. (**H**) AAL blot analysis of PON1 fucosylation and immunoblot analysis of PON1 expression in PON1 immunoprecipitates from

*Figure 3 continued on next page*

*Figure 3 continued*

indicated patient sera. Coomassie-stained SDS-PAGE gel of the same PON1 immunoprecipitates and input (10% of total protein) for PON1 immunoblot are also shown. Representative of two independent experiments. (**I**) HLE analysis of PON1 fucosylation in indicated crude patient sera. Values indicate the mean absorbance at 450 nm from three replicates. Representative of two independent experiments. Beside shows quantification of paraoxonase activity in the same crude patient sera. Values indicate mean fluorescence units at 412 nm from three replicates. Representative of two independent experiments. ROC curves for both PON1 fucosylation and paraoxonase activity are shown. For statistical analysis, the nonparametric Kruskal-Wallis test was used. (**J**) Representative confocal images of indicated DR clones stained for RCAS1 (golgi marker; green), PON1 (red), and DAPI (nuclei; white). The co-localization histogram plot of the indicated line is shown. Representative of two independent experiments. (**K**) AAL blot analysis of PON1 fucosylation in PON1 immunoprecipitates from indicated subcellular fractionated H1993-GR. Middle panel shows glycoprotein stained SDS-PAGE gel of subcellular fractionated cell lysates. Bottom panel shows immune blot analysis of RCAS1 in the same cell lysates. Representative of two independent experiments. (**L**) HLE analysis of PON1 fucosylation in indicated subcellular fractionated DR clone lysates upon SLC35C1 RNAi for 48 hr. Values indicate absorbance at 450 nm (means ± SD of three biological replicates). ***$p<0.001$, Student's *t*-test. NS, not significant. (**M**) Representative confocal images of H1993-GR upon SLC35C1 RNAi for 48 hr. GR clones were stained for SLC35C1 (white) and DAPI (nuclei; blue). (**N**) Genes co-expression network of PON1 queried in the CCLE. All nodes represent statistically significant co-expression with a gene. Top 20 PON1 co-expressing genes are highlighted. Colored nodes indicate cellular localization of protein-coding genes queried in The Human Protein Atlas. (**O**) Representative confocal images of indicated DR clones stained for PON1 (red), PON3 (green), and DAPI (nuclei; white). The co-localization histogram plot of the indicated line is shown. (**P**) HLE analysis of PON1 fucosylation and quantification of paraoxonase activity in Golgi/ER fractionated H1993-GR lysates upon SLC35C1, PON1, or PON3 RNAi for 48 hr. Values are relative to siControl (means ± SD of three biological replicates). ***$p<0.001$, Student's *t*-test. NS, not significant. (**Q**) GDP-Fuc activity analysis of FUT8 in cross-linked FUT8 and PON1 co-immunoprecipitates from Golgi/ER fractionated H1993-GR lysates. Values indicate luminescence units and are relative to control reaction (means ± SD of three biological replicates). *$p<0.05$, ***$p<0.001$, two-tailed Mann–Whitney *U* test.

The online version of this article includes the following source data and figure supplement(s) for figure 3:

**Source data 1.** Uncropped blots and gels (labeled and unlabeled) for *Figure 3D, G, H and K*.

**Source data 2.** Uncropped blots (labeled and unlabeled) for *Figure 3—figure supplement 2A*.

**Source data 3.** Uncropped blots (labeled and unlabeled) for *Figure 3—figure supplement 3D*.

**Figure supplement 1.** In-gel N-glycome analysis.

**Figure supplement 2.** PON1 N-glycosylation regulates PON1 secretion.

**Figure supplement 3.** Transcriptional regulation of PON3 directly impacts targeted therapy-induced core fucosylation of PON1 scaffolds in the golgi prior to secretion.

*1C*). We previously identified PON1 to be systemically fucosylated in sera of late-stage metastatic small cell LC (SCLC) patients in an integrated glycoproteomics screen (*Ahn et al., 2014*). In the gefitinib-treated H1993 cell secretome, we confirmed strong fucosylation of PON1, which appeared to have two isoforms: one with an apparent molecular mass of ~55 kDa while the other is ~45 kDa (*Figure 3D*). Intracellularly, PON1 has both nuclear and cytoplasmic isoforms where a~40 kDa cytoplasmic isoform is selectively enriched in LC patient tissues and cell lines (*Aldonza et al., 2017*). To quantitatively validate secretome PON1 fucosylation in drug-stressed cancer cells and DR clones, we employed PON1 fucosylation-specific hybrid lectin ELISA (HLE; *Figure 3E*). Despite different cell lineages, different oncogenic drivers, and different drugs, we found a widespread elevation of fucosylated secretome PON1 levels in multiple cancer cells upon targeted therapy (*Figure 3F*). Similarly, PON1 fucosylation is enriched in secretomes derived from DR clones (at least those that have detectable PON1 gene expression) (*Figure 3F*) and is strikingly elevated in LC patient sera upon osimertinib treatment (*Figure 3—figure supplement 2A* and *Figure 3G and H*). Immunoprecipitated PON1 from core fucosylation-enriched patient sera reveals that the ~55 kDa isoform of secreted PON1 is favorably N-glycosylated than the ~45 kDa isoform, both of which have aberrantly high levels in osimertinib-treated patient sera (*Figure 3G and H*).

Using receiver operating characteristic (ROC) curves, we investigated whether fucosylated PON1 can discriminate between non-treated and osimertinib-treated LC patient sera. PON1 fucosylation discriminated against the conditions with high sensitivity and specificity with the associated area under the curve (AUC) of 0.901, based on HLE measurements (*Figure 3I*, left). We previously reported that systemic serum PON1 is diminished in LC patients where fucosylated serum PON1 is increased (i.e. extensive disease). We hypothesize that this inverse relationship reflects an N-linked glycosylation-dependent control of PON1 activity. Supporting this idea, serum paraoxonase and arylesterase activities of PON1 were significantly differentiated between non-treated and osimertinib-treated LC patient sera (*Figure 3I*, right and *Figure 3—figure supplement 2B*). Also, both of these enzyme activities

significantly discriminated the treatment group with AUCs ranging from 0.76 to 0.89. In LC patient tissues, PON1 fucosylation is associated with relapse and discriminated it from non-relapsed LC with an AUC of 0.77 (*Figure 3—figure supplement 2C*). Next, we characterized intracellular PON1 fucosylation in DR clones. PON1 is primarily localized in the Golgi and has active fucosylation in the Golgi/ ER fractions of GR and vemurafenib-resistant (VR) clones (*Figure 3J and K*). SLC35C1 RNAi significantly reduced Golgi-enriched PON1 fucosylation and promiscuously induced overall fucosylation in the nucleus of both DR clones (*Figure 3L and M*), indicating a functional defect in the transport of GDP-Fuc along the secretory pathway.

To identify direct regulators of PON1 fucosylation, we examined CCLE-annotated PON1 protein interactors based on co-expressing genes. We first clustered hits based on Spearman's correlation and identified the cellular localization of each protein (*Figure 3N*). Among the top 20 proteins, only paraoxonase 3 (PON3) and VWA7 showed co-localization with PON1 in the secretory pathway. We were intrigued by PON3, also a serum paraoxonase known to both preferentially interact and share numerous conserved PTM (i.e. N-glycosylation) sites with PON1 (*Harel et al., 2004*). In both GR and VR clones, there is an active co-localization between PON1 and PON3 (*Figure 3O*). While PON1 expression did not discriminate non-relapsed and relapsed BC patient tissues, high PON3 expression correlated well with relapse (*Figure 3—figure supplement 2D*). In addition, PON3 expression is increased in various drug-stressed cells and DR clones (*Figure 3—figure supplement 2E*). In Golgi/ ER of H1993-GR, both PON3 and SLC35C1 RNAi, but not PON1 RNAi, inhibited PON1 fucosylation (*Figure 3P*), demonstrating that PON3 directs PON1 fucosylation prior to secretion. Similarly, PON1 RNAi was not sufficient to inhibit secretome PON1 fucosylation (*Figure 3—figure supplement 2F*). Regardless, the same transient PON1 RNAi was sufficient to inhibit PON1 fucosylation in whole cell and cytoplasmic intracellular fractions of H1993-GR (*Figure 3—figure supplement 2F*), while stable PON1 RNAi via shRNA completely obliterated PON1 fucosylation to undetectable levels in both cellular fractions and secretome of a control PON1-expressing cell line (*Figure 3—figure supplement 2F*).

Moreover, only PON1 RNAi, not PON3 or SLC35C1 RNAi, impeded Golgi/ER-specific paraoxonase activity (*Figure 3P*), reflecting known differences between the two PONs in hydrolyzing paraoxon (*Draganov et al., 2005*). In a cross-linking GDP-Fuc activity assay, we showed that PON3 or SLC35C1 RNAi could ablate FUT8-directed transfer of fucose moiety from GDP-Fuc to N-glycan GlcNAc residue of PON1 (*Figure 3Q*), implying direct functional regulation of PON1 fucosylation by PON3. Confirming the depletion of secretome PON1 fucosylation by a glycosidase, we showed that in the secretome, PON1 is de-N-glycosylated upon exogenous addition of PNGase F onto cultures of sensitive cells and DR clones but without marked changes in GDP-Fuc activity on PON1 in Golgi/ER or PON1 secretion in sensitive cells, DR clones, and cells engineered to overexpress PON1 (*Figure 3—figure supplement 2H–J*).

As a preliminary attempt to functionally examine how PON1 core fucosylation is directly regulated by an interacting co-factor, we employed a technique in which immunoprecipitation of cross-linked PON1 and known PON1 activity regulators—PON3, von Willebrand factor (VWF), and myeloperoxidase (MPO)—from cell lysates or secretomes was performed in a multiplexed fashion on barcoded fluorescent beads (see Methods); and then further assayed for fucosylation and enzyme activities (*Figure 3— figure supplement 3A*). The degree of PON1 cross-linking with each of various other proteins was quantified using a PON1 antibody. Hinted by its strong co-expression with PON1 and functional implication on PON1 core fucosylation, EGFR-TKI-induced or EGFR-TKI resistance-associated abundance of PON1-interacting PON3 in whole cell lysates, golgi fractions, and secretomes are all linked with increased PON1 core fucosylation and PON1-bound core N-glycan enrichment (*Figure 3—figure supplement 3B and C*). Similar signatures were observed with PON1-interacting VWF (*Figure 3— figure supplement 3B and C*), at least in golgi fractions and secretomes of EGFR-TKI-treated cells and DR clones. While the plasma glycoprotein VWF has been associated with serum PON1 enzyme activity in various disease settings (*Chen, 2014*), its direct regulatory effects on PON1 is yet to be explored. In contrast, MPO is associated with decreased PON1 interaction, decreased core fucosylation, and lesser PON1-bound N-glycans across EGFR-TKI-treated cells and DR clones (*Figure 3— figure supplement 3B and C*). Although both PON1 and MPO are HDL-associated enzymes known to physically form ternary complexes with HDL (*Huang et al., 2013*), the results might reflect the contradicting oxidizing function of MPO as opposed to the antioxidant nature of PON1 (*Huang et al., 2013*;

*Aggarwal et al., 2021*). In addition, transcriptional control of both PON1 and PON3 by SREBP2, a master transcription factor (TF) for PON promoter activity, tightly regulate PON1-PON3 interaction and PON3-mediated PON1 core fucosylation in the golgi and secretomes of EGFR-TKI-treated cells and DR clones (*Figure 3—figure supplement 3D and E*). These data suggest that PON3's regulatory effect on PON1 core fucosylation is dependent on its direct binding with PON1 and its effects on PON1 enzyme activity, and that direct control of PON3 is sufficient to regulate PON1 core fucosylation. Taken together, these results suggest that core fucosylated PON1 is a major component of the constitutive N-glycome of the cancer TIS and a signature of targeted therapy resistance.

## Core fucosylation enhances PON1 stability and prompts PON1 for secretion in therapy-resistant cancer cells

Given our prior knowledge on how systemic serum PON1 activity is diminished in LC patients and mouse model profiled with high serum PON1 fucosylation (*Ahn et al., 2014*; *Aldonza et al., 2017*), we hypothesized that therapy-induced protein glycosylation rewires the maturation steps of PON1 in the secretory pathway (*Figure 4A*). PON1 has 23 predicted N-glycosylation sites with four Asn-X-Ser/Thr sequons–consensus amino acid sequences that determine N-glycosylation efficiency–all scored above the 'high potential' threshold (*Figure 4B*). PON1 is predicted to be mostly folded and has a negative net electrical charge (–16 at pH 7). All four sequons of PON1 (N227, N253, N270, N324) and their immediate vicinity have either neutral (0) or negative (ranging from –0.4—0.2) net charge (*Figure 4B*). Among the four sequons, N324(GT) and N270(IS) are well conserved throughout species while N253(WT) is uniquely conserved in mammals (*Figure 4C*). Both N253(WT) and N324(GT) sequons are located in the outer region of PON1's β-propeller structure while the other sequons are found in the innermost tunnel structure near the calcium-binding sites (*Figure 4D*). Whether or not these indicate a preference for aberrant glycosylation remains an open question. Regardless, the net charge, polarity, and X amino acid (in Asn-X-Ser/Thr) of sequons and their vicinity can generate preferable environments for aberrant protein N-glycosylation (*Manwar Hussain et al., 2018*; *Shakin-Eshleman et al., 1996*).

To structurally map the bound N-glycans on PON1, we analyzed our previous tandem mass spectrometry (MS/MS) data (*Ahn et al., 2014*). We determined six aberrantly fucosylated glycans released from immunoprecipitated PON1, where $GlcNAc_2Man_3 + HexNAc_2Fuc_1$ putative glycan structures are commonly present (*Figure 4E*). Two of the most abundant glycans (peaks 1 and 2) were identified to have high FUT8 substrate specificity, while the rest (peaks 3–6) have either low specificity or not yet identified (*Figure 4E*, bottom; *García-García et al., 2020*; *Tseng et al., 2017*). To probe PON1 fucosylation in a site-specific manner, we introduced single-point mutation in two PON1 sequons [N253(WT) and N324(GT)]–predicted to display loss of N-glycosylation along with protein destabilization upon Asp→Gly mutation–and transfected the wild-type (WT) or mutant constructs into sensitive cells, DR clones, and PON1-edited cells (*Figure 4F*). Both PON1 mutants selectively reduced PON1 core fucosylation and prevented efficient GDP-Fuc transfer (*Figure 4G and H*). Note that N253G displayed more robust effects than N324G. These mutants only had subtle effects on overall secretome N-glycosylation and did not alter gene expression of PON1, PON3, fucose salvage factors (FUK, GMD), GDP-fucose transporter (SLC35C1), and fucosylatransferase (FUT8) (*Figure 4—figure supplement 1A and B*). WT or PON1 mutants did not have significant effects on the response of sensitive cells to EGFR-TKIs while both N253G and N324G mutants, not FL, sensitized both GR and ER clones to EGFR inhibition (*Figure 4—figure supplement 1C*), suggesting that PON1 fucosylation is a resistance selected mechanism.

To validate the predicted effects of N253G and N324G mutations on PON1 stability, we assayed PON1 folding and synthesis upon protein cleavage by trypsin or de novo protein synthesis inhibition by cycloheximide (CHX) treatment. Immunoblotting of whole GR clone lysates revealed that N253G remarkably promoted PON1 misfolding by completely sensitizing PON1 to cleavage by trypsin. N324G induced a noticeable PON1 cleavage only at a higher trypsin concentration (*Figure 4I*). In the Golgi/ER of GR clone, similar effects were also afforded on PON1 when tested using ELISA and on protein glycosylation after PON1 immunoprecipitation (*Figure 4J* and *Figure 4—figure supplement 1D*). In Golgi/ER of A549 cells, where there is basal PON1 expression, N253G did not alter PON1's sensitivity to trypsin. Conversely, the same mutation rendered PON1 from PON1-overexpressing cells sensitized to trypsin (*Figure 4—figure supplement 1E*). Furthermore, EGFR-TKI resistance or PON1

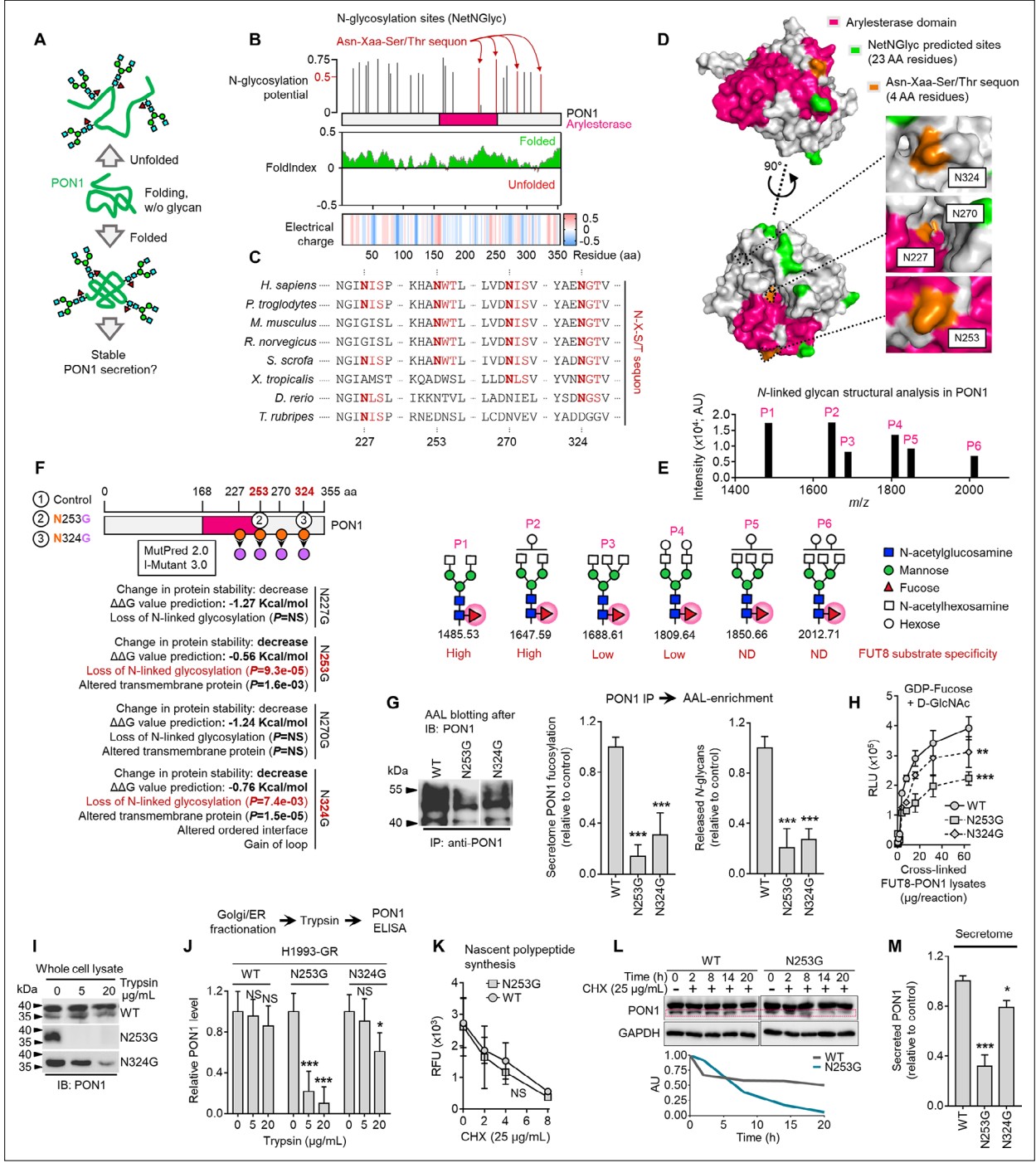

**Figure 4.** Core fucosylation impacts PON1 folding and stability prior to secretion in therapy-resistant cancer cells. (**A**) Hypothetical model of N-glycosylation control of PON1 stability. (**B**) PON1-WT N-glycosylation site prediction using NetNGlyc 1.0, folding prediction using FoldIndex, and charge prediction using EMBOSS.>0.5 threshold score means significant glycosylation potential. Unfolded regions are depicted in red, folded regions in green. Positive charged is marked in red shades, negative charge in blue, and neutral charge in white. (**C**) Conservation of indicated PON1 sequons throughout species. (**D**) Closed conformation surface structure of PON1 (PDB ID: 1V04) highlighting arylesterase domain and predicted N-glycosylation sites and sequons. The 3D surface view was visualized using PyMOL. (**E**) N-glycan structural analysis of PON1 from our previous tandem MS/MS dataset. The m/z 1647.62 [(M+Na)+corresponding to GlcNAc2Man3+HexNAc2Hex1Fuc1] is the base peak (not visualized). Putative structure visualization of indicated monosaccharides and FUT8 substrate specificity were based on CID data and known glycobiology. (**F**) Prediction of PON1 stability, structural and functional properties upon indicated in silico N→G substitution at specific sequons using MutPred 2.0 and I-Mutant 3.0. Two N→G substituted sequons (N253G and N324G) with statistically significant potential of loss of N-glycosylation were chosen for validation experiments. (**G**) AAL blot analysis of PON1 immunoprecipitates from H1993-GR upon transfection with indicated PON1-WT, PON1-N253G, or PON1-N324G constructs for

*Figure 4 continued on next page*

*Figure 4 continued*

36 hr. Representative of two independent experiments. Beside shows HLE analysis of secretome PON1 fucosylation and N-glycan release assay in AAL-enriched PON1 immunoprecipitates from H1993-GR upon similar transfection. Values are relative to PON1-WT (means ± SD of three biological replicates). ***p<0.001, Student's *t*-test. (H) GDP-Fuc activity analysis of FUT8 in cross-linked FUT8 and PON1 co-immunoprecipitates from H1993-GR upon transfection with constructs as in G. Values indicate luminescence units and are relative to control reaction (means ± SD of three biological replicates). **p<0.01, ***p<0.001, two-tailed Mann–Whitney *U* test. (I) Immunoblot analysis of PON1 expression in H1993-GR upon transfection with constructs as in G. Lysates were exogenously treated with or without indicated trypsin concentration. Representative of two independent experiments. (J) ELISA analysis of PON1 expression in H1993-GR upon transfection with constructs as in G. Golgi/ER fractionated cell lysates were exogenously treated with or without indicated trypsin concentrations. Values are relative to no treatment (means ± SD of three biological replicates). *p<0.05, **p<0.01, ***p<0.001, Student's *t*-test. NS, not significant. (K) EZClick labeling analysis of polypeptide synthesis in H1993-GR upon transfection with constructs as in G and treated with or without 25 µg/mL CHX concentrations for indicated times. Values indicate raw fluorescence units (means ± SD of two biological replicates). For statistical analysis, Student's *t*-test was used. NS, not significant. (L) Immunoblot analysis of PON1 expression in H1993-GR upon transfection with constructs as in G and treated with or without 25 µg/mL CHX for indicated times. GAPDH was used as a loading control. Blot intensity quantification of the lower PON1 kDa isoform is shown. Representative of two independent experiments. (M) ELISA analysis of secretome PON1 expression in H1993-GR upon transfection with constructs as in G. Values are relative to WT (means ± SD of three biological replicates). *p<0.05, ***p<0.001, Student's *t*-test.

The online version of this article includes the following source data and figure supplement(s) for figure 4:

**Source data 1.** Uncropped blots and gels (labeled and unlabeled) for *Figure 4G, I and L*.

**Source data 2.** Uncropped gels (labeled and unlabeled) for *Figure 4—figure supplement 1B and D*, 1 G, and 1 H.

**Figure supplement 1.** Core fucosylation directs PON1 secretory process in DR clones and PON1-overexpressing cells.

**Figure supplement 2.** PON1 secretome fucosylation promotes TIS-induced resistance rebound in regressing cell admixtures.

**Figure supplement 3.** PON1-N253 sequon is important for drug resistance-promoting effect of targeted therapy-induced secretome PON1 fucosylation in vivo.

overexpression delayed the degradation of nascent polypeptides upon CHX treatment (*Figure 4—figure supplement 1F*). In the GR clone, N253G had no significant effect on overall protein synthesis (*Figure 4K*). In addition, EGFR-TKI resistance delayed the degradation of total fucosylated proteins and Golgi-specific PON1-immunoprecipitated glycoproteins (*Figure 4—figure supplement 1G and H*). In the GR clone, N253G accelerated the degradation of the lower kDa isoform of PON1, presumably its fucosylated form (*Figure 4L*). More importantly, N253G significantly ablated PON1 secretion while N324G displayed a modest effect (*Figure 4M*). Unexpectedly, N253G inhibited the intracellular arylesterase, but not paraoxonase, activity in GR clone (*Figure 4—figure supplement 1I*). This is consistent with our hypothesis that N-glycosylation of PON1 governs its enzyme activity. Taken together, our data suggest that core fucosylation promotes PON1 stability prior to secretion in DR clones and PON1-overexpressing cells. This offers an answer to our long-standing question of how fucosylation affords a more stable, degradation-resistant PON1 state in the secretion, which seems to involve a rewired enzyme activity.

To investigate the functional consequences of PON1-specific core fucosylation inhibition on TIS-driven therapy resistance, we performed similar PON3 RNAi and PON1 site-directed mutagenesis experiments in sensitive cells followed by 2D cell admixture assays using both sensitive cells and DR clones; and in mouse xenografts derived from our H1993-GR model subjected to EGFR-TKI therapy. Inhibition of PON1 core fucosylation via PON3 silencing or PON1-N253G mutation in sensitive cells significantly prevented the population expansion of DR clones in regressed cell admixtures at day 5 (*Figure 4—figure supplement 2*). More importantly, PON1-N253G markedly sensitized EGFR-TKI-resistant tumors in mice without observable weight loss (*Figure 4—figure supplement 3A and B*), and led to systemic decrease of both EGFR-TKI-induced serum PON1 core fucosylation and PON1-bound N-glycans in xenografted mice (*Figure 4—figure supplement 3C*). Collectively, these consistently support our hypothesis that PON1 core fucosylation is a critical and functional component of the cancer TIS that promotes targeted therapy resistance.

## Blockade of secretome core fucosylation confines therapy resistance via UPR effectors and a pro-inflammatory niche

To generally address the mechanism by which TIS de-N-glycosylation prevents the rebound of DR clones in a regressing tumor model, we preliminarily employed a focused TF screen that mapped changes in 45 TF-driven intracellular signaling pathways using a dual-luciferase activity reporter array

following a direct admixture assay. In a retrieved fraction of GR clone upon TIS de-N-glycosylation in 2D admixture, ER stress (ATF6), amino acid response (AAR element; ATF2, ATF3, ATF4), androgen receptor (AR) pathways, and E2F transcription were distinctively up-regulated, while stem cell factors (SOX2, NANOG, OCT4), interferon-stimulated response (ISR element; STAT1, STAT2), STAT3, and hypoxia (HIF) signaling were selectively repressed (*Figure 5A*). We validated this expression signature in 3D admixtures of DR clone (H1993-GR or PC9-ER) and sensitive cells (*Figure 5B* and *Figure 5—figure supplement 1A*). In these admixtures, PON1-N253G transfection in sensitive cells pheno-copied the effects of TIS de-N-glycosylation on intracellular signaling, senescence, regressing TME cues, kinase phospho-proteome, and growth of GR and ER clones (*Figure 5C* and *Figure 5—figure supplement 1B–G*). These data point to a cascade of ER stress and UPR-regulated translational reprogramming events as mediators in blocking the growth of DR clones upon TIS de-N-glycosylation or fucosylated PON1 inhibition.

To probe these processes, we focused on ATF6, an ER-localized UPR-specific stress sensor (*Boyce and Yuan, 2006*) and a top hit from our TF screen. In a CM co-culture, both TIS de-N-glycosylation and fucosylated PON1 inhibition actively induced Golgi/ER-localized ATF6 with enriched co-localization with fucosylated scaffold residues in DR clones (*Figure 5D* and *Figure 5—figure supplement 1E and H*), which probably implicates ER stress-induced translocation/sorting of ATF6 from ER to Golgi. Next, we asked whether these stress-induced responses are orchestrated with oxidative stress given the phenotypic response of cell admixtures to inhibition of TIS-specific PON1 fucosylation via PON1-N253G mutation. Indeed, de-N-glycosylating TIS and inhibiting TIS-specific fucosylated PON1 in regressing cell admixtures markedly stimulated the production of reactive oxygen and nitrogen species (ROS/RNS), a hallmark of redox imbalance (*Figure 5E* and *Figure 5—figure supplement 1I*). In DR clones co-cultured in modified TIS with scarce PON1 fucosylation, silencing ATF6 prohibited the generation of intracellular ROS/RNS (*Figure 5F* and *Figure 5—figure supplement 1J*). In gefitinib-treated 3D cell admixtures, PON1-N253G-bearing sensitive cells restrained the growth of GR clone while ATF6 RNAi reverted this effect (*Figure 5G and H*). PON1-N253G promoted a pro-inflammatory environment in the same cell admixtures with increased levels of IL-6, TNF-α, and GM-CSF cytokines in the secretome while ATF6 RNAi antagonized this induced cytokine signature (*Figure 5I*). Meanwhile, de-N-glycosylation of the TIS or inhibition of TIS-specific fucosylated PON1 markedly induced ATF3 and ATF4—both of which contain an amino acid response element (AARE) in their promoters (*Ron and Walter, 2007*) which was one of the hits identified from our TF screen (*Figure 5A*)—in DR clones but this effect is negligible in the golgi (*Figure 5—figure supplement 1K*). Regardless, silencing ATF3 or ATF4 decreased intracellular ROS/RNS levels in DR clones co-cultured in modified TIS where fucosylated PON1 is inhibited (*Figure 5—figure supplement 1L*). These localization-specific effects of ATF6, ATF3, ATF4 can likely be explained by differences in their retention and translocation in ER-golgi (*Ron and Walter, 2007*).

To assess in more detail how perturbations in PON1 fucosylation influence the growth of DR clones, we established cell models to differentially modify secretome PON1 fucosylation. When overexpressed at a high degree (>700 fold) in cells without detectable PON1 (H460 and H1993), we observed an active PON1 fucosylation in the secretion. Stably knocking-out PON1 in wild-type PON1-expressing cells (A549) did not alter PON1 fucosylation intracellularly or in the secretion while stably knocking-out SLC35C1 in PON1-overexpressing cells led to suppression. Additionally, transfection of PON1-N253G construct or exogenous PNGase F treatment in PON1-overexpressing cells mitigated secretome PON1 fucosylation (summary in *Figure 5J*). Accordingly, modified cells with de-N-glycosylated secretome PON1 have increased ROS/RNS and a more pro-inflammatory secretome (*Figure 5K and L*). More importantly, PON1 fucosylation-enriched TIS amplified the growth of GR and ER clones while cells with de-N-glycosylated PON1 prevented this growth coupled with increased intracellular caspase activities (*Figure 5M and N*). Furthermore, UPR target genes were consistently up-regulated in DR clones co-cultured with TIS or PON1-overexpressing cell secretomes with de-N-glycosylated PON1 (*Figure 5—figure supplement 1M*). In the same DR clones, restricted growth is associated with nuclear translocation of XBP1 (*Figure 5—figure supplement 1N*), indicating ATF6-induced transcription factor activation.

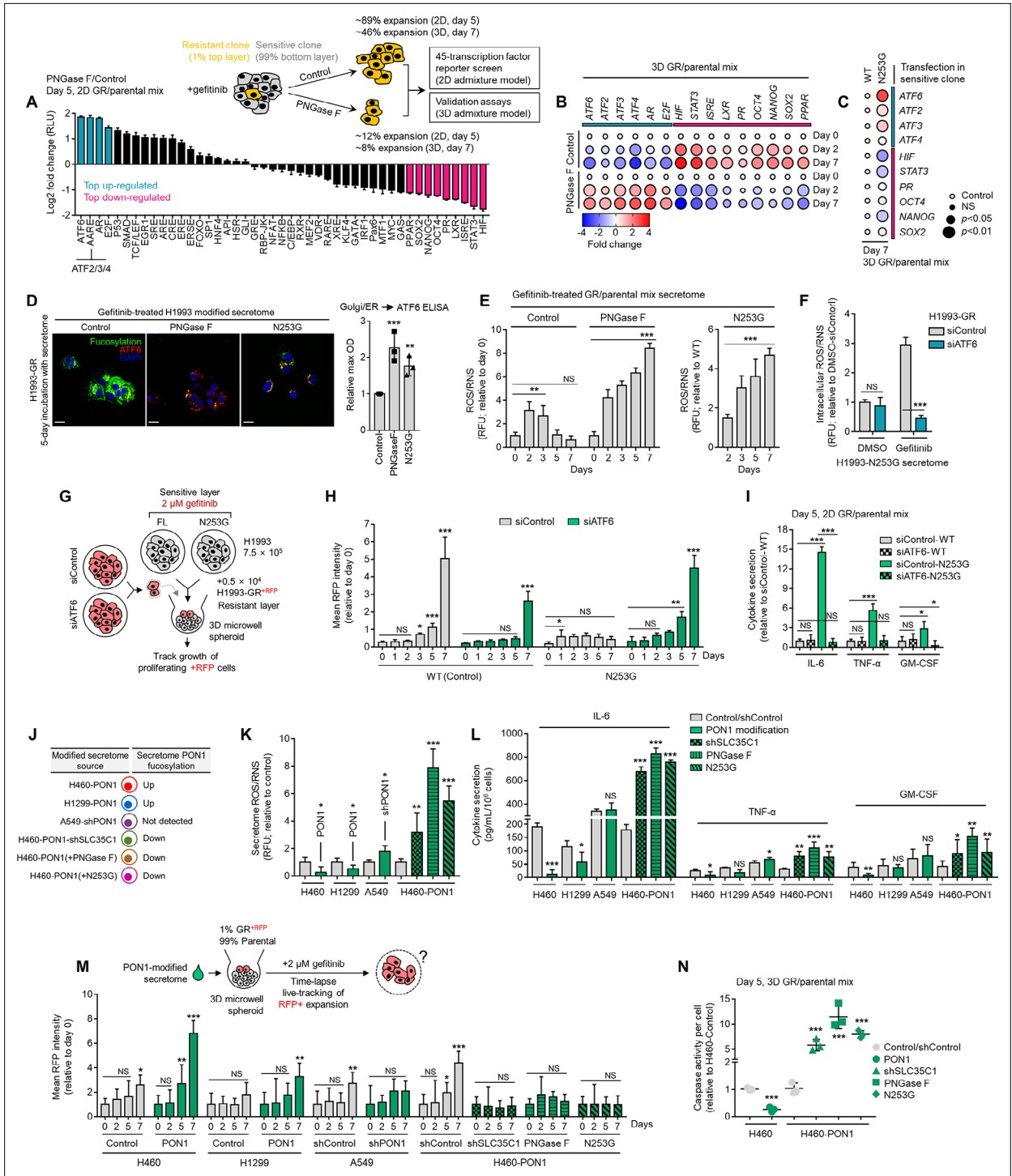

**Figure 5.** Secretome PON1 fucosylation promotes resistance via neutralization of inflammatory response and ROS. (**A**) Cignal 45-pathway array of reporter transcriptional activities in indicated cell admixtures treated with 1 µM gefitinib and incubated with or without 10 µg/mL recombinant PNGase F for 5 days. The reporter array is based on a dual-luciferase system that measures the activity of indicated 45 transcription factors. Log2 values were normalized by control condition and represented as fold changes in luciferase units (means ± SD of two biological replicates). Highlighted top up-/down-regulated hits are all statistically significant (p<0.001, Dunnett's test). (**B**) Dot plot visualization of indicated gene expression by qPCR analysis in 3D cell admixtures with the same conditions as in A, except treated with 2 µM gefitinib for 2 or 5 days. Color indicates log-transformed fold change values relative to day 0 control and normalized to GAPDH levels (means ± SD of three biological replicates) while size indicates p values; Student's t-test. NS, not significant. (**C**) Dot plot visualization of indicated gene expression by qPCR analysis in 3D cell admixtures with same conditions as in B upon transfection with PON1-WT or PON1-N253G construct for 36 hr. Color indicates log-transformed fold change values relative to WT control

*Figure 5 continued on next page*

*Figure 5 continued*

and normalized to GAPDH levels (means ± SD of three biological replicates) while size indicates p values; Student's t-test. NS, not significant. (**D**) Representative confocal images of H1993-GR grown for 5 days in indicated secretomes from 1 μM gefitinib-treated H1993 cells exogenously treated with total 8 U PNGase F or transfected with PON1-WT or PON1-N253G construct for 36 hr. GR clones were stained for fluorescein-conjugated AAL (core fucosylation; green), ATF6 (red), and DAPI (nuclei; blue). Beside shows ELISA analysis of ATF6 expression in Golgi/ER fractionated H1993-GR with the same conditions. Values are relative to WT (means ± SD of three biological replicates). \*\*p<0.01, \*\*\*p<0.001, Student's *t*-test. (**E**) ROS/RNS detection in secretomes from 1 μM gefitinib-treated cell admixtures as in A or C. Values are relative to day 0 or WT (means ± SD of two biological replicates). \*\*p<0.01, \*\*\*p<0.001, Student's *t*-test. NS, not significant. (**F**) Intracellular ROS/RNS detection in H1993-GR upon ATF6 RNAi for 48 hr and grown in secretomes from PON1-N253G-transfected H1993 cells treated with or without 1 μM gefitinib for 72 hr. Values are relative to DMSO siControl (means ± SD of two biological replicates). \*\*\*p<0.001, Student's *t*-test. NS, not significant. (**G**) Schematic of sequentially layered admixture. (**H**) Tracking of RFP-tagged H1993-GR upon ATF6 RNAi in 3D cell admixtures as in G. Sensitive cells were transfected with PON1-WT or PON1-N253G for 36 hr. Values are relative to day 0 (means ± SD of two biological replicates). \*p<0.05, \*\*p<0.01, \*\*\*p<0.001, two-tailed Mann–Whitney *U* test. NS, not significant. (**I**) Sandwich ELISA analysis of indicated cytokines in secretomes from cell admixtures prepared as in D, except in 2D. Values are relative to siControl WT (means ± SD of two biological replicates). \*P<0.05, \*\*\*P<0.001, Student's *t*-test. NS, not significant. (**J**) Modified secretomes from PON1-edited cells with varying PON1 fucosylation. (**K**) ROS/RNS detection in secretomes described as in J. Values are relative to control (means ± SD of three biological replicates). \*p<0.05, \*\*p<0.01, \*\*\*p<0.001, Student's *t*-test. (**L**) Sandwich ELISA analysis pf indicated cytokines in secretomes described as in J. Values are relative to control/shControl (means ± SD of two biological replicates). \*p<0.05, \*\*p<0.01, \*\*\*p<0.001, Student's *t*-test. NS, not significant. (**M**) Tracking of RFP-tagged H1993-GR in 3D cell admixtures described in the schematic. Admixtures were grown in secretomes described as in J. Values are relative to day 0 (means ± SD of two biological replicates). \*p<0.05, \*\*p<0.01, \*\*\*p<0.001, two-tailed Mann–Whitney *U* test. NS, not significant. (**N**) Caspase activity analysis in 3D cell admixtures as in M and grown in secretomes described as in J for 5 days. Values are relative to control/shControl (means ± SD of two biological replicates). \*\*\*<0.001, Student's *t*-test.

The online version of this article includes the following source data and figure supplement(s) for figure 5:

**Source data 1.** Uncropped blots (labeled and unlabeled) for *Figure 5—figure supplement 1D*.

**Figure supplement 1.** Inhibition of TIS-specific PON1 fucosylation prevents resistance rebound via suppression of kinase phospho-proteome and induction of UPR target genes.

**Figure supplement 2.** Validation of target gene control upon RNAi or stable overexpression in cells.

## Transcriptomics reveals resistance-associated genes mediated by secretome PON1 core fucosylation

To broadly identify resistance-relevant genes that modulate responses to changes in secretome fucosylation, we performed a transcriptome-wide analysis of gene expression in DR clones at 48 hr after co-culture with altered secretome conditions in vitro (*Figure 6A*). In contrast to our TF-signaling screen performed on direct cell admixtures, this assay employs a CM co-culture set-up that is considerably more amenable to global profiling of genes driving the response to reactive secretomes. The similarity in gene expression profiles was observed in replicate samples and among conditions that represent control (H1993 and H460), positive fucosylation regulation (gefitinib-treated H1993 and H460-PON1), or negative PON1 fucosylation regulation (PNGase-treated and PON1-N253G-transfected H460-PON1; *Figure 6—figure supplement 1A*). Secretome de-N-glycosylation and PON1 fucosylation inhibition led to 433 altered gene expression, and pathway analysis revealed that overlapped genes were enriched for negative regulators of the cell cycle and regulators of transcription and metabolism (*Figure 6B*). The same conditions generated a more down-regulated DR clone transcriptome associated with the regulation of receptor signaling pathways, cell communication, and cell proliferation, among others (*Figure 6B*). Enriched secretome fucosylation led to fewer differentially expressed genes that are mostly involved in cellular metabolic reactions (*Figure 6—figure supplement 1B*). Only one overlapping altered gene was detected between the two conditions promoting secretome fucosylation (gefitinib treatment and PON1 overexpression; *Figure 6—figure supplement 1C*). To identify molecular drivers of the DR clone's response to the suppression of secretome PON1 fucosylation, we integrated the data of differentially expressed genes in both PNGase F treatment and PON1-N253G transfection conditions. Secretome de-N-glycosylation led to 135 down- and 65 up-regulated genes, and secretome PON1 fucosylation inhibition resulted in 150 down- and 83 up-regulated genes, all with statistically significant p values (*Figure 6C*). 21 genes were up-regulated while 90 genes were down-regulated in both conditions (*Figure 6D*). This analysis highlighted C19orf25, RPS27L, CLDN2, PAQR3, and SOX4 as putative blockers; while THBS1, F3, TAGLN, ANKRD1, and DKK1 as positive regulators of secretome PON1 fucosylation-mediated DR clone outgrowth (*Figure 6E*), all of which were validated in a separate gene expression analysis (*Figure 6—figure supplement 2A*).

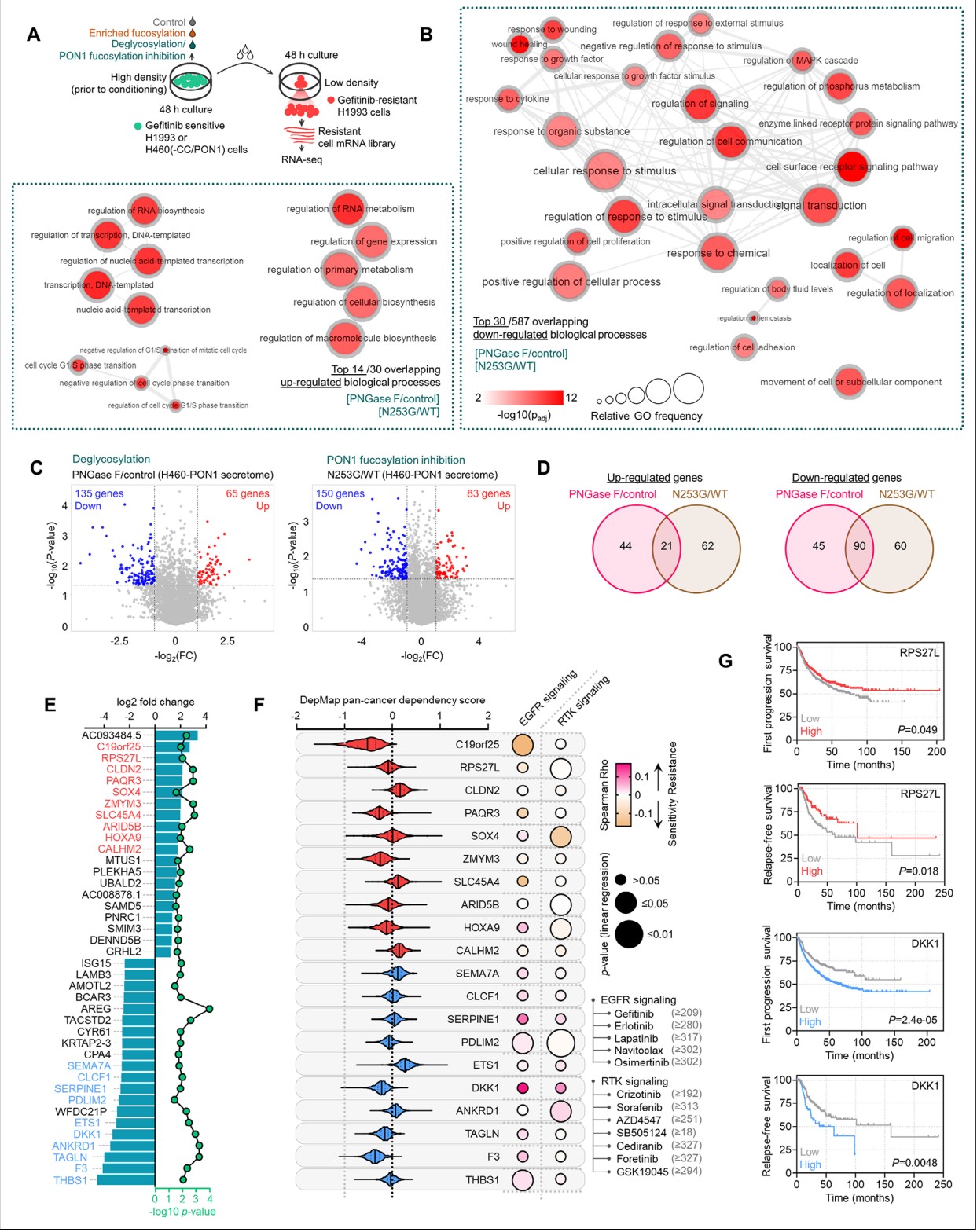

**Figure 6.** Transcriptome-wide analysis reveals modulator genes associated with secretome PON1 fucosylation-induced therapy resistance. (**A**) Schematic of co-culture conditions and preparation of transcript library from H1993-GR for RNA-seq. (**B**) GO analysis of gene expression changes in H1993-GR grown in indicated conditions showing enriched GO terms. The size of the circle indicates the frequency of the GO term in the underlying GOA database while color indicates adjusted -log10 p-value. Highly similar GO terms are linked by edges in the graph, where the line width indicates the degree of similarity. (**C**) Volcano plots showing differentially expressed genes deregulated by indicated conditions. Significantly up-regulated genes

*Figure 6 continued on next page*

*Figure 6 continued*

are in red, while down-regulated genes in blue. (**D**) Venn diagram indicating overlap of up-regulated or down-regulated genes in indicated conditions. (**E**) Log2 fold changes and -log10 p values of indicated top 20 overlapped up-regulated or down-regulated genes in indicated conditions as in D. Data are means. p Values were calculated using a two-tailed Mann–Whitney U test. (**F**) Violin plots depicting dependency scores of indicated top differentially expressed genes from two conditions as in D. Scores reflect data from 23 different cancer lineages. Central lines indicate the median. Data was obtained from the DepMap RNAi screen. Beside shows plot visualization of the correlation between indicated pan-cancer gene dependency and drug response screened in GDSC. Size of circle refers to linear regression p-value while color corresponds to Spearman's rank coefficients. (**G**) Kaplan-Meier plots of FP or RFS in multiple lung cancer patient cohorts. Patient survival data were stratified by indicated gene expression (low or high) in their primary tumors based on the microarray (FP) or RNA-seq (RFS) data. p Values were calculated using a log-rank test.

The online version of this article includes the following figure supplement(s) for figure 6:

**Figure supplement 1.** RNA-seq of DR clones cultured in PON1 fucosylation-modified secretomes.

**Figure supplement 2.** Expression validation and pharmacogenomics data on identified PON1 fucosylation-associated resistance modulator genes.

**Figure supplement 3.** PON1 fucosylation-associated resistance modulator genes control the expansion of DR clones in response to fucosylated TIS.

To elucidate the implication of these genes in targeted therapy resistance, we analyzed genome-scale loss-of-function screening data from the Cancer DepMap project. We observed variable pan-cancer dependency signatures among the top 10 up- and down-regulated hits from our initial screen and found that only C19orf25 is denoted as 'commonly essential' in a pan-cancer ranking scheme. Regardless, many of these genes display high dependency scores in a fraction of cancer cell lines (*Figure 6F*). We then examined whether these dependency profiles correlate with drug sensitivity. Indeed, many of these gene dependencies strongly correlate with either a drug-sensitive or a drug-resistant state to inhibitors of EGFR or RTK signaling, albeit varied *p* values mainly due to different cancer lineages screened. Intriguingly, top overlapping up-regulated genes upon secretome de-N-glycosylation, and PON1 fucosylation inhibition mostly correlates with a drug-sensitive state, while the down-regulated genes are more associated with resistance to EGFR-TKIs or RTK inhibitors (*Figure 6F*). Validating these results, their pan-cancer expression profiles also correlated with broadly similar drug sensitivity signatures (*Figure 6—figure supplement 2B*). In large LC patient cohorts, high expression of two up-regulated gene hits, RPS27L and C19orf25, are correlated with increased first progression survival or RFS, while high expression of two down-regulated gene hits, DKK1 and THBS1, are associated with poor survival outcomes after therapy (*Figure 6G* and *Figure 6—figure supplement 2C*). We next investigated whether these hits are functionally implicated in the response of DR clones to TIS fucosylation by employing a CM co-culture assay where RFP-labeled, low-density DR clones were subjected to RNAi or lentiviral target gene overexpression followed by culture in modified TIS and then tracked overtime (*Figure 6—figure supplement 3A*). Predominantly, RPS27L RNAi and DKK1 overexpression both displayed specific phenotypic responses to TIS de-N-glycosylation or TIS-specific fucosylated PON1 inhibition. In both modified TIS conditions, silencing RPS27L expression or over-expressing DKK1 promoted growth acceleration and EGFR-TKI resistance of DR clones reversing the supposed growth limitation and sensitization under such TIS modifications (*Figure 6—figure supplement 3B and C*). Further, both targeted approaches prevented senescence induction and stimulated TME-associated resistance genes upon de-N-glycosylation of TIS or inhibition of TIS-specific PON1 fucosylation (*Figure 6—figure supplement 3D*). Collectively, our data indicate that modulatory genes controlling DR clone response to inhibited secretome PON1 fucosylation are functionally associated with drug sensitivity to targeted therapies and are potential therapeutic targets to limit DR clone outgrowth.

## Discussion

Here, we identify a distinct N-glycome signature of the pan-cancer TIS. Using complementing fuco-sylation enrichment and detection approaches, we show that an induced secretome <60 kDa protein fucosylation is systemically aberrant in cancer cells, tumor xenografts, and patients upon targeted therapy. This modification appeared to be selected during resistance evolution as cell-derived DR clones, DR clone-derived tumor xenografts, and relapsed cancer patients display the same secretome aberration. Counterintuitively, TIS marks both response and resistance to targeted therapy (*Obenauf et al., 2015*; *Massagué and Obenauf, 2016*). Subsequent regression of tumors and their TME in response to targeted therapy lead to the release of TIS that feeds the outgrowth of minority DR

clones and survival of other cellular components (i.e., stromal cells) of the targeted microenvironment (*Obenauf et al., 2015*; *Massagué and Obenauf, 2016*). We reveal that core fucosylation of the TIS augments this effect. De-N-glycosylating the TIS by a glycosidase suppressed critical resistance-mediating survival cues and promoted a senescent state in regressing cell admixtures. Thereby glycans bound to N-glycosylated scaffolds of the TIS, not the released N-glycans per se, are required to establish a resistance-conferring niche. Mechanistically, directly blocking the transport of GDP-Fuc into Golgi or transfer of fucose onto proteins prevent the population rebound of remnant DR clones, encouraging a more drug-responsive cell population.

To date, there are 11 FUTs and two POFUTs known to catalyze fucose transfer from donor GDP-Fuc to various acceptor scaffolds such as glycoproteins and glycolipids, following Golgi-specific transport of GDP-Fuc by SLC35C1 (*Blanas et al., 2018*; *Schneider et al., 2017*; *Keeley et al., 2019*). These enzymes can compete in a mutually exclusive fashion to synthesize glycans in the Golgi and are exploited during tumorigenesis. While previous studies have implicated FUTs in multidrug resistance in several cancer types (*Keeley et al., 2019*), there currently has no systemic analysis that describes the degree and scope of this connection. Mapping the pan-cancer pharmacogenomic profiles of these FUTs and SLC35C1 revealed that their expression broadly correlated with resistance to multiple targeted therapies while their inhibitory cues (i.e. promoter methylation and GDP-Fuc binding site mutations) are widely associated with a drug-sensitive state. We show that the FUK-SLC35C1-FUT8 core fucosylation axis is significantly correlated with patient relapse in both large and small patient cohorts. This fucose metabolism pathway appears to be a prerequisite in driving our observed secretome fucosylation in drug-stressed cells and DR clones. High-degree pan-cancer expression of FUT8 and its activity in the Golgi entail the substrate specificity of FUT8 in fucosylating scaffolds in the secretory pathway. In a much broader context, we corroborated this by showing that the gene set encoding for CCS components contains a subset of glycosylation genes that display increased expression signatures. We reveal that FUT8 or SLC35C1 can directly regulate the distinct <60 kDa secretome fucosylation specifically in DR clones but not in sensitive cells. We note that >100 kDa secretome fucosylation can effectively be mediated by either of the two factors in sensitive cells, uncovering differential target processing of core fucosylated products prior to their secretion.

Our discovery of PON1 fucosylation as a component of the pan-cancer TIS contextualizes its systemic regulation in cancer patients upon therapy. Our previous study along with others suggests a compelling serological biomarker potential for fucosylated PON1 in advanced SCLC and early HCC (*Ahn et al., 2014*; *Aldonza et al., 2017*; *Sun et al., 2012*; *Zhang et al., 2015*). While it is conceivable that overabundance of fucosylated PON1 in the secretion is due to overacting FUTs (i.e. FUT8) and fucose metabolic reactions in the liver, it does not provide an intuitive explanation for reduced serum PON1 level and restricted enzyme activity in multiple cancer patients and mouse models profiled previously (*Ahn et al., 2014*) and in this study. This lack thereof has led us to examine how fucosylation influences the stability of PON1 prior to its secretion. Our data show that core fucosylation at a sequon located in the terminal region of the arylesterase domain, a conserved site among mammals, determines PON1 stability and assures proper folding prior to the secretion of PON1 from DR clones or PON1-overexpressing cells. This indicates that induction of the core fucosylation pathway rewires the maturation (i.e. folding) of PON1 along the secretory route, generating a more degradation-resistant PON1 with altered enzyme activity. We speculate that in disease states where there is an abundant serum fucosylation, non-fucosylated or less fucosylated PON1 cannot persist longer because of proteolytic insults in the blood. Among the three-member PON family, PON1 and PON3 are both secretory antioxidants bound to high-density lipoprotein (HDL) and share considerable structural homology (*Harel et al., 2004*; *Draganov et al., 2005*). Our data revealing PON3-mediated PON1 fucosylation in the Golgi hence establishes altered Golgi redox homeostasis prior to the secretion of proteins. We also showcased that such PON3-dependent PON1 fucosylation can be directed both at protein interaction and transcriptional control of PON3. Together with the data on VWF and MPO and the contrasting consequences of their interactions with PON1, it can be argued that regulation of PON1's N-glycosylation state might depend on how interacting co-factors influence PON1 enzyme activity in the golgi or in secretion, at least in the setting of targeted therapy or resistance. Regardless, these are potential entry points to develop strategies to specifically control PON1 fucosylation independent of directly manipulating PON1, fucose salvage pathway, or GDP-fucose processing and attachment, all of which have physiological importance in non-disease states.

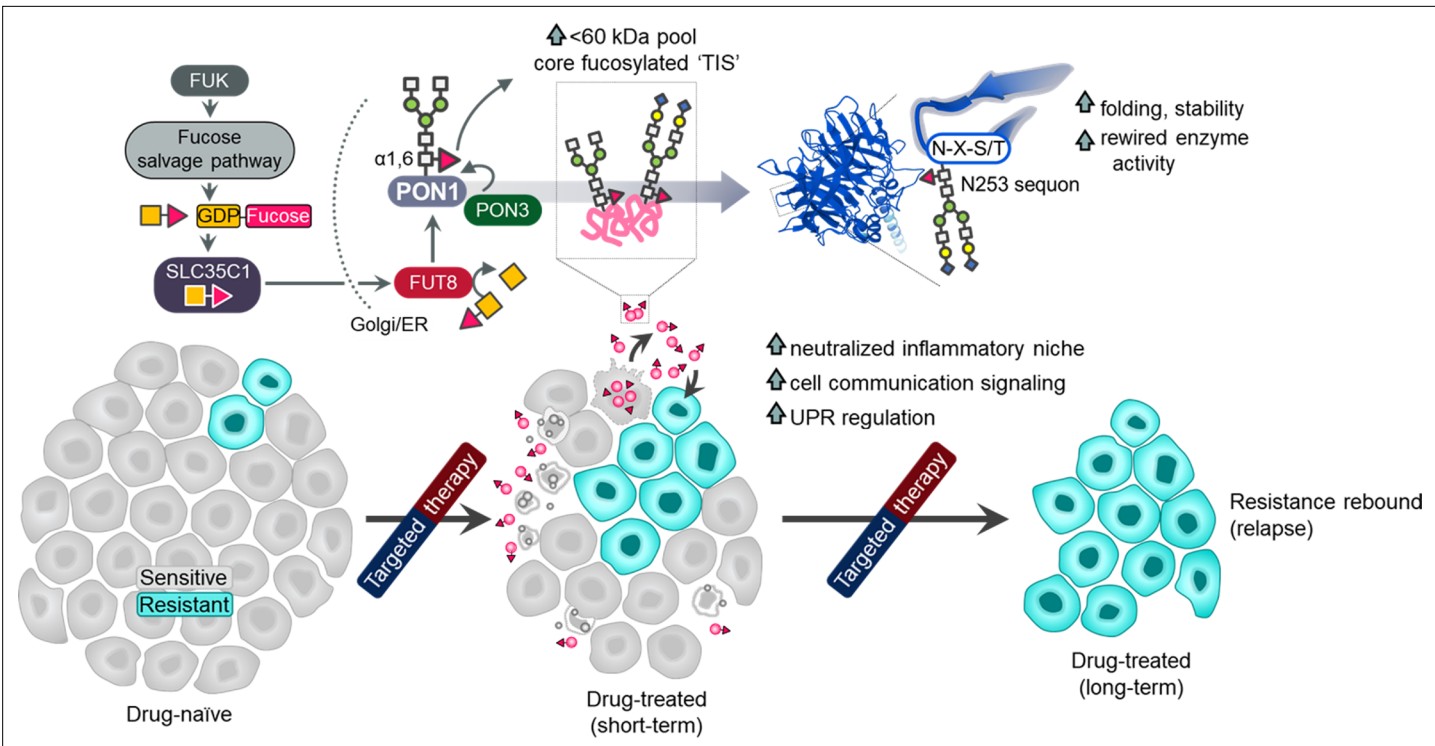

**Figure 7.** Schematic summary of the model proposed in this study. Drug-sensitive cancer cells responding to kinase inhibitors induce a core fucosylated, complex secretome of mostly <60 kDa proteins stimulating the clonal expansion of DR cancer cells, potentially contributing to disease relapse. Secretome core fucosylation is driven by the fucose salvage-SLC35C1-FUT8 axis. Among such secreted proteins that are heavily core fucosylated is the antioxidant PON1. Prior to secretion, PON1 is differentially fucosylated at multiple sites with the sequon at N253 critical for protein folding and stability. PON1 core fucosylation is tightly regulated by PON3 in the golgi, resulting to a more stable, degradation-resistant PON1 with a rewired enzyme activity in the secretion. Minority DR clones respond to the globally fucosylated TIS or core fucosylated secretome PON1 via activation of TF and gene effectors important for UPR, cell communication regulation and inflammatory niche neutralization.

Indeed, our pathway-focused screen on cell admixtures reveals that defective secretome PON1 fucosylation in the TIS promotes the expression of TFs that regulate response to oxidative stress and pro-inflammatory niche and repression of hypoxia in a suppressed DR clone. Concurrently, our transcriptome-wide analysis in a CM co-culture setting demonstrates that genes negatively regulating the response to stimulus and inflammatory signals, and cell communication act as modulators upon inhibition of secretome PON1 fucosylation. Thus, targeted strategies to control them might limit therapy resistance.

To this end, we report an aberrant signature of secretome core fucosylation functionally associated with multi-targeted therapy resistance in different cancer lineages. Our study highlights the fucosylation of PON1 as a component of a complex, reactive secretome induced upon targeted therapy and in turn stimulates resistance (*Figure 7*). This proof-of-concept study underscores new insights into the biological basis of cancer recurrence. We acknowledge that while our findings are all reproducible, they still require further validation, perhaps using appropriate patient-derived models. Regardless, the generality of our findings implicates that targetable aberration in secretome fucosylation and modulatory factors controlling response to this niche should be considered in managing clinical cancer relapse.

## Materials and methods
### Data reporting and statistics

No statistical methods were used to predetermine sample size. The experiments were not randomized unless otherwise stated. The investigators were not blinded to allocation during experiments and outcome assessment. All quantitative data are presented as means ± SD unless otherwise specified.

Student's $t$, Mann–Whitney, Dunnett's, Wilcoxon rank-sum, Kruskal-Wallis, Mantel–Cox and chi-squared tests; and ROC analyses were performed with GraphPad Prism 8.4 or 9.5. The number of samples or biological replicates (n) is indicated in each figure panel. For bioinformatics, all adjusted p values ($p_{adj}$) were adjusted to control for the false discovery rate (FDR) using the Benjamini-Hochberg procedure. Statistical significance was defined as $p < 0.05$.

## Human cancer patient samples and ethics statement

All samples were selected and categorized randomly. Patients underwent surgical resection of their primary or metastatic tumors at Seoul National University Hospital. Cancer diagnosis was primarily based on core needle biopsied samples. Genotyping was primarily based on plasma and cerebrospinal fluid (CSF) analysis. Blood samples from the first cohort were routinely obtained. Tissue samples from the second and third cohorts were obtained by surgical resection. In the first cohort, 14 samples of plasma and 30 samples of sera from treatment-naïve and osimertinib-treated NSCLC patients with EGFR-activating mutations were obtained in a routine diagnosis. In the second cohort, 53 paired lung cancer tumor tissues and adjacent normal tissues were obtained during surgery. Detailed information on preoperative (neoadjuvant) systemic chemotherapy was not disclosed to the researchers but relapse (recurrence) information for all patients is available. In the third cohort, 33 breast cancer tissue samples were obtained during surgery. Patients received primary systemic therapy (PST) and adjuvant chemotherapy. Pathological complete response following PST was defined as complete disappearance of all invasive cancer or only residual ductal carcinoma in situ. In all cohorts, post- or preoperative radiation therapy was not performed. Blood and tissue processing and histopathological data interpretation were overseen by expert pathologist co-authors (HSR, SC, TMK). Clinicopathologic information from three patient cohorts was abstracted from medical records and de-identified as shown in *Supplementary file 1—supplementary file 1a-c*. Formalin fixation and paraffin embedding (FFPE) processing of human patient samples was performed under standard clinical pathology and sectioned at 3~5 µm at SNU Hospital. Source DNAs and RNAs were extracted either from archived FFPE (third cohort) or fresh frozen (second cohort) tumor and adjacent normal tissues. Lysates were obtained from frozen tumors. Frozen samples were 'snap-frozen' in liquid nitrogen and stored at −80 °C. For plasma collection, samples were centrifuged at 1600 $g$ for 10 min within an hour of the blood draw, then additional centrifugation of 20,000 $g$ for 10 min was carried out. For serum collection, blood was allowed to clot for 15–30 min at room temperature (RT) prior to the same centrifugation. All aliquots were stored at −80 °C. Each aliquot was thawed no more than twice prior to use. Multiple Affinity Removal System (MARS) HSA/IgG spin columns (Agilent) were used to deplete albumins and IgGs from blood samples. Depleted samples were concentrated using Amicon Ultra-2 mL Centrifugal Filters (Merck; 3 k molecular weight cut-off [MWCO]) according to manufacturer's instructions. Tumor tissue lysates were subjected to the same downstream depletion and concentration as described above.

## Cell lines

Human H292, H1993, H358, HCC4006, H460, H1299, and A549 cell lines [American Type Culture Collection (ATCC) nos. CRL-1848, CRL-5909, CRL-5807, CRL-2871, HTB-177, CRL-5803, and CCL-185, respectively; obtained in 2014–2016] were grown under standard conditions in RPMI 1640 (Welgene) supplemented with 10% fetal bovine serum (FBS) alternative Fetalgro bovine growth serum (RMBIO) or EqualFETAL bovine serum (Atlas biologicals), 2 mM L-glutamine, and penicillin (100 U/ml)–streptomycin (100 µg/ml; Invitrogen). PC9 and HCC827 [originally provided by J. K. Rho (Asan Medical Center, University of Ulsan, Seoul, Korea)], EBC-1 [Japanese Collection of Research Bioresources (JCRB) Cell Bank no. JCRB0820], HCC78 [German Collection of Microorganisms and Cell Cultures (DSMZ) GmbH no. ACC563], H3122 [originally provided by P. A. Jänne (Dana-Farber Cancer Institute, Boston, MA, USA)], and SKBR3 [originally provided by D. M. Helfman (KAIST, Daejeon, Korea)], all obtained in 2017, cell lines were grown in RPMI 1640 with the same supplementation as mentioned above. Human H1975, H2009, A375, HEK-293T cell lines [ATCC nos. CRL-5908, CRL-5911, CRL-1619, CRL-3216; obtained in 2017] were grown in Eagle's minimum essential medium (Merck), DMEM/F12 (Gibco), and DMEM (Welgene) with the same supplementation as mentioned above except without additional L-glutamine and contained in addition 1 µg/mL amphotericin B. Mouse LL/2 (LLC1; ATCC no. CRL-1642; obtained in 2015) was grown in BME with Earle's salts (Merck) with the same supplementation as mentioned above. All cells were grown in a humidified incubator at 37 °C with 5% $CO_2$

and were tested regularly for mycoplasma contamination. All cell lines used were negative for mycoplasma (Cosmogenetech mycoplasma test service).

## Animal studies

All experiments using animals were performed in accordance to our protocols approved by the Institutional Animal Care and Use Committee (IACUC) of Seoul National University (SNU) (through the College of Pharmacy and College of Veterinary Medicine) and KAIST (through GSMSE). 4~6-week-old male athymic mice (BALB/c-nu) purchased from Orient Bio were used for animal experiments with human cell lines. Mice were transferred, housed, and bred under standard pathogen-free (SPF) conditions in common animal facilities of SNU and KAIST, and were fed with free access to standard diet (PMI LabDiet) and water. For tumor xenograft assays, mice were randomly assigned to one of two experimental groups (control or kinase inhibitor) in each of the four cohorts consisting of four to five mice per cohort. For subcutaneous tumor cell transplantations, freshly passaged cell suspensions ($1\times10^7$ cells for H1993 cohort; $1.2\sim1.4 \times 10^7$ cells for PC9 cohort; and $2\times10^6$ cells for HCC827 cohort) were prepared in 100~200 μL culture media:growth factor-reduced Matrigel (Corning) in a 1:1 ratio and injected into the right flank of each mouse (except in H1993-GR PON1 cohort; see details below) using 26-gauge 0.5 mL Sub-Q needle syringe (BD). Up to two sites on the flanks were injected per mouse. Mice were treated when their tumor volumes reached 70–100 mm³ (H1993 cohort), 120–150 mm³ (PC9 cohort), and 50–100 mm³ (HCC827 cohort). Gefitinib (200 or 400 mg/kg body weight), erlotinib (10 mg/kg body weight), and osimertinib (5 mg/kg body weight) were dissolved in 100~200 μL vehicle solution (Tween 80-EtOH-H2O in 1:1:98 ratio) and were orally administered to mice in appropriate cohorts once daily per designated timelines for each treatment (14-day schedule in a 22-day cycle for H1993 cohort, 14-day schedule in a 28-day cycle for PC9 cohort, and 21-day schedule in a 28-day cycle for HCC827 cohort). The control group was treated with an equal volume of vehicle with DMSO. For the H1993-GR PON1 cohort, WT and mutant clones were prepared as $4\times10^6$ cell suspension in 50 μL media and Matrigel mixture and injected as described above. Mice were treated when their tumor volumes reached 90–100 mm³. Gefitinib was administered as described above. Tumor volume was determined using digital caliper measurements, monitored for the whole duration, and calculated using the following formula: tumor volume = (D×d2)/2, in which D and d refer to the long and short tumor diameter, respectively. The body weight of each mouse was also monitored for toxicity. Mice were sacrificed at days 22 (H1993 cohort), 28 (PC9 and HCC827 cohorts), and 40 (H1993-GR PON1 cohort). All mice were maintained under continuous sedation by administering 3~4% isoflurane via an anesthesia mask during surgery and prior to euthanasia. Mice were sacrificed by cervical dislocation at the end of each cycle or when total tumor volume was larger than 1500 mm³.

For tumor sampling, residual (regressed) tumors after therapy and time matched tumors from vehicle-treated control mice were collected via surgical excision at least 1 week after the last dose of targeted therapy or at the end of cycle. Excised tumors were cleaned of surrounding fat tissue, washed multiple times in PBS, and were stored in RNAlater solution at 4 °C until ready for DNA and RNA experiments, formalin-fixed at 4 °C for FFPE processing, or flash frozen and stored at –80 °C until ready for other experiments. FFPE processing of xenograft tumors was performed at histology core of SNU College of Veterinary Medicine and sectioned at 5 μm using RM2245 semi-motorized rotary microtome (Leica). For whole blood sampling, end-of-cycle mice were anesthetized with 3~4% isoflurane and 0.1~0.2 mL of blood was collected from the retro-orbital sinus using a microcapillary pipet into a collection tube containing dipotassium EDTA (K2 EDTA). To process serum, collected blood was allowed to clot for 30 min at RT and centrifuged at 1350 RCF for 15 min. Sera were collected off the top into a collection tube. Sera were immediately stored at –80 °C. Sera proteins and tumor lysates were subjected to the same downstream depletion and concentration as described above.

## Drug-resistant clones

To generate DR clones, sensitive cell lines were seeded at low density and continuously exposed to gradually increasing concentrations of the drug for at least 12 weeks and for as long as >52 weeks. All clones were derived and expanded from colonies and maintained at specific drug concentrations. Clones were passaged every 2 or 3 days with adding fresh drug concentration. Characterization of resistance is summarized in *Figure 1—figure supplement 6* (references in the figure: *Bach et al., 2018a*; *Kim et al., 2019*; *Bach et al., 2018b*).

## Cell secretomes preparation

To generate secretomes, $3\times10^6$ sensitive cells and $7\times10^6$ DR clones were plated on 15 cm plates in standard media and allowed to adhere overnight. The media was then replaced with fresh media with 2% dialyzed FBS and indicated drugs for 48 hr. FBS was dialyzed in-house (against 0.15 M NaCl until glucose reached <5 mg/dL) using 10 k MWCO dialysis tubing (Fisher Scientific) at 4 °C for 6 hr. Secretomes were centrifuged at 1000 r.p.m. for 5 min, vacuum filtered using 0.45 µm cellulose acetate membranes (Whatman), and immediately placed on ice. For 2D co-culture, secretomes were stored at 4 °C, warmed prior to use, and were used only within 48 hr. For 3D co-culture, only freshly prepared secretomes were used and were further concentrated using Amicon Ultra-15 mL Centrifugal Filters (3 k MWCO). For biochemical assays, secretomes were further concentrated using Amicon Ultra-15 mL Centrifugal Filters (3 k, 10 k, 30 k, 50 k, 100 k MWCO as indicated) and depleted of albumins and IgGs using MARS HAS/IgG spin columns. Aliquots were 'snap-frozen' in liquid nitrogen and stored at −80 °C until use. Aliquots were thawed only once.

## N-glycosylation/fucosylation assays

For enrichment of core fucosylated proteins/lipids, we used AAL as a probe to capture scaffolds with bound fucose linked (α1,6) to N-acetylglucosamine or fucose linked (α1,3) to N-acetyllactosamine-related structures. We note that AAL also reversibly binds fucose attached to nucleic acids. Bio-spin columns (Bio-Rad) were packed with 1.5 mL agarose bead-bound AAL (Vector Laboratories). Agarose beads were initially maintained in an inhibiting solution [10 mM HEPES (pH 7.5), 0.15 M NaCl, 10 mM fucose, 0.04% NaN3] at 4 °C. Concentrated secretomes (500 µL) or sera/plasma (40 µL) were thawed at 4 °C in ice, diluted in 1.5 mL AAL adsorption buffer (AffiSpin-AAL kit; GALAB), incubated in ice for 5 min, loaded onto packed bio-spins, and incubated at 4 °C for at least 12 hr. Unbound proteins/lipids were removed by flow-through (only by gravity) and washing with adsorption buffer and PBS. Fucosylated proteins were eluted twice with 50 uL AAL elution buffer B1 or 40 µL glycoprotein eluting solution for fucose-binding lectins (Vector Laboratories) at 4 °C for 1 hr per round. Remaining bound fucosylated proteins were forcedly eluted. Samples were scaled up to produce at least 80 µL eluted proteins. All reagents and columns were pre-chilled in ice prior to use. Eluted proteins were precipitated using the trichloroacetic acid (TCA)-sodium deoxycholate (DOC) method as described previously with minor modifications (*Bensadoun and Weinstein, 1976*). Protein concentrations were calculated from absorbance measured using the Bradford reagent (Bio-Rad) at 595 nm in a microplate reader (Multiskan SkyHigh, Thermo Scientific or VersaMax, Molecular Devices).

We developed a sandwich ELLA assay to quantify fucosylated proteins in AAL-enriched samples (*Figure 1—figure supplement 7*). 96-well microtiter plates (Koma Biotech) were coated with 0.4 µg native, unconjugated MAL II, SNA, LCA, BTL, PSA, UEA1, ConA, or RCA1 lectins (Vector Laboratories) in 100 µL coating buffer (15 mM Na2CO3, 35 mM NaHCO3, 0.02% NaN3, pH 9.6) at 37 °C for 2 hr. The plates were additionally incubated with 0.1 mL oxidation buffer (20 mM NaIO4) per well. Lectin solution was removed by three washes with PBS-Tween-20 (0.05%; PBST). Plates were then blocked with 3% bovine serum albumin (BSA) in PBST for 1 hr at RT. Concentrated lectin-pooled cell secretomes, cell/tissue lysates, or sera/plasma were added to each well and incubated at RT for 2 hr. The plate was gently washed three times with PBST to remove unbound proteins. A total of 100 µL of 4 µg/mL biotinylated AAL (Vector Laboratories) was added and incubated at RT for 90 min. Lectin solution was removed and HRP-conjugated streptavidin (Biolegend) was added and incubated at RT for 90 min followed by two additional washes with PBST. 1-Step Turbo TMB-ELISA substrate solution (Thermo Scientific) was used for detection. Automated washing was performed using Well-wash Microplate Washer (Thermo Scientific). Absorbance was measured at 450 nm in a microplate reader (Multiskan SkyHigh or VersaMax). Results are indicated as fold change (relative to control) and normalized samples per amount of total lectin-pooled glycoproteins improving an optimized range of concentration of target glycosylation quantification.

For the N-glycan release assay, we optimized a previously described protocol to quantify the glycosidase-induced release of N-glycans (*Wang et al., 2019*). Briefly, 20 µL concentrated samples were mixed with 2.5 µL sodium phosphate or citrate buffer (500 mM, pH 7.5) and 10 µL total 8 U PNGase F or 10 U Endo S/F1 and incubated at 37 °C for 12 hr in a humidified chamber and heat-quenched at 95 °C for 5 min. Reactions were then mixed with 20 µL 2.5 M TCA solution, vortexed for 5 min, and centrifuged at 12,000 $g$ for 30 min. 15 µL supernatants were mixed with 7.5 µL of 4 M NaOH,

12.5 µL 1.7 mM aqueous WST-1 solution, and incubated for 1 h at 50 °C. For in-gel N-glycan release, in-gel proteins were trypsin digested overnight (see details below). Samples containing extracted peptides were reduced in a refrigerated SpeedVac (SRF110, Thermo Scientific) until at least 10 µL was reached. Reduced samples were mixed with 10 µL H2O and subjected to the same protocol as mentioned above. Absorbance was measured at 584 nm in a microplate reader (Multiskan SkyHigh). The amount of released N-glycans was quantified using maltose (Sigma), an N-glycan mimic in this assay, as an external standard. Results were normalized as in ELLA.

AAL blotting was performed as described previously (*Ahn et al., 2014*). Briefly, AAL-enriched precipitated samples (10–15 µg concentrated cell secretome or 3–5 µg of sera/plasma proteins) were subjected to 12% SDS-PAGE using Mini-PROTEAN Tetra Cell system (Bio-Rad) or omniPAGE WAVE Maxi system (Cleaver Scientific). The gels were wet or semi-dry transferred to nitrocellulose membranes (Whatman) using XCell SureLock (Thermo Scientific) or TE 77 (Cytiva). The membranes were blocked with 1 x Carbo-free blocking solution (Vector Laboratories) at 4 °C for at least 2 hr and incubated with 5–20 µg/mL of biotinylated AAL at RT for 1 hr. Membranes were washed three times with PBST, incubated with HRP-conjugated streptavidin at RT for 1 hr, washed three times with PBST, and developed using an ECL system (Amersham or Bio-Rad). Blots were detected using ChemiDoc MP system (Bio-Rad).

Glycoprotein staining of SDS-PAGE gels was performed using the GelCode glycoprotein staining kit (Pierce) according to the manufacturer's protocol. SDS-PAGE gels were scanned (Samsung Xpress) or imaged (ChemiDoc MP). Stained glycols appear as magenta/pink bands.

For HLE of target protein fucosylation, we modified a previously described protocol using an ELISA starter kit (Koma Biotech) (*Ahn et al., 2014*). Briefly, 96-well microtiter plates were coated with 120 ng PON1 (18155–1-AP, Protein Tech), AFP (ab3980, Abcam), or A1AT (ab9399, Abcam) monoclonal antibodies in 100 µL coating buffer at 37 °C for 3 hr. A total of 100 µrL of oxidation buffer was added per well for 30 min and blocked with 3% BSA in PBS for 2 hr at RT. The plates were washed four times with PBST. All AAL-enriched samples were diluted 10-fold in PBS, 100 µL of each sample was added to each well and incubated at RT for 2 hr. After multiple washes with PBST, 2 µg/mL biotinylated AAL was added and incubated at RT for 90 min. Lectin solution was removed, and HRP-conjugated streptavidin (Biolegend) was added and incubated at RT for 90 min followed by two additional washes with PBST. A 1-Step Turbo TMB-ELISA substrate solution (Thermo Scientific) was used for detection. Absorbance was measured at 450 nm in a microplate reader (Multiskan SkyHigh).

For lectin fluorescent staining of cells and paraffin sections, we used 15 µg/mL fluorescein-labeled AAL (Vector Laboratories) or 4 µg/mL FITC- or Alexa Fluor 594-conjugated UEA1 (Thermo Scientific) according to manufacturer's protocol and following standard immunofluorescence protocols. Confocal microscopy was carried out using a ZEISS LSM 780 ApoTome microscope (Carl Zeiss). Wide-field fluorescence microscopy was performed using EVOS M7000 (Thermo Scientific).

GDP-Fuc activity of FUT8 was assayed using GDP-Glo glycosyltransferase assay kit (Promega) following manufacturer's protocol. Luminescence was read on a luminometer (POLARstar Omega, BMG Labtech or GloMax Navigator, Promega).

For in-culture and exogenous secretome/serum de-N-glycosylation, 10 µg/mL recombinant PNGase F (9109 GH, R&D Systems; 36405.01, SERVA) and 8 U PNGase F (P0704L, NEB; P7367, Sigma) were used, respectively, unless otherwise specified, for indicated times described in each figure description. PNGase F was not removed in any of the in-culture experiments for indicated incubation periods, except in *Figure 2—figure supplement 3* where PNGase F-treated media was replaced with drug-containing fresh media for drug sensitivity assay. Protein/lysate sample de-N-glycosylation using PNGase F (NEB), Endo S (NEB), or Endo F1 (Sigma) was performed following manufacturer's protocol with slight modifications on incubation period.

## Cell tracking experiments

To fabricate a 3D tumor spheroid array, polydimethylsiloxane (PDMS)-based positive master mold with an array of 225 spherical microwells (15×15) was prepared as we previously described (*Cha and Kim, 2019*). The mold was immersed in 70% (v/v) ethanol and sterilized for 30 min in UV before use. Agarose powder (LPS solution) was added to RPMI 1640 at a concentration of 3% (w/v) and heated for a short time to dissolve completely. Before gelation, the fully-melted agarose solution was poured in a 35 mm cell culture dish (3 mL/dish; SPL), and the sterilized master mold was immediately inserted into

the gel solution to create the microwells. After the agarose was solidified at RT for 20 min, the master mold was gently removed. PBS (3 mL/dish) was added to the agarose-based microwell to keep them hydrated before use. Cell admixtures seeded into these microwells can immediately form spheroids (*Video 1* and *Figure 2B*). Cell line variability in the number and size of spheroids is observed per well (*Figure 1—figure supplement 6D*).

For tracking experiments in cell admixtures, sensitive cells were labeled with CellTracker-Green (CMFDA) while DR clones with CellTracker-Red (CMTPX) or -Deep Red (Thermo Scientific) or transfected with pCAG-LifeAct-RFP (Ibidi) according to the manufacturer's protocol. RFP-labeled clones were stably selected using geneticin (Thermo Scientific) following the manufacturer's instructions. Labeled cells and clones were filtered using a cell strainer (Merck) and seeded either as 'one pot' or 'sequential layer' 2D and 3D admixtures as detailed (*Figure 2A* and *Figure 2—figure supplement 5A*). 3D admixtures were live imaged while 2D admixtures were fixed with 3.7% formaldehyde in PBS for 5 min and washed with PBS prior to imaging. Live imaging of 3D tumor spheroids was performed using an inverted fluorescence microscope (Nikon Eclipse Ti) equipped with a CFI Apochromat TIRF objective inside a live-cell workstation (LICES, N-Biotek). Time-lapse images were acquired at 15 min frame intervals to minimize photobleaching and phototoxicity by high illumination and analyzed by 3D reconstruction of stacked axes. Imaging of 2D admixtures was performed using an inverted fluorescence microscope (Leica DMI3000 B) or using an automated fluorescence imaging system (EVOS M7000, Thermo Scientific). Fixed admixtures were also analyzed using an image-based cytometer (Arthur, NanoEntek or Tali, Thermo Scientific) for validation. Post-imaging cell tracking and fluorescence analyses were performed using the plug-in TrackMate in Fiji/ImageJ.

## Plasmids, RNAi, and transfections

Sixty nM to 120 nM target-specific smart pool (mix of at least two different sequences each; *Supplementary file 1—supplementary file 1d*) of short interfering RNAs (siRNAs) or non-targeting scrambled siRNA/siLuciferase (IDT Korea) were delivered with Lipofectamine 3000 (Life Technologies) or DharmaFECT (Dharmacon) according to manufacturer's instructions. Target siRNAs were obtained from IDT Korea, Life Technologies, Origene, or Bioneer. Unless otherwise specified, most assays were analyzed 48 hr post-transfection. pCMV6-AC-Myc-DDK and pCMV6-FUT8-Myc-DDK (Origene) expression plasmids were delivered with Lipofectamine 3000 or FuGene 6 (Promega) according to the manufacturer's instructions. Transfections were performed for 48 hr. To establish PON1 and SLC35C1 knockout (KO) cells, pLKO.1-puro or pLKO.1 plasmid encoding target shRNA constructs (*Supplementary file 1—supplementary file 1d*; selected from TRC shRNA Library, Broad; purchased from Origene) were cloned as previously described. The sequence of the constructs was verified by DNA sequencing (Origene or Bioneer). Scrambled shRNA (Addgene) was used as shControl. Lentiviral co-transfection of 8 μg of cloned transgene plasmids, 1 μg pMD2.G (envelope plasmid; Addgene), and 3 μg psPAX2 (packaging plasmid; Addgene) was performed using iN-fect (Intron Biotechnology) in HEK293T cells following manufacturer's protocol and transduction in indicated cell lines using standard procedures. Lentivirus titer was determined using Lenti-X p24 rapid titer kit (Takara Bio). 2–8 μg/mL puromycin was added gradually to select stable cell lines for two weeks. Stably selected KO cells were maintained in 0.1 μg/mL puromycin-containing complete media. To establish PON1- or DKK1-overexpressing cells, bicistronic pLVX-EF1α-IRES-puro (Takara Bio) encoding the CDS of human PON1 or DKK1 single mRNA transcript was cloned as previously described. The empty vector was used as a control (-CC). Co-transfection of plasmids, transduction, and selection were performed as above, except infected cells were selected in 1~3 μg/mL puromycin. Validation of targeted overexpression or RNAi is shown in *Figure 5—figure supplement 2*.

## Site-directed mutagenesis

To generate mutant PON1 constructs, a PCR-based Q5 site-directed mutagenesis kit (NEB) was used according to the manufacturer's instructions. PON1 cDNA template was cloned into pcDNA3.1 (Genscript) as described previously (*Boado et al., 2008*). The mutagenesis primers were designed using the Primer X Tool (http://bioinformatics.org/primerx/). WT or mutant PON1 constructs were transfected using Xfect transfection reagent (Clontech) according to the manufacturer's protocol.

## Gene expression analysis

Whole RNA (1–3 µg total per 10 µL volume) was isolated using RNAeasy mini kit (QIAGEN) or TRIzol (Life Technologies) following the manufacturer's protocol. Tumor tissues were homogenized using a handheld homogenizer (TissueRuptor, QIAGEN) or a bead mill-based homogenizer (TissueLyser LT, QIAGEN) in RLT-ME buffer (QIAGEN). Complementary DNA (cDNA) was generated using the Transcriptor First Strand cDNA synthesis kit (Roche). RNA was treated with deoxyribonuclease I (DNase I; Takara) and reverse-transcribed using RevertAid reverse transcriptase (Fermentas). cDNA was amplified by an SYBR Green PCR master mix (Applied Biosystems). Differential RNA levels were assessed using Taqman gene expression assays (Life technologies). Quantitative polymerase chain reaction (qPCR) was performed using SureCycler 8800 (Agilent), CFX Opus (Bio-Rad) or AriaMx (Agilent) Real-Time PCR Systems. Relative gene expression was normalized to internal control genes: GAPDH or ACTB. For nucleic acids extraction (total RNA and genomic DNA) from FFPE tumor samples, we used FFPE All-Prep kit (QIAGEN) following the manufacturer's protocol. A small portion of specimens was prepared from ~80 µm slices of FFPE tumor blocks, followed by dewaxing using Deparaffinization Solution (QIAGEN). Purified RNA was subjected to reverse transcription PCR (RT-PCR) and qPCR as above. Primers used in this study are detailed in *Supplementary file 1—supplementary file 1e*.

## Immunoblotting, immunoprecipitation, and cross-linking-mediated coimmunoprecipitation

Subcellular fractionation and enrichment of golgi and ER was carried out using Minute Golgi/ER enrichment kit (Invent Biotech), and isolation of nuclear and cytosolic extracts was carried out using NE-PER Nuclear and Cytoplasmic Extraction Reagents (Pierce) following the manufacturer's instructions. Whole-cell pellets were lysed as described previously (*Aldonza et al., 2020*). Following surgery, xenograft tumors were flash-frozen in liquid nitrogen. A portion of the frozen tumor excised from mice was thawed on ice and homogenized in Complete Lysis Buffer (Active Motif) for whole lysate extraction Protein concentrations were determined using Bradford reagent. Samples were boiled for 5 min in Laemmlli buffer. Equivalent amounts of proteins (usually 30–50 µg) were separated by SDS-PAGE (usually on 7.5, 10, and 12% gels). For immunoprecipitation, PON1 monoclonal antibody was coupled to protein G-Sepharose 4B beads (GE Healthcare) and eluted as described previously (*Aldonza et al., 2021*). Proteins were transferred onto Immobilon PVDF membranes (Millipore) using a semidry (TE 77 or Trans-Blot Turbo) or wet (XCell SureLock) transfer systems. The detection system was Clarity Max Western ECL Substrate (Bio-Rad) and Western Lightning Plus-ECL (PerkinElmer). Secondary antibodies were either goat antibodies to mouse immunoglobulin G–horseradish peroxidase (IgG-HRP; DACO), mouse IgGκ-HRP (Santa Cruz Biotechnology) or donkey antibodies to rabbit IgG-HRP (GE Healthcare). Blots were detected using ChemiDoc MP system (Bio-Rad). For cross-linking-mediated coimmunoprecipitation, cells were pre-starved in media containing 2% dialyzed FBS prior to cross-linking using 1 mM ethylene glycolbis(succinimidylsuccinate) (EGS) for 45 min at 4 °C as previously described. Briefly, lysates were diluted twofold in assay buffer and incubated with capture beads for FUT8 (protein G-agarose beads; Abcam) overnight. Lysates were clarified by centrifugation at 16,400 *g* for 15 min, and then precleared for 30 min with agarose resin. The lysate was then incubated with protein A/G agarose, immunoglobulin G control, or PON1 antibody overnight at 4 °C. The next day, the resin was washed six times with lysis buffer and then incubated with 2 M hydroxylamine HCl in phosphate-buffered saline (pH 8.5) for 6 h at 37 °C. The resin was then removed, and the supernatant was used for the indicated assay. Primary antibodies used for immunoblotting were PON1 (ab24261, Abcam), RCAS1 (12290, CST), SREBP2 (ab30682, Abcam), B-actin (ab8227, Abcam), and GAPDH (6C5, Santa Cruz Biotechnology). Antibodies used for (co)immunoprecipitation were PON1 (17A12, Santa Cruz Biotechnology), PON3 (ab109258, Abcam), and FUT8 (ab191571, Abcam).

## Sandwich- and bead-based ELISA

Sandwich-based ELISA kits were used to detect PON1 (RayBiotech), FUT8 (LSBio), and ATF6 (Novus Biologicals) following the manufacturer's protocol. For ELISA detection of SLC35C1, 96-well microtiter plates were manually pre-coated with SLC35C1 antibody (CSB-PA839285LA01HU, Cusabio) similar to HLE. Absorbance was measured at 450 nm in a microplate reader (Multiskan SkyHigh). For bead-based ELISA, established conjugatable ELISA antibodies for PON3 (ab242568, Abcam), VWF (AB7356, Sigma), and MPO (EPX01A-12038–901, Invitrogen) and Quantikine calibrated standards (R&D Systems)

were used. Cross-linking-mediated coimmunoprecipitation was performed as described above, incubated with identifiable capture beads targeting non-PON1 proteins, washed with wash buffer and then incubated with a biotinylated antibody for PON1 for 5 hr at 4 °C. After multiple washes, the beads were incubated with streptavidin-conjugated phycoerythrin for 30 min. The capture antibody was conjugated to unconjugated Bio-Plex Pro Magnetic COOH Beads (Bio-Rad) using Bio-Plex Amine Coupling Kit (Bio-Rad), except for MPO (ProcartaPlex system, Invitrogen) and used in a multiplexed fashion. Linearity of the assay was validated during measurement by dilution series of both lysates and standards. Flow cytometry-based multiplex quantification was performed using Bio-Plex 3D (Bio-Rad).

## Polypeptide synthesis assay

EZClick global protein synthesis kit (Biovision) was used to detect nascent protein synthesis following the manufacturer's protocol. This assay is based on an alkyne analog of puromycin, O-Propargyl-puromycin (OP-puro). OP-puro stops translation by forming covalent conjugates with the nascent polypeptide chains. Truncated polypeptides are rapidly turned over by the proteasome and can be detected based on a click reaction with the fluorescent azide. Fluorescence was measured by flow cytometry using LSR Fortessa (BD Biosciences). Excitation and emission wavelengths were set at 440 and 530 nm, respectively. Analysis was done using BD FACSDiva software.

## Trypsin sensitivity and CHX chase assays

To evaluate the folding status of PON1, we exogenously treated lysates with trypsin as described previously (*Xu et al., 2013*). Briefly, lysates were clarified by centrifugation at 17,800 *g* at 4 °C for 10 min. One mg/mL 50 µL aliquots of cleared lysates were incubated with 2 µL of indicated trypsin concentration (Promega) at 4 °C for 15 min. 50 µL stop buffer (1×SDS sample buffer, 100 mM dithiothreitol, 10×protease inhibitor cocktail) was added to the samples and incubated at 100 °C for 5 min. Thirty µg of each sample was separated by SDS-PAGE and immunoblotted. To evaluate PON1 stability, cultured cells in six-well plates were incubated with 25 µg/mL CHX (Sigma) at indicated times. Cells were subjected to immunoblotting or other assays as indicated.

## Phospho-RTK array and kinase phosphorylation assays

Phosphorylated RTKs were measured using PathScan human RTK signaling antibody array kit (R&D Systems) according to the manufacturer's instructions. Tyrosine 1068 phosphorylation of EGFR, pan-tyrosine phosphorylation of MET and HER3/ErbB3, and tyrosine 1150/1151 phosphorylation of Insulin Receptor β were assessed by solid-phase sandwich ELISA (CST PathScan kits) following manufacturer's protocol. The assay quantitatively detects endogenous levels of the indicated targets. Absorbance was measured at 450 nm in a microplate reader (Multiskan SkyHigh).

## Cignal 45-pathway reporter array

Cignal 45-pathway reporter arrays (QIAGEN) were used to simultaneously measure the activity of 45 transcription factors/signaling pathways according to the manufacturer's protocol. Briefly, cell admixtures grown for 5 days under different conditions were transferred to Cignal Finder 96-well plates (at least 30,000 cells/well). Reporter constructs resident in each well were introduced into cells via reverse transfection. Cell admixtures were grown in Opti-MEM (Gibco) supplemented with 5% FBS and 0.1 mM MEM non-essential amino acids (NEAA; Gibco) for 48 hr. Cell admixtures were then lysed, and luciferase activity was measured using dual-emission optics of a plate reader (POLARstar Omega or GloMax Navigator).

## Proliferation, survival, cell cycle, apoptosis, and senescence assays

Cell proliferation and survival were assessed by sulforhodamine B (SRB) and colony formation assays, as we previously described (*Aldonza et al., 2021*). Cell sorting, cell cycle analysis by quantitation of DNA content, and cell death detection in the sub-G1 peak were performed by flow cytometry as we previously described using FACSCalibur (BD Biosciences) (*Aldonza et al., 2021*). Analysis was done using BD CellQuest Pro software. At least 20,000 cells were used for each analysis. Changes in the percentage of cell distribution at each phase of the cell cycle were determined. To isolate apoptotic bodies, cells grown in indicated conditions were transferred to serum-free media with 0.35% BSA, and cell debris was collected after 24 hr. Cells were centrifuged at 300 *g* for 10 min, the remaining

cell debris was removed, and the soluble secretome was collected. The mixture was centrifuged at 16,500 g for 20 min using a super-speed vacuum centrifuge (Vision Scientific). To detect senescence, we measured SA-β-gal activity using senescence β-galactosidase staining kit (CST) following the manufacturer's protocol. SA-β-gal positive cells were quantified based on three independent images from differently stained regions analyzed by digital inverted light microscopy (40×phase-contrast; Leica DMi1). To evaluate senescent gene signature and SASP activity, we measured the gene expression of p16, LNMB1, IL-1α, IL-6, MMP-3, MMP-9, CXCL-1, CXCL-10, and CCL20 by qPCR.

## Caspase activity and intracellular ATP assays

Caspase 3/7 and 9 activities were assessed using a fluorescence-based Apo-ONE homogenous caspase 3/7 assay kit (Promega) and luminescence-based caspase-glo 9 assay system (Promega), respectively, following the manufacturer's protocol. Excitation and emission wavelengths were set at 560 and 590 nm, respectively. Fluorescence was read on dual-monochromator fluorometer (SpectraMax Gemini EM, Molecular Devices). Luminescence was read on a luminometer (GloMax Navigator). For ATP measurement, cells were seeded in 96-well plates and were subjected to indicated treatment/culture conditions all in nutrient-restricted media (10% dialyzed FBS). ATP levels were measured using the luminescence-based ATPLite system (Perkin-Elmer) following the manufacturer's instructions in a luminometer (GloMax Navigator).

## Enzyme activity assays

Paraoxonase activity was assessed based on 4-nitrophenol formation as described previously (*Elkiran et al., 2007*). Paraoxon (O,O-Diethyl O-(4-nitrophenyl) phosphate; Sigma) was used as a substrate. Absorbance was measured at 412 nm. One unit of paraoxonase activity was defined as 1 nM of 4-nitrophenol formed per min. Arylesterase activity was assessed based on phenol formation as described previously (*Elkiran et al., 2007*). Phenylacetate (Sigma) was used as a substrate. Absorbance was measured at 217 nm in a microplate reader (Multiskan SkyHigh). One unit of arylesterase activity is equal to 1 mM of phenylacetate hydrolyzed per min.

## ROS/RNS and cytokine measurements

Free radical ROS/RNS was measured using OxiSelect in vitro ROS/RNS assay kit (Cell Biolabs) according to the manufacturer's protocol. This assay used a DCFH probe, and oxidative reactions were measured against H2O2 or DCF standard. Excitation and emission wavelengths were set at 480 and 530 nm, respectively. IL-6, TFN- α, and GM-CSF levels were quantified using ELISA kits pre-coated with indicated capture antibodies per manufacturer's instructions (Sigma). IL-6 levels were preliminarily detected using a Q-Plex Human cytokine screen (16-plex; Quansys Biosciences). Absorbance was measured at 450 nm in a microplate reader (Multiskan SkyHigh).

## Immunofluorescence

Cells were plated onto 0.1% gelatin-coated glass-bottom 30 mm dishes (except for 3D tumor spheroids) and incubated overnight unless otherwise specified. The cells were fixed with 4% paraformaldehyde (in PBS) for 8 min at RT, quenched for 1 min in 10 mM Tris (in PBS) at RT, and permeabilized in 0.1% Triton X-100 (in PBS). Cells were then blocked in 2% bovine serum albumin (BSA) (in PBS containing 0.01% Triton X-100) for 30 min at RT and incubated with primary antibodies diluted in 2% BSA for 2 h. Alexa Fluor or fluorescein isothiocyanate–conjugated secondary antibodies were used to label primary antibodies. DAPI (4',6-diamidino-2-phenylindole; 0.35 µg/ml) was used to counterstain the nuclei. Cells were mounted using VECTASHIELD Mounting Medium (Vector Laboratories). Confocal microscopy was carried out using a ZEISS LSM 780 ApoTome microscope (Carl Zeiss). Primary antibodies used for immunofluorescence were PON1 (ab24261, Abcam), PON3 (ab42322, Abcam), ATF6 (PA5-20215, Invitrogen), RCAS1 (12290, CST), and XBP1 (ab37152, Abcam).

## Immunohistochemistry

Human or mouse FFPE tumor tissue sections were deparaffinized in xylene alternative (Histo-Clear, EMS; 3×5 min) and rehydrated in EtOH/H2O gradient series (100%, 95%, 70%, 40%, 5 min each). The rehydrated sections were washed in TBS for 10 min. Epitopes were unmasked using the heat-induced retrieval method with the use of the pre-heated Tris (10 mM Tris base, 0.05% Tween 20, pH

8.0) and citrate buffers (10 mM sodium citrate, 0.05% Tween 20, pH 6.0). Sections were pressure-cooked for 12 min in Tris buffer, transferred to citrate buffer, heated for 12 min, cooled at RT for 40 min, and washed with TBS containing 0.1% Twee-20 for 10 min. Sections were permeabilized with 0.3% Triton X-100 in TBS for 45 min and washed in TBS (2×5 min). Endogenous peroxidase activity was quenched in a peroxidase solution (0.3% $H_2O_2$ in TBS), and sections were blocked in a universal blocking solution (10% normal donkey serum in 1% BSA/TBS) or carbo-free blocking solution (for fucosylation detection) for 2 h. Slides were blotted to remove the serum, and then primary antibodies were applied at predetermined concentrations (1:400 or 1:800). Slides were incubated overnight at 4 °C in a humidified chamber and washed with TBS (3×5 min). Biotinylated link and HRP-conjugated secondary antibodies were applied onto sections and were incubated for 2 hr in a dark humidified chamber RT followed by washing. Replicate slides were also stained H&E (Vector Labs) according to the manufacturer's protocol. A VECTASHIELD hard-set mounting medium (Vector Labs) was used to mount the slides. The positive staining density was measure using a Leica CCD camera connected to a Leica DMi1 microscope. Biotinylated AAL (20 µg/ml) was used to detect fucosylation. Primary antibodies used were PON1 (18155–1-AP, Protein tech) and PON3 (OTI1A5, Thermo Scientific).

## Label-free proteomics

Precipitated AAL-enriched secretomes (45 µg) were run on a 1 mm thick 10% SDS-PAGE gel and stained with CBB G-250 staining solution (Bio-Rad) at RT for 1.5 hr. 30–70 kDa lane portions were excised into 2×2 mm cubes and transferred to Protein Lo-Bind tubes (Eppendorf). Excised gels were partitioned into tubes, and destained multiple times in 75 mM ammonium bicarbonate (Sigma) and 40% EtOH (1:1) in a shaking rack. Destained gel pieces were washed with 100 mM ammonium bicarbonate and acetonitrile (1:1), vortexed, and incubated at RT for 15 min. Gel pieces were diluted with 100 mM ammonium bicarbonate and reduced with 10 mM dithiothreitol at 51 °C for 1 hr. Gel pieces were cooled down to RT for 30 min followed by alkylation with 20 mM of iodoacetamide at RT for 45 min in the dark. Gel pieces were dehydrated in 100% acetonitrile and dried in a SpeedVac. In-gel proteins were digested with trypsin at a protein:enzyme ratio of 20:1 at 37 °C for 12 hr in a shaking incubator. Peptides were extracted in 100 µL extraction buffer (5% acetic acid/acetonitrile; 1:2) and incubated at 37 °C for 15 min in a shaking incubator. Tryptic peptide mixture was eluted from the gel with 0.1% acetic acid.

Mass spectrometry was performed as we described previously (*Ahn et al., 2014*). Briefly, nanospray liquid chromatography-tandem mass spectrometry (LC-MS/MS) was performed on an LTQ-Orbitrap mass spectrometer (Thermo Electron) coupled to Agilent 1200 series G1312B binary pump SL and NanoLC AS-2 autosampler (Eksigent Technologies). Peptide mixtures (2 µL per sample) were loaded via the autosampler on 75 µm (inner diameter) fused silica capillary columns with electrospray tip packed with C18 reversed-phase resin (Magic C18, 5 µm particles, 200 Å pore size; Michrom BioResources). Peptides were separated by reversed-phase liquid chromatography with mobile phases as we described previously. The tandem mass spectra were processed using Sorcerer 3.4 beta2 (Sorcerer Web interface). All MS/MS samples were analyzed using SEQUEST Cluster (Thermo Scientific), and Mascot generic format (MGF) files were set to query the human IPI v3.68 database. Searches were performed with and without oxidation of methionine and carbamidomethyl modification of cysteine as variable modifications. False positives and false discovery rates were calibrated through the decoy option during data search in Sorcerer to reduce noise effects. Scaffold v4.0.5 (Proteome Software) was used to validate MS/MS-based peptide and protein identification. PeptideProphet was used to validate peptide and protein assignments to MS/MS spectra (>95% probability). Subtractive proteomic analysis for each dataset was performed by normalization using total ion current (TIC; normalized by average of all the TIC values of the spectra assigned to a protein). MS RAW files were processed in MaxQuant (*Tyanova et al., 2016*), version 1.5.5.2. The FASTA file Homo_sapiens.GRCh38.pep.all.fa was downloaded from Ensembl.

## RNA-seq

Low-density H1993-GR were grown under indicated conditions for 48 hr (*Figure 6A*), and the whole RNA was extracted using the RNAeasy mini kit. 2×101 paired-end RNA-seq libraries were constructed using TruSeq stranded total RNA H/M/R prep kit and sequenced using the Novaseq6000 system (Illumina). Raw paired-end sequencing reads were mapped to the human genome (build hg38) with

HISAT2 v2.1.0 using default parameters except with the options '--dta' and '--dta-cufflinks'. Stringtie v.2.0.6 was used to quantify the expression of genes and transcripts by employing transcriptome information from GENCODE v27. Ballgown package was used to perform differential gene expression analysis generating FPKM for each gene. Genes with FDR <0.05, fold change larger than 2 or smaller than 0.7-fold, and average read counts larger than 10 were treated as differentially expressed genes. Gene ontology analysis was performed using DAVID 6.8. GO of biological process or molecular function were detected and summarized. GO terms with $P<0.01$ were selected as significant. Semantic-similarity network visualization of GO terms was done using REVIGO (http://www.revigo.irb.hr). Hierarchical clustering was performed using pheatmap library in R. Row-value filtered FPKM values were analyzed using default options. Heatmap colors indicate z-score in each row (*Figure 6—figure supplement 1A*).

## Bioinformatics

Drug response (as IC50 per drug and cell line) and gene expression data (as log2 transformed RMA normalized basal expression or RNA-seq TPM expression per cell line) were derived from GDSC (v2, accessed from http://www.cancerRxgene.org) and CCLE (v2, accessed from http://www.depmap.org) projects. All IC50s are expressed in µM. Categorical grouping of cell lines per cancer type was done and plotted in R (see code availability). The discretization threshold for each drug (log IC50/cell line) was determined as described previously (*Iorio et al., 2016*; *Boado et al., 2008*; *Aldonza et al., 2020*). Cell lines without corresponding drug or expression measurement were not included in the analysis. Drug sensitivity data were evaluated using IC50 values (including extrapolated values) for GDSC and activity area for CCLE. Correlation analysis between drug response and gene expression per cancer type was performed by quantitatively matching pre-processed values. All Spearman's correlation coefficients, relative quantitation, and plotting were performed in Python (see code availability). Only correlations with p<0.05 are shown. Summary reference on drug categories and target pathways is accessible in GDSC (available as Excel file TableS1F.xlsx in their database). The mutation dataset was obtained from CCLE (v2, accessed from http://www.cbioportal.org), and FUT domain information was searched in the Pfam database (http://www.pfam.xfam.org). CTRP dataset was analyzed using the CARE algorithm (http://www.care.dfci.harvard.edu).

CCS gene set (n=1810) was obtained and filtered from UniProt as described previously (*Robinson et al., 2019*). Glycosylation (N-/O-glycosylation) gene sets were obtained by conducting gene set analysis using the GOs 'glycosylation', 'protein N-linked glycosylation', and 'protein O-linked glycosylation' from MSigDB. The GO 'glycosylation' contains 22 annotated sub-GOs. These gene sets were used in their complete form and were not filtered. Both total and overlapping glycosylation genes (n=264 or 19 for overall glycosylation, n=81 or 1 for N-linked, and n=193 or 18 for O-linked) with the CCS gene set were included in the analysis. For CCS, missing values on FC_2 were removed, and 0 was considered as a missing value. p Values were calculated by two-sided Student's *t*-test and adjusted ($p_{adj}$) to control for FDR using the Benjamini-Hochberg procedure. Methylation data analysis was performed using the pre-processed reduced representation bisulfite sequencing (RRBS) dataset from CCLE v2. Drug sensitivity data were obtained from GDSC as mentioned above.

PON1 co-expressing genes were obtained from CCLE v2 based on RNA-seq RPKM mRNA abundance data. Interaction rank was based on Spearman's correlation and *p*-value. Quantitative analysis and plotting were done in Python (see code availability). Localization for each gene-encoding protein was queried in the Human Protein Atlas (http://www.proteinatlas.org).

N-glycosylation sites from the PON1 protein sequence were predicted using NetNGlyc (http://www.cbs.dtu.dk/services/NetNGlyc). Folded and charged regions within PON1 were visualized with FoldIndex (http://www.fold.weizmann.ac.il/fldbin/findex) and EMBOSS charge prediction tool (http://www.bioinformatics.nl/cgi-bin/emboss/charge). Functional protein stability and folding effects of specific amino acid substitutions were predicted using MutPred v2 (http://www.mutpred.mutdb.org) and I-Mutant v3 (http://www.gpcr2.biocomp.unibo.it/cgi/predictors/I-Mutant3.0/I-Mutant3.0.cgi).

Cancer dependency profiles were obtained from the DepMap portal (http://www.depmap.org/portal) RNAi screen dataset (CRISPR Avana Public 20Q2). Dependency scores across all cancer types were grouped by lineage type as predefined by DepMap and were subsequently used for correlation analysis with drug response for indicated targeted therapy obtained from GDSC v2 (AUC values). Spearman's correlation coefficients and linear regression-derived *p* values were obtained from

pre-computed associations in the DepMap portal. Lineages with less than four cell lines for a specific gene inquiry were removed from the dataset. Raw essentiality scores were derived from the Profiling Relative Inhibition Simultaneously in Mixtures (PRISM) drug screen and Project Achilles gene dependency screen, both from the Broad Institute.

For patient survival analysis (first progression or RFS), the data were queried in the KM plotter (http://www.kmplot.com/analysis) for lung cancer or pan-cancer. For co-occurrence gene analysis, data from the breast cancer METABRIC cohort were used (accessed from the cBioPortal). The co-occurring genes in patients with indicated fucosylation gene copy number amplification, deep deletion, mRNA upregulation, or mRNA downregulation were stratified.

## Code availability

The main scripts used for data analyses and plotting are described in detail and are available upon request from MBDA or from https://github.com/borrisHUBO/Aldonza-et-al.-bioRxiv (copy archived at *Aldonza, 2023*).

## Acknowledgements

We thank members of Je-Yoel Cho and Yoosik Kim labs for support and feedback. We are grateful to Thannaree Chottitisupawong for establishing lapatinib- and PHA605752-resistant cell lines; Jung-Mo Ahn for help with N-glycosylation peptide mapping; Stephanie Tan and Monica Prayogo for support in secretome preparation; Sang Kook Lee for PC9- and HCC827-derived cell lines and xenografts; Eui-Cheol Shin, Eugene Cho, David Helfman, Seung-Jae Lee, Ki Jun Jeong, and Jinyoung Kang for critical kits, reagents, antibodies, and plasmids; core animal, histology, FACS, and confocal microscopy facilities of SNU College of Pharmacy, SNU College of Veterinary Medicine, KAIST Biological Sciences, and KAIST GSMSE for technical support. We also thank the glycobiology Twitter (#glycotime) and Open Memeing Frame communities for open discussion and experimental advice. This research was supported by the KAIST College of Engineering Global Initiative Convergence Research (grant no. N11190234), the Basic Science Research Program (grant nos. NRF-2019R1C1C1006672 and NRF-2018R1A6A3A01012494), the Bio & Medical Technology Development Program (grant no. NRF-2016M3A9B6026771), the Science Research Center (SRC) Program (grant no. NRF-2021R1A5A1033157) under the Directorate for Basic Research in Science & Engineering, awarded as part of the Comparative Medicine Disease Research Center (CDRC) initiative through the National Research Foundation (NRF) funded by the Korean government's Ministry of Science and ICT, and the Korea Health Technology R&D Project through the Korea Health Industry Development Institute (KHIDI), funded by the Korean government's Ministry of Health & Welfare (grant no. HI14C1324). MBDA is supported by the Hyundai Motor Chung Mong-Koo Foundation Global Scholarship (FHS-20–008).

## Additional information

#### Competing interests

Mark Borris D Aldonza: There is currently a pending patent filing for the biomarker signature revealed in this study for which MBDA is listed as an inventor (application number: 10-2021-0048888 (2021-04-15)). The authors declare that they have no other competing interests. Junghwa Cha: There is currently a pending patent filing for the biomarker signature revealed in this study for which JC is listed as an inventor (application number: 10-2021-0048888 (2021-04-15)). The authors declare that they have no other competing interests. Insung Yong: There is currently a pending patent filing for the biomarker signature revealed in this study for which IY is listed as an inventor (application number: 10-2021-0048888 (2021-04-15)). The authors declare that they have no other competing interests. Dongwook Kim, Hye-Jin Sung: is currently employed at ProtanBio Inc, a disease biomarker venture company of Seoul National University. The authors declare that they have no other competing interests. Pilnam Kim: There is currently a pending patent filing for the biomarker signature revealed in this study for which PK is listed as an inventor (application number: 10-2021-0048888 (2021-04-15)). The authors declare that they have no other competing interests. Je-Yoel Cho: There is currently a pending patent filing for the biomarker signature revealed in this study for which J-YC is listed as an

inventor (application number: 10-2021-0048888 (2021-04-15)). J-YC is CEO of ProtanBio Inc, a disease biomarker venture company of Seoul National University. The authors declare that they have no other competing interests. Yoosik Kim: There is currently a pending patent filing for the biomarker signature revealed in this study for which Yoosik K is listed as an inventor (application number: 10-2021-0048888 (2021-04-15)). The authors declare that they have no other competing interests. The other authors declare that no competing interests exist.

## Funding

| Funder | Grant reference number | Author |
| --- | --- | --- |
| Korea Advanced Institute of Science and Technology | N11190234 | Yoosik Kim |
| National Research Foundation of Korea | NRF-2019R1C1C1006672 | Yoosik Kim |
| National Research Foundation of Korea | NRF-2018R1A6A3A01012494 | Pilnam Kim |
| National Research Foundation of Korea | NRF-2016M3A9B6026771 | Je-Yoel Cho |
| National Research Foundation of Korea | NRF-2021R1A5A1033157 | Je-Yoel Cho |
| Ministry of Health and Welfare | HI14C1324 | Je-Yoel Cho |
| Hyundai Motor Chung Mong-Koo Foundation | Global Scholarship FHS-20–008 | Mark Borris D Aldonza |

The funders had no role in study design, data collection and interpretation, or the decision to submit the work for publication.

## Author contributions

Mark Borris D Aldonza, Conceptualization, Data curation, Formal analysis, Investigation, Methodology, Project administration, Resources, Software, Validation, Visualization, Writing – original draft, Writing – review and editing, Conceived the project, Designed, performed, and analyzed most of the experiments, Wrote the manuscript with Yoosik K and JYC; Junghwa Cha, Formal analysis, Investigation, Methodology, Validation, Visualization, Established the tumor spheroid culture, Performed and analyzed live-cell imaging experiments; Insung Yong, Data curation, Formal analysis, Investigation, Methodology, Validation, Established the tumor spheroid culture, Performed and analyzed live-cell imaging experiments; Jayoung Ku, Data curation, Formal analysis, Visualization, Analyzed RNA-seq data, Helped with RNA-seq data analysis; Pavel Sinitcyn, Software, Formal analysis, Visualization, Methodology, Analyzed LC-MS/MS data, Helped with proteomics data analysis; Dabin Lee, Data curation, Investigation, Performed LC-MS/MS experiments; Ryeong-Eun Cho, Data curation, Investigation, Generated stable RFP cells, Assisted in fucosylation experiments; Roben D Delos Reyes, Software, Formal analysis, Investigation, Assisted in computational and data analysis using public datasets; Dongwook Kim, Resources, Methodology, Advised on LC-MS/MS sampling and experiments; Soyeon Kim, Resources, Data curation, Prepared patient serum samples, Established some resistant cell lines; Minjeong Kang, Data curation, Investigation, Generated stable RFP cells, Assisted in fucosylation experiments; Yongsuk Ku, Data curation, Investigation, Generated stable RFP cells, Assisted in fucosylation experiments; Geonho Park, Data curation, Assisted in RNA work; Hye-Jin Sung, Resources, Methodology, Assisted in generating stable PON1-modified cell lines; Han Suk Ryu, Resources, Methodology, Provided and curated all IRB-approved human cancer patient specimens; Sukki Cho, Resources, Methodology, Provided and curated all IRB-approved human cancer patient specimens; Tae Min Kim, Resources, Data curation, Methodology, Provided and curated all IRB-approved human cancer patient specimens; Pilnam Kim, Resources, Supervision, Funding acquisition, Investigation, Methodology, Writing – original draft, Project administration, Writing – review and editing, Supervised the project; Je-Yoel Cho, Funding acquisition, Investigation, Project administration, Resources, Supervision, Writing – original draft, Writing – review and editing, Supervised the project, Wrote the manuscript with MBDA; Yoosik Kim, Funding acquisition, Investigation, Project administration, Resources,

Supervision, Writing – original draft, Writing – review and editing, Supervised the project, Wrote the manuscript with MBDA

### Author ORCIDs
Mark Borris D Aldonza ⬥ http://orcid.org/0000-0002-0771-129X
Insung Yong ⬥ http://orcid.org/0000-0002-7828-4264
Jayoung Ku ⬥ http://orcid.org/0000-0002-4112-4582
Pavel Sinitcyn ⬥ http://orcid.org/0000-0002-2653-1702
Roben D Delos Reyes ⬥ http://orcid.org/0000-0003-1368-6817
Pilnam Kim ⬥ http://orcid.org/0000-0002-5592-4599
Yoosik Kim ⬥ http://orcid.org/0000-0003-3064-4643

### Ethics
Human subjects: All human blood and tissues from three cohorts of patients diagnosed to have lung adenocarcinoma or squamous cell carcinoma or breast carcinoma were collected and analyzed with approved protocols in accordance with the ethical requirements and regulations of the Institutional Review Board of Seoul National University Hospital after securing written informed consent (IRB Nos. 1104-086-359 and B-1201/143-003).

### Decision letter and Author response
Decision letter https://doi.org/10.7554/eLife.75191.sa1
Author response https://doi.org/10.7554/eLife.75191.sa2

---

## Additional files

### Supplementary files
• Supplementary file 1. Patient clinical information. (a) Information of human lung cancer patients (cohort #1). (b) Clinicopathologic information of human lung cancer patients (cohort #2). (c) Clinicopathologic information of human breast cancer patients (cohort #3). (d) List of siRNA and shRNA sequences. (e) List of oligonucleotide qPCR primers.

• Transparent reporting form

### Data availability
All sequencing data produced for this publication has been deposited to the NCBI Gene Expression Omnibus (GEO) database under the accession number GSE160205. The mass spectrometry proteomics data have been deposited to the ProteomeXchange Consortium via the PRIDE partner repository with the dataset identifier PXD022240. Source data has been provided containing raw images of blots and gels. Other data associated with this study are present in the paper, supplementary files, or source data. Questions regarding the data should be directed to MBDA, Yoosik K, or JYC. Reagents, and cell lines described here are accessible through a materials transfer agreement.

The following datasets were generated:

| Author(s) | Year | Dataset title | Dataset URL | Database and Identifier |
|---|---|---|---|---|
| Sinitcyn P, Kim Y | 2023 | Widespread multi-targeted therapy resistance via drug-induced secretome fucosylation | https://www.ebi.ac.uk/pride/archive/projects/PXD022240 | PRIDE, PXD022240 |
| Aldonza MD, Ku J, Kim Y | 2023 | Widespread multi-targeted therapy resistance via drug-induced secretome fucosylation | https://www.ncbi.nlm.nih.gov/geo/query/acc.cgi?acc=GSE160205 | NCBI Gene Expression Omnibus, GSE160205 |

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
