## [Editor Report]

This study demonstrates that elevated secretome fucosylation is a pan cancer signature of both response and resistance to multiple FDA approved targeted therapies using both disease relevant cell lines and in vivo model systems. The authors go on to identify the antioxidant protein PON1 as a critical regulator of therpy induced changes in the secretome. Lastly, the authors define the resistance associated transcription factors and gene modulated by changes in the secretome. Collectively, these studies have the potential to define the mechanisms of drug resistance and identify novel targetable pathways for cancer treatment.

---

## [Decision Letter]

**Decision letter after peer review:**

Thank you for submitting your article "Widespread multi-targeted therapy resistance via drug-induced secretome fucosylation" for consideration by *eLife*. Your article has been reviewed by 3 peer reviewers, and the evaluation has been overseen by a Reviewing Editor and YM Dennis Lo as the Senior Editor. The following individual involved in review of your submission has agreed to reveal their identity: Yatrik M Shah (Reviewer #3).

Essential revisions:

The reviewers were uniformly positive about the work and the impact that it would have in the cancer field. There were however several concerns raised that should be addressed.

1) A more complete explanation of how targeted therapies increase PON1 fucosylation to drive treatment resistance.

2) Formatting of the data and data presentation should be improved as discussed by both reviewers 1 and 3.

*Reviewer #1 (Recommendations for the authors):*

– The authors widely use dot-scatterplot figures to represent their data, however, this representation is not easily interpreted by readers not used to reading these datasets (vs heatmaps). Additionally, sometimes these plots are overused (could be represented in a simpler format), or data is not sufficiently explained in the text or figure legends.

– Given the density of data, and mixed results of sensitivity/resistance, it may be better to move some of the data from Figure 1 to the supplement, and highlight the key information needed for readers (Figure 1B).

– For blots, samples to be compared direction should be run on the same blot (Figure 1F, for example), and sometimes the blot separation is not clearly indicated (Figure 3H, 4G). Coomassie blue staining in the supplemental figure shows higher staining in the treated samples than the untreated indicating potential for unequal loading, though as they are on separate blots, it's difficult to assess.

– In Figure 4, WT would be a more appropriate term for the control (as all proteins are full length, truncating mutants are not used) and graphs labeled with the construct names (WT, N253G, or N324G) instead of 1, 2, 3 would enhance clarity for readers.

– The transition from Figure 5 to Figure 6 is confusing. The authors already showed that UPR/ATF6 could phenocopy their findings, yet they jump back into another omics approach, which did not seem linked to the previous figure.

– Figure 7 (mechanism schematic) needs a legend and explanation.

*Reviewer #2 (Recommendations for the authors):*

Detailed comments:

Figure 1: 1.A – FUT4 is missing in the GDSC data set, please clarify if it is not present in the data set/what happened.

1.D – The study on DNA methylation of these FUTs seems to be out of context. The study focuses the N-glycome in the secretome, which is a phenomenon at protein level. Although altered fucosylation-related gene expression could result from a number of different reasons, it is not clear why looking at the methylation status is so important here, please justify further.

1.F – There seems to be a decrease in the AAL level in the 60-180 KDa range in the drug treated samples, is there any explanation as to why drug treatment decreased fucosylation of higher molecular weight proteins? The non-treated and drug-treated patient samples should have been run on the same gel. It is unclear how much of the fucosylation-increased/decreased regions of the blots are due to differential gel running conditions. Moreover, the quality of these blots is insufficient for publication and should be run again and imaged at lower exposures. The same issue applies for the blots in Figure 2E.

Figure 2: 2.B – At 00.00 why is the total cell number or spheroid size different in the control vs gefitinib groups? Please clarify if this is a timepoint at a few days after drug treatment; if this is the case, please show the untreated control.

Figure 3: 3.H – The blots shown here are insufficient quality for publication. All of the samples should be shown on the same blot. Please show the PON1 and AAL blot separately; it's not clear if lane 17- 20 actually has PON1 band or if it's just AAL. Furthermore, as this is an IP experiment, blots showing input and loading controls are required. 3.J: It is well established that protein glycosylation occurs in Golgi/ER, so it is unclear what this figure adds to this study. 3. (O-Q) show that PON1 fucosylation requires PON3 but how is unclear. The authors should provide further explanation for the mechanistic role of PON3.

Figure 4: The authors compare the effects of WT vs. N253/324G mutants in vitro; the authors should use the same cell lines in an in vivo model to delineate the contributions to therapeutic resistance.

Figure 5: The authors focused on ATF6 only but is it possible that ATF 3 and 4 might also play a role in combination to 6 since all these are known to play a role in stress response and based on 3.C the fold change differences between these 3 is not that much. The story of ATF6 is not robust here and it feels like it's not adding much to the big picture of the paper.

Figure 6: This figure shows just the superficial screening of how is secreted fucosylated PON1 working, it would have been better if the authors provided some validation of the transcriptomics signature they saw. The validation of downstream mechanism would strengthen the study.

*Reviewer #3 (Recommendations for the authors):*

The manuscript through several unbiased analyses identifies a fucosylation secretome which is altered in drug resistant cancer cells. Moreover, the authors identify PON1 fucosylation as essential for drug sensitivity to targeted therapies. This is highly data rich manuscript that provides an interesting mechanism of drug resistance. The work provides a very precise role of the tumor secretome during drug resistance and relies on not only unbiased cell line assays, but also large cancer databases and human patient data. The major issue with the manuscript is that it is a very difficult manuscript to read and follow. Some of this is due to the sheer amount of data, but also how it is presented and described. In addition, the major experimental concern is the lack of direct evidence that indeed alteration in tumor oxidative stress and inflammation are critical downstream pathways by which PON1 mediates drug resistance.

– The data mining approach to implicate fucosylation in drug resistance is string in Figure 1, however the functional data to demonstrate an increase in fucosylation following TKIs and DR clones is not supported by the data (Figure 1L and M).

– Not very clear why PON1 fucosylation is not lower after PON1 siRNA (Figure 3P).

– More details are needed for figure 5A, it's not very clear what the assay measures. I think in general this is a major issue with the manuscript that most conclusions are made using unbiased approaches but the details on how the data was generated is lacking. Makes the figures very difficult to interpret.

– It is not clear if PON1 fucosylation-induced therapy resistance is an essential mechanism in vivo.

---

## [Author Response]

Reviewer #1 (Recommendations for the authors):– The authors widely use dot-scatterplot figures to represent their data, however, this representation is not easily interpreted by readers not used to reading these datasets (vs heatmaps). Additionally, sometimes these plots are overused (could be represented in a simpler format), or data is not sufficiently explained in the text or figure legends.

We thank the reviewer for bringing up this issue. Given the numeric features and number of samples we want to capture per use of dot plot and to visualize more information in a limited space, and considering the abundance of data in both main and supplement figures, we revised the description across all figures containing dot plots to improve readability and interpretation.

– Given the density of data, and mixed results of sensitivity/resistance, it may be better to move some of the data from Figure 1 to the supplement, and highlight the key information needed for readers (Figure 1B).

We appreciate this suggestion. Following the reviewer’s request, we moved select data (N-glycan release assay and MWCO-sandwich ELLA assay data) to Figure 1 supplements (Figure 1—figure supplement 5) and reorganized Figure 1 so we could highlight three important findings aside from the results of large-scale pharmacogenomics data analysis: (I) consistent specific signatures of fucosylated TIS and DS clone secretomes in lung cancer (LC) patients and pan-cancer cell models; (II) overall abundance and biomarker potential of fucosylated proteins in sera of LC patients that underwent osimertinib therapy and in tissues of sequential therapy-relapsed LC patients (Figure 1K); and (III) specificity of our observed core fucosylation signatures in the context of the secretory process—both in the golgi prior to secretion and in the secretion (Figure 1L and M). We believe the revised Figure 1 emphasize these findings better also considering the substantial amount of data that support the main findings (12 separate figure supplements for Figure 1 alone).

– For blots, samples to be compared direction should be run on the same blot (Figure 1F, for example), and sometimes the blot separation is not clearly indicated (Figure 3H, 4G). Coomassie blue staining in the supplemental figure shows higher staining in the treated samples than the untreated indicating potential for unequal loading, though as they are on separate blots, it's difficult to assess.

We thank the reviewer for noticing this issue. First, we should note that all patient sera samples were originally ran on a single midi-SDS-PAGE gel format but were incised prior to AAL and HRP-streptavidin incubations (incisions and original membranes are shown in Author response image 1).

We added text to indicate this and highlight the blot incisions clearly in the figure description. We replaced the original incised Coomassie blue stain gel for patient sera with a replicate uncut stained midi-gel scanned in a ChemiDoc instead of a printer scanner (Figure 1—figure supplement 4A). In an effort to replicate this in a single mini-SDS-PAGE gel format, we performed additional AAL enrichment and AAL blotting from select similar patient sera samples in Figure 1G (Figure 1—figure supplement 4B). Due to insufficient amounts of the remaining patient serum samples required for enrichments, we were unable to re-process the rest of the samples presented in Figure 1G. Regardless, as shown in Author response image 1, the marked enrichment signature of 30~60 kDa serum protein fucosylation in osimertinib cohort can still be observed albeit the relatively poor quality of the blot and samples. We should note that the serum samples have been in our -80C deep freezer storage for over 5 years since the time we received them from the hospital and have experienced multiple freeze-thaw cycles. These factors could account for the poorer quality of AAL blots.

**Author response image 1. sa2fig1:** 

– In Figure 4, WT would be a more appropriate term for the control (as all proteins are full length, truncating mutants are not used) and graphs labeled with the construct names (WT, N253G, or N324G) instead of 1, 2, 3 would enhance clarity for readers.

We thank the reviewer for this suggestion. We replaced all labeling from “FL” to “WT” in the text and figures according and relabeled the full construct names in the figures.

– The transition from Figure 5 to Figure 6 is confusing. The authors already showed that UPR/ATF6 could phenocopy their findings, yet they jump back into another omics approach, which did not seem linked to the previous figure.

We appreciate the reviewer’s criticism. The experimental goals and co-culture setup to model and modify the TIS-driven expansion of resistant clones are different between Figure 5 and 6. In Figure 5, we aim to address the general mechanism of limited DR clone expansion via TIS de-N-glycosylation by an active PNGase F applied on a direct admixture setup. In this context, we reason that a focused TF screen covering 45 TF-driven intracellular signaling is appropriate to tease out signaling drivers of the DR clones’ response to the expansion-limiting condition. We exhaustively validated the results of the screen in both admixture and CM co-culture set-up, with additional data as a result of this revision (Figure 5 and Figure 5—figure supplement 1).

In Figure 6, we aim to broadly identify resistance-associated modulator genes important for the response of DR clones to the secretomes with altered N-glycosylation relevant to both global secretome N-glycosylation and PON1-specific N-glycosylation (core fucosylation). In this context, we reason that a transcriptome-wide analysis is more appropriate to profile modulator genes in an unbiased manner. As a result of the revision, we also validated the results of RNAseq assays via qPCR and performed additional functional assays to evaluate the effects of RNA-seq hits in our models (Figure 6—figure supplement 2 and 3).

So as not to confuse the readers and improve interpretations, we added texts in the Results and Discussion sections to highlight these differences in terms of experimental goals and set-ups, and mention the overall insights gained from these approaches.

– Figure 7 (mechanism schematic) needs a legend and explanation.

We thank the reviewer for noticing this. We added a description to detail an appropriate summary for the schematic and mechanism model.

Reviewer #2 (Recommendations for the authors):Detailed comments:Figure 1: 1.A – FUT4 is missing in the GDSC data set, please clarify if it is not present in the data set/what happened.

We thank the reviewer for noticing this. Indeed, FUT4 data is not present in the GDSC dataset, in both the old and updated versions. We added text in the figure description to mention this.

1.D – The study on DNA methylation of these FUTs seems to be out of context. The study focuses the N-glycome in the secretome, which is a phenomenon at protein level. Although altered fucosylation-related gene expression could result from a number of different reasons, it is not clear why looking at the methylation status is so important here, please justify further.

We thank the reviewer for asking this important question. There is a growing body of evidence that alterations in glycosylation events and signatures at the protein level are influenced by tight regulation of glycosylation-related genes, such as those that encode for glycoenzymes (such as FUTs), by DNA methylation and other epigenetic mechanisms in disease contexts such as cancer. Since the RRBS data we used for analysis are from CCLE, we made the attempt to test this connection and correlate with drug sensitivity data from CCLE and GDSC. In many cases, FUT methylation profiles contradicted the correlation between FUT gene expression and drug resistance signatures in aggregate. Therefore, we reason that the signatures we found between FUT expression and drug sensitivity can be further supported by the signatures we found between FUT promoter methylation and drug sensitivity, strengthening the potential biological relevance. We added texts in the Results section to highlight this including a key reference.

1.F – There seems to be a decrease in the AAL level in the 60-180 KDa range in the drug treated samples, is there any explanation as to why drug treatment decreased fucosylation of higher molecular weight proteins? The non-treated and drug-treated patient samples should have been run on the same gel. It is unclear how much of the fucosylation-increased/decreased regions of the blots are due to differential gel running conditions. Moreover, the quality of these blots is insufficient for publication and should be run again and imaged at lower exposures. The same issue applies for the blots in Figure 2E.

We thank the reviewer for bringing up this important point and the technical issue on AAL blots. First, because of the varying signatures we have observed in AAL blots, even after independent repetitions, from both patient sera and cell-derived secretomes in terms of the >60 or >100 kDa fucosylation signatures we did not have the confidence to draw conclusions from them, unlike in the case of the consistent and striking differential 30~60 kDa fucosylation signatures we observed in both patient sera and cell-derived secretomes. Similarly, in our new AAL blots from mouse models, it is difficult to draw conclusions about the >60 or >100 kDa fucosylation other than the strong 30~60 kDa fucosylation signatures. In our quantitative measurements via sandwich ELLA, we only observed signatures of decreased >100 kDa in cellderived TIS and DR clone secretomes but not in patient and mouse serum samples (Figure 1— figure supplement 10C and Figure 1—figure supplement 11E). Regardless, we now highlighted this and offered an outlook in the Results section.

Regarding the AAL blot of patient samples, we should note that the samples were originally ran on a single midi-SDS-PAGE gel format but were incised prior to AAL and HRP-streptavidin incubations (incisions and original membranes are shown in Author response image 2). We replaced the incised CBBstained midi SDS-PAGE gel where we ran the original samples with a replicate uncut midi gel (Figure 1—figure supplement 4A). Unfortunately, we were only able to enrich and re-run select patient samples on a single mini-format as other samples had insufficient amounts for AAL enrichment. As shown in Author response image 2, the marked enrichment signature of 30~60 kDa serum protein fucosylation in osimertinib cohort can still be observed albeit the relatively poor quality of the blot and samples. We should note that the serum samples have been in our -80C deep freezer storage for over 5 years since the time we received them from the hospital and have experienced multiple freeze-thaw cycles. These factors could account for the poorer quality of AAL blots.

Regarding the quality of AAL blots for patient serum samples and cell-derived secretomes, we have performed several attempts to improve the overall quality from sampling (i.e., 3K MWCO dialysis, albumin and IgG deletion, shorter lectin enrichment, and multiple flow-through washes prior to elution) to detection of bound AAL (i.e., low concentration of HRP-streptavidin, shorter ECL incubation prior to detection). After all these attempts, the blots shown in the original figures are relatively the best versions of the blots we have acquired. In Author response image 3 you can see blots for patient samples imaged at both normal and shorter exposures.

**Author response image 3. sa2fig3:** 

Figure 2: 2.B – At 00.00 why is the total cell number or spheroid size different in the control vs gefitinib groups? Please clarify if this is a timepoint at a few days after drug treatment; if this is the case, please show the untreated control.

We thank the reviewer for bringing up this point. We should note that these differences are observed because of the timing of imaging. We now highlighted in the figure description that these are only representative live-imaging panels wherein the top panel, we started the imaging experiment prior to the formation and until the settlement of the spheroid (<5.5 hours of formation) while in the lower panel, we started the imaging once the spheroids have formed and settled (after ~6 hours in the incubator). Representative live-imaging videos for this panel are provided as Supplementary files 4 and 5. In all relevant live-imaging based measurements in Figure 2 and associated figure supplements, imaging and quantitative tracking were performed at a similar rate and schedule for both DMSO and EGFR-TKI treatment conditions (representative images in Figure 1—figure supplement 4A).

Figure 3: 3.H – The blots shown here are insufficient quality for publication. All of the samples should be shown on the same blot. Please show the PON1 and AAL blot separately; it's not clear if lane 17- 20 actually has PON1 band or if it's just AAL. Furthermore, as this is an IP experiment, blots showing input and loading controls are required.

We thank the reviewer for bringing up this issue. While AAL blotting of AALenriched patient sera and cell secretomes have been very challenging in terms of acquiring clean blots (as described above), we observed much cleaner blots when performing AAL blotting after PON1 immunoprecipitation. As suggested, we re-assayed select patient serum samples for PON1 immunoprecipitation and downstream AAL blotting and PON1 immunoblotting which now include proper input and loading controls aside from separate blot detection of AAL and PON1 (Figure 3H). We also added text in the Results section to describe N-glycosylation differences between the two secreted PON1 isoforms observed (~55 and ~45 kDa). We moved the original AAL blot after PON1 immunoblotting to the figure supplement (Figure 3—figure supplement 2A).

3.J: It is well established that protein glycosylation occurs in Golgi/ER, so it is unclear what this figure adds to this study.

We appreciate this criticism. In our previous study (Aldonza et al., 2017), we observed that PON1 has both nuclear and cytoplasmic isoforms in cancer cells. Although it can be obvious that PON1 is processed in the golgi since it is a secreted protein, our current data is the first to show that PON1 is actively present in the golgi and that it is core fucosylated in the golgi prior to secretion, at least in the context of targeted therapy and resistance. This is important to specify that golgi-specific isoform of PON1 is heavily fucosylated than the cytoplasmic and nuclear isoforms (Figure 3K), since nuclear proteins can also be glycosylated. This piece of data can also be connected to our observation of increased overall core fucosylation and enrichment of the fucosylated 30~60 kDa proteins in the golgi of drug-treated cells and DR clones (Figure 1L and M).

Reference: Aldonza MBD, Son YS, Sung H, Ahn JM, Choi Y, Kim Y, Cho S, Cho J (2017) Paraoxonase-1 (PON1) induces metastatic potential and apoptosis escape via its antioxidative function in lung cancer cells Oncotarget 8:42817-42835.

3. (O-Q) show that PON1 fucosylation requires PON3 but how is unclear. The authors should provide further explanation for the mechanistic role of PON3.

We thank the reviewer for this important question. Given the known strong regulatory effect of PON3 on PON1’s enzyme activity and our data pointing to the rewired enzyme activity of PON1 when core fucosylated (Figure 4M and Figure 4—figure supplement 1I), we first speculated that regulation of PON1 enzyme activity might be relevant for the aberrant PON1 core fucosylation. To test this, we employed known regulators of PON1 enzyme activity and potential PON1-interacting proteins (PON3, VWF, and MPO), subjected them to multiplex cross-linking co-immunoprecipitation with PON1, and performed relevant downstream assays to evaluate interaction and effects on PON1 levels and PON1-specific core fucosylation (Figure 3—figure supplement 3A). The results, now described in the Results section, showed that PON1-bound positive regulators of PON1 enzyme activity (PON3 and VWF) have stronger interaction with PON1 in drug-treated cells and DR clones than untreated cells, while PON1bound negative regulator of PON1 enzyme activity (MPO) has lesser interaction with PON1 in drug-treated cells and DR clones than untreated cells (Figure 3—figure supplement 3B). These changes in interaction with PON1 are also more pronounced in the golgi fraction and secretomes than in the whole cell fractions. More importantly, PON1-bound positive regulators of PON1 enzyme activity (PON3 and VWF) strongly promoted PON1 core fucosylation while PON1-bound negative regulator of PON1 enzyme activity (MPO) de-N-glycosylated PON1 (Figure 3—figure supplement 3C). These data hint to the possibility that PON3’s regulatory effects on PON1 core fucosylation might be dependent on its effect on PON1 enzyme activity.

Next, we wanted to test whether transcriptional regulation of PON3 is sufficient to direct PON3PON1 interaction and PON3-dependent PON1 core fucosylation. Indeed, transcriptional control of PON3 by SREBP2, a master transcription factor (TF) for PON promoter activity, tightly regulate PON1-PON3 interaction and PON3-mediated PON1 core fucosylation in the golgi and secretomes of drug-treated cells and DR clones (Figure 3—figure supplement 3D and E). We now added these details in the Results section.

Figure 4: The authors compare the effects of WT vs. N253/324G mutants in vitro; the authors should use the same cell lines in an in vivo model to delineate the contributions to therapeutic resistance.

We thank the reviewer for this important request. As suggested, we performed an in vivo xenograft study using PON1-WT and PON1-N253G cell lines (Figure 4—figure supplement 3). PON1-N253G markedly sensitized EGFR-TKI-resistant tumors in mice without observable weight loss (Figure 4—figure supplement 3A and B), and led to systemic decrease of both EGFR-TKI-induced serum PON1 core fucosylation and PON1-bound Nglycans in xenografted mice (Figure 4—figure supplement 3C). These in vivo data strongly corroborate our results in vitro. We now added these details in the Results section.

Additional in vivo studies were also performed to validate key findings presented in Figure 1 (Figure 1—figure supplement 11 and 12).

Figure 5: The authors focused on ATF6 only but is it possible that ATF 3 and 4 might also play a role in combination to 6 since all these are known to play a role in stress response and based on 3.C the fold change differences between these 3 is not that much. The story of ATF6 is not robust here and it feels like it's not adding much to the big picture of the paper.

We thank the reviewer for these suggestions. We agree that aside from ATF6, checking the potential roles of ATF3 and ATF4 in this context is important. We performed additional experiments to check whether ATF3 and ATF4 are important in the intracellular response of DR clones to growth-limiting effects of the globally de-N-glycosylated TIS or inhibited TIS-specific PON1 fucosylation by employing the similar experimental set-up we used to assay ATF6 (Figure 5—figure supplement 1E). TIS de-N-glycosylation or inhibition of TISspecific fucosylated PON1 markedly induced ATF3 and ATF4—both of which contain an amino acid response element (AARE) in their promoters which was one of the hits identified from our TF screen (Figure 5A)—in DR clones but this effect is negligible in the golgi (Figure 5—figure supplement 1K). Regardless, silencing ATF3 or ATF4 decreased intracellular ROS/RNS levels in DR clones co-cultured in modified TIS where fucosylated PON1 is inhibited (Figure 5—figure supplement 1L). These localization-specific effects of ATF6, ATF3, and ATF4 can likely be explained by differences in their retention and translocation in ER-golgi. We now added these details in the Results section. We believe that our validation assays provide a strong suite of preliminary data in characterizing the roles of ATF6 and UPR-associated phenotypes and so are important in contextualizing the result of the focused TF screen in this context. As mentioned by the reviewer, we recognize that such data are not robust enough to deeply elucidate the biology of how UPR and ATF6 facilitate the response of the DR clones to the growth-limiting effects of the de-N-glycosylated TIS or inhibited TIS-specific PON1 core fucosylation. As such, we acknowledge this in the Discussion section.

Figure 6: This figure shows just the superficial screening of how is secreted fucosylated PON1 working, it would have been better if the authors provided some validation of the transcriptomics signature they saw. The validation of downstream mechanism would strengthen the study.

We agree with the reviewer. As suggested, we performed qPCR assay to validate the gene expression of top RNA-seq hits, both the upregulated and downregulated DEGs, in our models (Figure 6—figure supplement 2A). The results strongly corroborated the transcriptomics signatures.

In addition, we investigated whether these hits are functionally implicated in the response of DR clones to TIS fucosylation by employing a CM co-culture assay where RFP-labeled, low-density DR clones were subjected to RNAi or lentiviral target gene overexpression followed by culture in modified TIS and then tracked overtime (Figure 6—figure supplement 3A). Predominantly, RPS27L RNAi and DKK1 overexpression both displayed specific phenotypic responses to TIS de-N-glycosylation or TIS-specific fucosylated PON1 inhibition. In both modified TIS conditions, silencing RPS27L expression or overexpressing DKK1 promoted growth acceleration and EGFR-TKI resistance of DR clones reversing the supposed growth limitation and sensitization under such TIS modifications (Figure 6—figure supplement 3B and C). Further, both targeted approaches prevented senescence induction and stimulated TME-associated resistance genes upon de-N-glycosylation of TIS or inhibition of TIS-specific PON1 fucosylation (Figure 6—figure supplement 3D). We believe that these data strengthen our results especially in indicating that modulatory genes controlling DR clone response to inhibited secretome PON1 fucosylation are functionally associated with drug sensitivity to targeted therapies and are potential therapeutic targets to limit DR clone outgrowth.

Reviewer #3 (Recommendations for the authors):The manuscript through several unbiased analyses identifies a fucosylation secretome which is altered in drug resistant cancer cells. Moreover, the authors identify PON1 fucosylation as essential for drug sensitivity to targeted therapies. This is highly data rich manuscript that provides an interesting mechanism of drug resistance. The work provides a very precise role of the tumor secretome during drug resistance and relies on not only unbiased cell line assays, but also large cancer databases and human patient data. The major issue with the manuscript is that it is a very difficult manuscript to read and follow. Some of this is due to the sheer amount of data, but also how it is presented and described. In addition, the major experimental concern is the lack of direct evidence that indeed alteration in tumor oxidative stress and inflammation are critical downstream pathways by which PON1 mediates drug resistance.

We thank the reviewer for acknowledging the strengths and detailing the weaknesses of our study. As the result of the revision, new data were added to strengthen specific conclusions (summarized at the beginning) and provide insights as to how PON1 core fucosylation is regulated, both directly and indirectly, and how modulator genes control the response of DR clones to the N-glycosylated TIS or TIS-specific core fucosylated PON1.

We also re-organized Figure 1 (having the most figure panels) and moved some panels to the figure supplements so only main results are highlighted in the main figure. We also subdivided some select figure supplements for Figure 1, 3, and 4 to enhance the flow and interpretability of the figures, especially as to how they relate to their respective main figures.

– The data mining approach to implicate fucosylation in drug resistance is string in Figure 1, however the functional data to demonstrate an increase in fucosylation following TKIs and DR clones is not supported by the data (Figure 1L and M).

We appreciate the reviewer’s criticism. The reviewer referenced the AAL blots from secretomes of TKI-treated cells and DR clones (now in Figure 1H and I). It is difficult to draw an overall “increase” or “decrease” signatures from these blots considering the technical, nonquantitative nature of the lectin blotting assay. Therefore, we focused on the differential protein size-specific signatures we can clearly observe from these lectin blots across cell secretomes, patient sera, and mouse sera. We should note that in several cases, in areas (i.e., specific kDa range) where there is an observed overabundance of glycosylated proteins, it is very challenging to perfectly image the whole blots without avoiding the appearance of “white smears” which could be caused by fast consumption of substrate causing absorption of very little light during detection. An example of this is shown in Author response image 4. We observed this even in cases where we exposed the blots for a very short amount of time (<5 seconds) and loaded relatively few amounts of samples (<2 ug of enriched sera or <10 ug of enriched cell secretomes). In such cases, people usually crop the regions they would like to highlight, but we thought presenting the full blot regions is more appropriate in this case.

After countless troubleshooting from sampling to detection, the representative blots shown in our figures were relatively of best quality. We should also note that despite these technical challenges, the differential 30~60 kDa protein fucosylation signature across cell secretomes, patient sera, and mouse sera were consistently observed and unaffected. Therefore, as to not be limited by the results we derive from these lectin blots, we extensively validated both the overall abundance and the differential size-signatures via total or MWCO sampling followed by conducting relevant downstream assays such as sandwich ELLA and N-glycan release assays, both of which were developed in-house.

**Author response image 4. sa2fig4:** 

– Not very clear why PON1 fucosylation is not lower after PON1 siRNA (Figure 3P).

We thank the reviewer for mentioning this result. The mentioned data points to the HLE analysis of PON1 fucosylation in the Golgi/ER fractions of H1993-GR clones upon PON1 siRNA. We got curious as to why the transient PON1 RNAi was not sufficient to inhibit PON1 fucosylation in the Golgi/ER. We assayed the same DR clones upon PON1 RNAi and measured PON1 fucosylation in the whole cell and cytoplasmic intracellular fractions and in cell secretome. Interestingly, while siPON1 was sufficient enough to significantly decrease PON1 fucosylation intracellularly, it was not sufficient to inhibit secretome PON1 fucosylation, corroborating our data on Golgi/ER (Figure 3—figure supplement 2F). Regardless, in a control A549 cell line, which relative expresses PON1 at high levels, stable PON1 RNAi via shRNA led to complete inhibition of PON1 fucosylation in cellular fractions and secretome (Figure 3— figure supplement 2F). Validation of target gene expression control for RNAi studies is shown in Supplementary file 7. We now added these details in the Results section.

– More details are needed for figure 5A, it's not very clear what the assay measures. I think in general this is a major issue with the manuscript that most conclusions are made using unbiased approaches but the details on how the data was generated is lacking. Makes the figures very difficult to interpret.

We appreciate the reviewer’s criticism. We added relevant description to describe the dual-luciferase system-based Cignal 45-pathway reporter array in the figure description and added text in Figure 5 schematic to indicate this.

– It is not clear if PON1 fucosylation-induced therapy resistance is an essential mechanism in vivo.

We agree with the reviewer that validating our PON1-specific results in vivo is important. To address this, we used the same PON1-WT and PON1-mutant DR clones (PON1N253), established tumor xenografts in mouse and subjected them to EGFR-TKI therapy, and performed tumor measurements and fucosylation assays. PON1-N253G markedly sensitized EGFR-TKI-resistant tumors in mice without observable weight loss (Figure 4—figure supplement 3A and B), and led to systemic decrease of both EGFR-TKI-induced serum PON1 core fucosylation and PON1-bound N-glycans in xenografted mice (Figure 4—figure supplement 3C). These in vivo data strongly corroborate our results in vitro. We now added these details in the Results section.

Additional in vivo studies were also performed to validate key findings presented in Figure 1 (Figure 1—figure supplement 11 and 12).